# Structured Semidefinite Programming for Recovering Structured Preconditioners

**Arun Jambulapati**
Simons Institute*
jmblpati@berkeley.edu

**Christopher Musco**
New York University
cmusco@nyu.edu

**Jerry Li**
Microsoft Research
jerrl@microsoft.com

**Kirankumar Shiragur**
Broad Institute of MIT and Harvard[†]
shiragur@stanford.edu

**Aaron Sidford**
Stanford University
sidford@stanford.edu

**Kevin Tian**
University of Texas at Austin[‡]
kjtian@cs.utexas.edu

## Abstract

We develop a general framework for finding approximately-optimal preconditioners for solving linear systems. Leveraging this framework we obtain improved runtimes for fundamental preconditioning and linear system solving problems including:

- **Diagonal preconditioning.** We give an algorithm which, given positive definite $\mathbf{K} \in \mathbb{R}^{d \times d}$ with $\mathrm{nnz}(\mathbf{K})$ nonzero entries, computes an $\epsilon$-optimal diagonal preconditioner in time $\widetilde{O}(\mathrm{nnz}(\mathbf{K}) \cdot \mathrm{poly}(\kappa^\star, \epsilon^{-1}))$, where $\kappa^\star$ is the optimal condition number of the rescaled matrix.

- **Structured linear systems.** We give an algorithm which, given $\mathbf{M} \in \mathbb{R}^{d \times d}$ that is either the pseudoinverse of a graph Laplacian matrix or a constant spectral approximation of one, solves linear systems in $\mathbf{M}$ in $\widetilde{O}(d^2)$ time.

Our diagonal preconditioning results improve state-of-the-art runtimes of $\Omega(d^{3.5})$ attained by general-purpose semidefinite programming, and our solvers improve state-of-the-art runtimes of $\Omega(d^\omega)$ where $\omega > 2.3$ is the current matrix multiplication constant. We attain our results via new algorithms for a class of semidefinite programs (SDPs) we call *matrix-dictionary approximation SDPs*, which we leverage to solve an associated problem we call *matrix-dictionary recovery*.

## 1 Introduction

Preconditioning is a fundamental primitive in the theory and practice of numerical linear algebra, optimization, and data science. Broadly, its goal is to improve conditioning properties (e.g., the range of eigenvalues) of a matrix $\mathbf{M}$ by finding another matrix $\mathbf{N}$ which approximates the inverse of $\mathbf{M}$ and is more efficient to construct and apply than computing $\mathbf{M}^{-1}$. This strategy underpins a variety of popular recently-developed tools, such as adaptive gradient methods for machine learning (e.g., Adagrad and Adam [DHS11, KB15]), and near-linear time solvers for combinatorially-structured

---

*Work completed at Stanford and the University of Washington.

[†]Work completed at Stanford.

[‡]Work completed at Stanford and Microsoft Research.

37th Conference on Neural Information Processing Systems (NeurIPS 2023).

matrices (e.g., graph Laplacians [ST04]). Despite widespread practical adoption of such techniques, there is a surprising lack of provably efficient algorithms for preconditioning.

Our work introduces a new tool, *matrix-dictionary recovery*, and leverages it to obtain the first near-linear time algorithms for several structured preconditioning problems in well-studied applications. Informally, the problem we study is as follows (see Section 4 for the formal definition).

> *Given a matrix* $\mathbf{M}$ *and a "matrix-dictionary"* $\{\mathbf{M}_i\}$, *find the best preconditioner*
> $$\mathbf{N} = \sum_i w_i \mathbf{M}_i \text{ of } \mathbf{M} \text{ expressible as a nonnegative linear combination of } \{\mathbf{M}_i\}. \tag{1}$$

We develop general-purpose solvers for the problem (1). We further apply these solvers to obtain state-of-the-art algorithms for fundamental tasks such as preconditioning linear systems and regression, and approximately recovering structured matrices, including the following results.

- **Diagonal preconditioning.** We consider the classical numerical linear algebra problem of *diagonal preconditioning* [vdS69]. Given $\mathbf{K} \in \mathbb{S}_{\succ \mathbf{0}}^d$, the goal is to find a diagonal $\mathbf{W} \in \mathbb{S}_{\succ \mathbf{0}}^d$ minimizing the condition number of $\mathbf{W}^{\frac{1}{2}} \mathbf{K} \mathbf{W}^{\frac{1}{2}}$. Theorem 1 obtains the first near-linear time algorithms for this problem when the optimal condition number of the rescaled matrix is small.

- **Semi-random regression.** We consider a related problem, motivated by semi-random noise models, which takes full-rank $\mathbf{A} \in \mathbb{R}^{n \times d}$ with $n \geq d$ and seeks $\mathbf{W} \in \mathbb{S}_{\succ \mathbf{0}}^n$ minimizing the condition number of $\mathbf{A}^\top \mathbf{W} \mathbf{A}$. Theorem 2 gives the first near-linear time algorithm for this problem, and applications of it reduce risk bounds for statistical linear regression.

- **Structured linear systems.** We robustify Laplacian system solvers, e.g., [ST04], to obtain near-linear time solvers for systems in dense matrices well-approximated spectrally by Laplacians in Theorem 3. We also give new near-linear time solvers for several families of structured matrices, e.g., dense inverse Laplacians and M-matrices,[4] in Theorems 4 and 5.

For the preconditioning problems considered in Theorems 1, 2, and 3, we give the first runtimes faster than a generic SDP solver, for which, state-of-the-art runtimes [JKL+20, HJS+22] are highly superlinear ($\Omega(d^{3.5})$ for diagonal preconditioning and $\Omega(d^{2\omega})$ for approximating Laplacians, where $d$ is the matrix dimension and $\omega > 2.3$ is the current matrix multiplication constant [AW21]). For the corresponding linear system solving problems in each case, as well as in Theorems 4 and 5, the prior state-of-the-art was to treat the linear system as generic and ran in $\Omega(d^\omega)$ time.

**Organization.** We begin by overviewing the main applications of our matrix-dictionary recovery framework in Sections 2 and 3, which respectively cover our results on diagonal preconditioning and structured linear algebra. These sections are self-contained (with some definitions of the matrix families we study in Section 3 deferred to the supplementary material), and can be read independently. In Section 4, we contextualize and formalize the general matrix-dictionary recovery problem (1) we introduce and study. We also provide our main meta-algorithm and its guarantees, and an overview of how the results of Sections 2 and 3 follow from applications of it. We finally compare our framework to related algorithms and give a more thorough runtime comparison in Section 5.

**Notation.** In Section 2 (focusing on diagonal preconditioning and semi-random regression) only, we refer to the matrices to be preconditioned as $\mathbf{K}$ or $\mathbf{A}$. This is for consistency with the numerical linear algebra literature, where $\mathbf{K}$ represents a positive definite kernel matrix, and $\mathbf{A}$ denotes the data in a regression problem $\min_x \|\mathbf{A}x - b\|_2$. In the rest of the paper, our notation will be consistent with (1). The $d \times d$ symmetric matrices are $\mathbb{S}^d$, the positive semidefinite (PSD) and definite (PD) cones are $\mathbb{S}_{\succeq \mathbf{0}}^d$ and $\mathbb{S}_{\succ \mathbf{0}}^d$; the remainder of our notation is standard and deferred to Section 2 of the supplement.

## 2 Diagonal preconditioning

When solving linear systems via iterative methods, one of the most popular preconditioning strategies is to use a diagonal matrix. This is appealing because diagonal matrices can be applied and inverted quickly. Determining the best diagonal preconditioner is a classical numerical linear algebra problem studied since the 1950s [FS55, vdS69, PG90], and has gained recent popularity due to its use in adaptive gradient methods [DHS11, KB15]. In Section 4, we discuss how diagonal preconditioning

---

[4]Inverse M-matrices are necessarily dense, see Appendix B of the supplementary material.

is an instance of (1) in the matrix-dictionary $\mathbf{M}_i = e_i e_i^\top$, where $e_i$ is the $i^{\text{th}}$ basis vector. Leveraging this viewpoint, we design algorithms for two natural instantiations of diagonal preconditioning.

**Outer scaling.** One formulation of the optimal diagonal preconditioning problem, which we refer to as *outer scaling*, asks to optimally reduce the condition number of positive definite $\mathbf{K} \in \mathbb{R}^{d \times d}$ with a diagonal matrix $\mathbf{W}$, i.e., return diagonal $\mathbf{W} = \mathbf{diag}\,(w)$ for $w \in \mathbb{R}^d_{>0}$ such that[5]

$$\kappa(\mathbf{W}^{\frac{1}{2}}\mathbf{K}\mathbf{W}^{\frac{1}{2}}) \approx \kappa_o^\star(\mathbf{K}) := \min_{\text{diagonal } \mathbf{W} \succ \mathbf{0}} \kappa(\mathbf{W}^{\frac{1}{2}}\mathbf{K}\mathbf{W}^{\frac{1}{2}}).$$

Given $\mathbf{W}$, a solution to $\mathbf{K}x = b$ can be obtained by solving the better-conditioned $\mathbf{W}^{\frac{1}{2}}\mathbf{K}\mathbf{W}^{\frac{1}{2}}y = \mathbf{W}^{\frac{1}{2}}b$ and returning $x = \mathbf{W}^{\frac{1}{2}}y$. The optimal $\mathbf{W}$ can be obtained via a semidefinite program (SDP) [QYZ20], but the computational cost of general-purpose SDP solvers outweighs benefits for solving linear systems. Outer scaling is poorly understood algorithmically; prior to our work, even attaining a constant-factor approximation to $\kappa_o^\star(\mathbf{K})$ without a generic SDP solver was unknown.

This state of affairs has resulted in the widespread use of heuristics for constructing $\mathbf{W}$, such as *Jacobi preconditioning* [vdS69, GR89] and *matrix scaling* [AZLOW17, CMTV17a, GO18]. The former strategy, where the preconditioner is taken as the inverse diagonal to $\mathbf{K}$, was notably highlighted by Adagrad [DHS11], which used Jacobi preconditioning to improve computational costs. However, both heuristics have clear drawbacks from theoretical or practical perspectives.

Prior to our work the best approximation guarantee known for Jacobi preconditioning was a result of van der Sluis [vdS69, GR89], which shows the Jacobi preconditioner is an $m$-factor approximation to the optimal preconditioning problem where $m \leq d$ is the maximum number of non-zeros in any row of $\mathbf{K}$: in dense matrices this is linear in the problem dimension and can be much larger than $\kappa_o^\star(\mathbf{K})$. We review and slightly strengthen this result in Appendix C of the supplement. We also prove a new *dimension-independent* baseline result of independent interest: the Jacobi preconditioner obtains condition number no worse than $(\kappa_o^\star(\mathbf{K}))^2$. Unfortunately, we exhibit a family of matrices showing this bound is tight, dashing hopes they solve outer scaling near-optimally. On the other hand, while sometimes effective as a heuristic [KRU14], matrix scaling algorithms target a different objective (normalizing row and column sums) and do not yield provable guarantees on $\kappa(\mathbf{W}^{\frac{1}{2}}\mathbf{K}\mathbf{W}^{\frac{1}{2}})$.

**Inner scaling.** Another formulation of diagonal preconditioning, which we refer to as *inner scaling*, takes as input a full-rank $\mathbf{A} \in \mathbb{R}^{n \times d}$ and asks to find an $n \times n$ positive diagonal $\mathbf{W}$ with

$$\kappa(\mathbf{A}^\top \mathbf{W}\mathbf{A}) \approx \kappa_i^\star(\mathbf{A}) := \min_{\text{diagonal } \mathbf{W} \succ \mathbf{0}} \kappa(\mathbf{A}^\top \mathbf{W}\mathbf{A}).$$

As a comparison, when outer scaling is applied to the kernel matrix $\mathbf{K} = \mathbf{A}^\top \mathbf{A}$, $\mathbf{W}^{\frac{1}{2}}\mathbf{K}\mathbf{W}^{\frac{1}{2}}$ can be seen as rescaling the columns of $\mathbf{A}$. On the other hand, in inner scaling we instead rescale rows of $\mathbf{A}$. Inner scaling has natural applications to improving risk bounds in a robust statistical variant of linear regression, which we comment upon shortly. Nonetheless, as in the outer scaling case, no algorithms faster than general SDP solvers are known to obtain even a constant-factor approximation to $\kappa_i^\star(\mathbf{A})$. Further, despite clear problem similarities, it is unclear how to best extend heuristics (e.g., Jacobi preconditioning and matrix scaling) for outer scaling to inner scaling.

**Our results.** We give the first nontrivial approximation algorithms (beyond generic SDP solvers) for both the outer and inner scaling problems, yielding diagonal preconditioners attaining constant-factor approximations to $\kappa_o^\star$ and $\kappa_i^\star$ in near-linear time.[6] $\mathcal{T}_{\text{mv}}(\mathbf{M})$ denotes the time required to multiply a vector by $\mathbf{M}$; this is at most the sparsity of $\mathbf{M}$, but can be substantially faster for structured $\mathbf{M}$.

**Theorem 1** (Outer scaling). *Let $\epsilon > 0$ be a fixed constant.[7] There is an algorithm, which given full-rank $\mathbf{K} \in \mathbb{S}^d_{\succ \mathbf{0}}$ computes $w \in \mathbb{R}^d_{\geq 0}$ such that $\kappa(\mathbf{W}^{\frac{1}{2}}\mathbf{K}\mathbf{W}^{\frac{1}{2}}) \leq (1 + \epsilon)\kappa_o^\star(\mathbf{K})$ with probability $\geq 1 - \delta$ in time $O(\mathcal{T}_{\text{mv}}(\mathbf{K}) \cdot (\kappa_o^\star(\mathbf{K}))^{1.5} \cdot \text{polylog}(\frac{d\kappa_o^\star(\mathbf{K})}{\delta}))$.*

**Theorem 2** (Inner scaling). *Let $\epsilon > 0$ be a fixed constant. There is an algorithm, which given full-rank $\mathbf{A} \in \mathbb{R}^{n \times d}$ for $n \geq d$ computes $w \in \mathbb{R}^n_{\geq 0}$ such that $\kappa(\mathbf{A}^\top \mathbf{W}\mathbf{A}) \leq (1 + \epsilon)\kappa_i^\star(\mathbf{A})$ with probability $\geq 1 - \delta$ in time $O(\mathcal{T}_{\text{mv}}(\mathbf{A}) \cdot (\kappa_i^\star(\mathbf{A}))^{1.5} \cdot \text{polylog}(\frac{n\kappa_i^\star(\mathbf{A})}{\delta}))$.*

---

[5]$\kappa(\mathbf{M})$ is the condition number of positive definite $\mathbf{M}$, i.e. the eigenvalue ratio $\lambda_{\max}(\mathbf{M})/\lambda_{\min}(\mathbf{M})$.

[6]We are not currently aware of a variant of our matrix dictionary recovery framework which extends to simultaneous inner and outer scaling, though it is worth noting that prior work [QGH+22] does obtain such a result via semidefinite programming. Obtaining such a variant is an interesting open problem for future work.

[7]We do not focus on the $\epsilon$ dependence and instead take it to be constant since, in applications involving solving linear systems, there is little advantage to obtaining better than a 2-approximation (i.e., $\epsilon = 1$).

Our methods pay a small polynomial overhead in the quantities $\kappa_o^\star$ and $\kappa_i^\star$, but notably suffer no dependence on the *original conditioning* of the matrices. Typically, the interesting use case for diagonal preconditioning is when $\kappa_o^\star(\mathbf{K})$ or $\kappa_i^\star(\mathbf{A})$ is small but $\kappa(\mathbf{K})$ or $\kappa(\mathbf{A}^\top\mathbf{A})$ is large, a regime where our runtimes are near-linear and substantially faster than directly applying iterative methods.

It is worth noting that in light of our new results on Jacobi preconditioning, the end-to-end runtime of Theorem 1 for solving linear systems (rather than optimal preconditioning) can be improved: accelerated gradient methods on a preconditioned system with condition number $(\kappa_o^\star)^2$ have runtimes scaling as $\kappa_o^\star$. That said, when repeatedly solving multiple systems in the same matrix, Theorem 1 may offer an advantage over Jacobi preconditioning. Our framework also gives a potential route to achieve the optimal end-to-end runtime scaling as $\sqrt{\kappa_o^\star}$, detailed in Appendix D of the supplement.

Beyond that which is obtainable by black-box using general SDP solvers, we are not aware of any other claimed runtime in the literature for solving the inner and outer scaling problems considered in Theorems 1 and 2. Directly using state-of-the-art SDP solvers [JKL+20, HJS+22] incurs substantial overhead $\Omega(n^\omega\sqrt{d} + nd^{2.5})$ or $\Omega(n^\omega + d^{4.5} + n^2\sqrt{d})$, where $\omega < 2.372$ is the current matrix multiplication constant [Wil12, Gal14, AW21, DWZ23, WXXZ23]. For outer scaling, where $n = d$, this implies an $\Omega(d^{3.5})$ runtime; for other applications, e.g., preconditioning $d \times d$ perturbed Laplacians where $n = d^2$, the runtime is $\Omega(d^{2\omega})$. Applying state-of-the-art approximate SDP solvers (rather than our custom ones, i.e., Theorems 6 and 7) appears to yield runtimes $\Omega(\mathrm{nnz}(\mathbf{A}) \cdot d^{2.5})$, as described in Appendix E.2 of [LSTZ20]. This is in contrast with our Theorems 1, 2 which achieve $\widetilde{O}\left(\mathrm{nnz}(\mathbf{A}) \cdot (\kappa^\star)^{1.5}\right)$. Hence, we improve existing tools by $\mathrm{poly}(d)$ factors in the main regime of interest where the optimal rescaled condition number $\kappa^\star$ is small. Concurrent to our work, [QGH+22] gave algorithms for constructing optimal diagonal preconditioners using interior point methods for SDPs, which run in at least the superlinear times discussed previously.

**Statistical aspects of preconditioning.** Unlike an outer scaling, a good inner scaling does not speed up a least squares regression problem $\min_x \|\mathbf{A}x - b\|_2$. Instead, it allows for a faster solution to the reweighted problem $\min_x \|\mathbf{W}^{\frac{1}{2}}(\mathbf{A}x - b)\|_2$. This has a number of implications from a statistical perspective. We explore an interesting connection between inner scaling preconditoning and *semi-random* noise models for least-squares regression, situated in the literature in Section 5.

As a motivating example of our noise model, consider the case when there is a hidden parameter vector $x_{\text{true}} \in \mathbb{R}^d$ that we want to recover, and we have a "good" set of consistent observations $\mathbf{A}_g x_{\text{true}} = b_g$, in the sense that $\kappa(\mathbf{A}_g^\top\mathbf{A}_g)$ is small. Here, we can think of $\mathbf{A}_g$ as being drawn from a well-conditioned distribution. Now, suppose an adversary gives us a superset of these observations $(\mathbf{A}, b)$ such that $\mathbf{A}x_{\text{true}} = b$, and $\mathbf{A}_g$ are an (unknown) subset of rows of $\mathbf{A}$, but $\kappa(\mathbf{A}^\top\mathbf{A}) \gg \kappa(\mathbf{A}_g^\top\mathbf{A}_g)$. This can occur when rows are sampled from heterogeneous sources. Perhaps counterintuitively, by giving additional consistent data, the adversary can arbitrarily hinder the cost of iterative methods. This failure can be interpreted as being due to overfitting to generative assumptions (e.g., sampling rows from a well-conditioned covariance, instead of a mixture): standard iterative methods assume too much structure, where ideally they would use as little as information-theoretically possible.

Our inner scaling methods can be viewed as "robustifying" linear system solving to such semi-random noise models (by finding $\mathbf{W}$ yielding a rescaled condition number comparable or better than the indicator of the rows of $\mathbf{A}_g$, which are not known a priori). In Section 6 of the supplement, we demonstrate applications of inner scaling in reducing the mean-squared error risk in statistical regression settings encompassing our semi-random noise model, where the observations $b$ are corrupted by (homoskedastic or heteroskedastic) noise. In all settings, our preconditioning algorithms yield computational gains, improved risk bounds, or both, by factors of roughly $\kappa(\mathbf{A}^\top\mathbf{A})/\kappa_i^\star(\mathbf{A})$.

## 3 Robust linear algebra for structured matrices

Over the past decade, the theoretical computer science and numerical linear algebra communities have dedicated substantial effort to developing faster solvers for regression problems in various families of combinatorially-structured matrices. Perhaps the most prominent example is [ST04], who gave a near-linear time solver for linear systems in graph Laplacian matrices.[8] A long line of exciting

---

[8]We recall definitions of the families of matrices we study here. We call a square matrix $\mathbf{A}$ a Z-matrix if $\mathbf{A}_{ij} \leq 0$ for all $i \neq j$. We call a matrix $\mathbf{L}$ a Laplacian if it is a symmetric Z-matrix with $\mathbf{L}\mathbb{1} = 0$. We call $\mathbf{M}$ an invertible $M$-matrix if $\mathbf{M} = s\mathbf{I} - \mathbf{A}$ where $s > 0$, $\mathbf{A} \in \mathbb{R}_{\geq 0}^{d \times d}$, and $\rho(\mathbf{A}) < s$ where $\rho$ is the spectral radius.

work has obtained improved solvers for these systems [KMP10, KMP11, KOSZ13, LS13, CKM$^+$14, PS14, KLP$^+$16, KS16, JS21], which have been used to improve the runtimes for a wide variety of graph-structured problems, including maximum flow [CKM$^+$11, Mad13, LS14], sampling random spanning trees [KM09, MST15, DKP$^+$17, Sch18], graph clustering [ST04, OV11, OSV12], and more [DS08, KRSS15, CMSV17, CMTV17b]. Additionally, efficient linear system solvers have been developed for solving systems in other types of structured matrices, e.g., block diagonally dominant systems [KLP$^+$16], M-matrices [AJSS19], and directed Laplacians [CKP$^+$16, CKP$^+$17, CKK$^+$18].

**Perturbations of structured matrices.** Despite the importance of these matrices with combinatorial structure, previously-developed solvers are in some ways quite brittle. For example, there are simple matrix families closely related to Laplacians for which the best-known runtimes for solving linear systems are achieved by ignoring problem structure, and using generic matrix multiplication techniques as a black box. Perhaps the simplest example is solving systems in *perturbed Laplacians*, i.e., matrices which admit constant-factor approximations by a Laplacian matrix, but which are not Laplacians themselves. This situation can arise when a Laplacian is used to approximate a physical phenomenon [BHV08]. We show that the framework we develop for (1) yields, as a consequence, robustifications and recovery routines building upon previously-developed solvers for structured linear systems. As a first example, we give the following perturbed Laplacian solver.

**Theorem 3** (Perturbed Laplacian solver). *Let $\mathbf{M} \succeq \mathbf{0} \in \mathbb{R}^{n \times n}$ be such that there exists an (unknown) Laplacian $\mathbf{L}$ with $\mathbf{M} \preceq \mathbf{L} \preceq \kappa^\star \mathbf{M}$, and that $\mathbf{L}$ corresponds to a graph with edge weights between $w_{\min}$ and $w_{\max}$, with $\frac{w_{\max}}{w_{\min}} \leq U$. For any $\delta \in (0, 1)$ and $\epsilon > 0$, there is an algorithm recovering a Laplacian $\mathbf{L}'$ with $\mathbf{M} \preceq \mathbf{L}' \preceq (1 + \epsilon)\kappa^\star \mathbf{M}$ with probability $\geq 1 - \delta$ in time $O(n^2 \cdot (\kappa^\star)^2 \cdot \text{poly}(\frac{\log \frac{n\kappa^\star U}{\delta}}{\epsilon}))$. Consequently, there is an algorithm for solving linear systems in $\mathbf{M}$ to $\epsilon$-relative accuracy with probability $\geq 1 - \delta$, in time $O(n^2 \cdot (\kappa^\star)^2 \cdot \text{polylog}(\frac{n\kappa^\star U}{\delta\epsilon}))$.[9]*

Theorem 3 can be viewed as solving a preconditioner construction problem, where we know there exists a Laplacian matrix $\mathbf{L}$ which spectrally resembles $\mathbf{M}$, and wish to efficiently recover a Laplacian with similar guarantees. Our matrix-dictionary recovery framework (1) captures the setting of Theorem 3 by leveraging an appropriate matrix-dictionary of edge Laplacians, discussed in Section 4. The conceptual message of Theorem 3 is that near-linear time solvers for Laplacians robustly extend through our preconditioning framework to efficiently solve matrices approximated by Laplacians. Beyond this specific application, our framework could be used to solve perturbed generalizations of future families of structured matrices.

**Recovery of structured matrices.** In addition to directly spectrally approximating and solving in matrices which are well-approximated by preconditioners with diagonal or combinatorial structure, our framework also yields solvers for new families of matrices. We show that our preconditioning techniques can be used in conjunction with properties of graph-structured matrices to provide solvers and spectral approximations for *inverse M-matrices* and *Laplacian pseudoinverses*. Recovering Laplacians from their pseudoinverses and solving linear systems in the Laplacian pseudoinverse arise when trying to fit a graph to data or recover a graph from effective resistances, a natural distance measure (see [HMMT18] for motivation and discussion of related problems). More broadly, the problem of solving linear systems in inverse symmetric M-matrices is prevalent and corresponds to statistical inference problems involving distributions that are multivariate totally positive of order 2 (MTP$_2$) [KR83, SH14, FLS$^+$17]. Our main results are the following.

**Theorem 4** (M-matrix recovery and inverse M-matrix solver). *Let $\mathbf{M}$ be the inverse of an unknown invertible symmetric M-matrix, let $\kappa$ upper bound its condition number, and let $U$ be the multiplicative range of $\mathbf{M}\mathbb{1}$.[10] For any $\delta \in (0, 1)$ and $\epsilon > 0$, there is an algorithm recovering a $(1 + \epsilon)$-spectral approximation to $\mathbf{M}^{-1}$ in time $O(n^2 \cdot \text{poly}(\frac{\log \frac{n\kappa U}{\delta}}{\epsilon}))$. Consequently, there is an algorithm for solving linear systems in $\mathbf{M}$ to $\epsilon$-relative accuracy with probability $\geq 1 - \delta$, in time $O(n^2 \cdot \text{polylog}(\frac{n\kappa U}{\delta\epsilon}))$.*

**Theorem 5** (Laplacian recovery and Laplacian pseudoinverse solver). *Let $\mathbf{M}$ be the pseudoinverse of unknown Laplacian $\mathbf{L}$, and that $\mathbf{L}$ corresponds to a graph with edge weights between $w_{\min}$ and $w_{\max}$, with $\frac{w_{\max}}{w_{\min}} \leq U$. For any $\delta \in (0, 1)$ and $\epsilon > 0$, there is an algorithm recovering a Laplacian $\mathbf{L}'$ with $\mathbf{M}^\dagger \preceq \mathbf{L}' \preceq (1 + \epsilon)\mathbf{M}^\dagger$ in time $O(n^2 \cdot \text{poly}(\frac{\log \frac{nU}{\delta}}{\epsilon}))$. Consequently, there is an algorithm for solving linear systems in $\mathbf{M}$ to $\epsilon$-relative accuracy with probability $\geq 1 - \delta$, in time $O(n^2 \cdot \text{polylog}(\frac{nU}{\delta\epsilon}))$.*

---

[9]See (6) and the following discussion for the definition of solving to relative accuracy.

[10]$\mathbf{M}\mathbb{1}$ is entrywise positive, as shown in Appendix B of the supplement.

Theorems 4 and 5 are perhaps a surprising demonstration of the utility of our techniques: just because a matrix family is well-approximated by structured preconditioners, it is not a priori clear that their inverses also are. However, we show that by applying recursive preconditioning tools in conjunction with our recovery methods, we can obtain analogous results for these inverse families. These results add to the extensive list of combinatorially-structured matrix families admitting efficient linear algebra primitives. We view our approach as a proof-of-concept of further implications in designing near-linear time system solvers for structured families via algorithms for (1).

Similarly to our results in Section 2, our results on solving matrix-dictionary recovery for graph-structured matrices (Theorems 3, 4, and 5) are the first we are aware of with runtimes improving upon black-box generic algorithms. In particular, for key matrices in each of these cases (e.g., constant-factor spectral approximations of Laplacians, inverse M-matrices, and Laplacian pseudoinverses) we obtain $\widetilde{O}(n^2)$ time algorithms for solving linear systems in these matrices to inverse polynomial accuracy. This runtime is near-linear when the input is dense and in each case when the input is dense the state-of-the-art prior methods were to run general linear system solvers using $O(n^\omega)$ time.

# 4 Matrix-dictionary recovery: a general preconditioning framework

Our general strategy for matrix-dictionary recovery, i.e., recovering preconditioners in the sense of (1), is via applications of a new custom approximate solver we develop for a family of structured SDPs. SDPs are fundamental optimization problems that have been the source of extensive study for decades [VB96], with numerous applications across operations research and theoretical computer science [GW95], statistical modeling [WSV00, GM12], and machine learning [RSL18]. Though there have been recent advances in solving general SDPs (e.g., [JKL+20, HJS+22] and references therein), the current state-of-the-art solvers have superlinear runtimes, prohibitive in large-scale applications. Consequently, there has been extensive research on designing faster approximate SDP solvers under different assumptions [KV05, WK06, AK07, BBN13, GHM15, AL17, CDST19].

We now provide context for our solver for structured "matrix-dictionary approximation" SDPs, state our algorithm and its guarantees, and summarize how it is used to obtain Theorems 1, 2, 3, 4, and 5.

**Positive SDPs.** One prominent class of structured SDPs are what we refer to as *positive SDPs*, namely SDPs in which the cost and constraint matrices are all positive semidefinite (PSD), a type of structure present in many important applications [GW95, ARV09, JJUW11, LS17, CG18, CDG19, CFB19, CMY20], including those in this paper. Positive SDPs generalize positive linear programming, itself a well-studied problem over the past several decades [LN93, PST95, You01, MRWZ16, AO19]. It was recently shown that a prominent special case of positive SDPs known as *packing SDPs* can be solved in nearly-linear time [ALO16, PTZ16], a fact that has had numerous applications in robust learning and estimation [CG18, CDG19, CFB19, CMY20] as well as in combinatorial optimization [LS17]. However, extending known packing SDP solvers to broader classes of positive SDPs, e.g. mixed packing-covering SDPs has been elusive [JY12, JLL+20], and is a key open problem in the algorithmic theory of structured optimization.[11] The mixed packing-covering SDP problem is parameterized by "packing" and "covering" matrices $\{\mathbf{P}_i\}_{i\in[n]}, \mathbf{P}, \{\mathbf{C}_i\}_{i\in[n]}, \mathbf{C} \in \mathbb{S}^d_{\succeq\mathbf{0}}$, and asks to find the smallest $\mu > 0$ such that there exists $w \in \mathbb{R}^n_{\geq 0}$ with $\sum_{i\in[n]} w_i\mathbf{P}_i \preceq \mu\mathbf{P}$ (packing into $\mu\mathbf{P}$) and $\sum_{i\in[n]} w_i\mathbf{C}_i \succeq \mathbf{C}$ (covering $\mathbf{C}$). Redefining $\mathbf{P}_i \leftarrow \frac{1}{\mu}\mathbf{P}^{-\frac{1}{2}}\mathbf{P}_i\mathbf{P}^{-\frac{1}{2}}$, $\mathbf{C}_i \leftarrow \mathbf{C}^{-\frac{1}{2}}\mathbf{C}_i\mathbf{C}^{-\frac{1}{2}}$ for all $i \in [n]$, (a slight strengthening of) this problem is equivalent to finding $w \in \mathbb{R}^n_{\geq 0}$ such that

$$\sum_{i\in[n]} w_i\mathbf{P}_i \preceq \mathbf{I} \preceq \sum_{i\in[n]} w_i\mathbf{C}_i, \tag{2}$$

or refuting its existence. This was studied by [JY12, JLL+20], and an important open problem in structured convex programming is designing a "width-independent" solver for testing feasibility of (2) up to a $1 + \epsilon$ factor (i.e. testing whether (2) is approximately feasible with an iteration count polynomial in $\epsilon^{-1}$ and polylogarithmic in other parameters). Such solvers have remained elusive beyond pure packing SDPs [ALO16, PTZ16, JLT20], even for basic extensions such as pure covering.

**Matrix-dictionary approximation SDPs.** In Theorem 6, we develop our main meta-algorithm, an efficient solver for specializations of (2) where the packing and covering matrices $\{\mathbf{P}_i\}_{i\in[n]}, \{\mathbf{C}_i\}_{i\in[n]}$

---
[11]A faster solver for general positive (mixed packing-covering) SDPs was claimed in [JLL+20], but an error was later discovered in that work, as is recorded in the most recent arXiv version [JLL+21].

are multiples of each other. As we will see, this family of structured SDPs, which we call *matrix-dictionary approximation SDPs*, is highly effective for capturing the forms of approximation required by (1). All our aforementioned preconditioning results follow via careful applications of our matrix-dictionary approximation SDP solver in Theorem 6 (and a generalization of it in Theorem 7).

Specifically, we develop efficient algorithms for the following main meta-problem we study. Given a set of matrices (a "matrix-dictionary") $\{\mathbf{M}_i\}_{i\in[n]} \in \mathbb{S}_{\succeq \mathbf{0}}^d$, a constraint matrix $\mathbf{B} \in \mathbb{S}_{\succ \mathbf{0}}^d$, a tolerance parameter $\epsilon \in (0,1)$, and $\kappa^\star \geq 1$ such that there exists $w^\star \in \mathbb{R}_{\geq 0}^n$ with

$$\mathbf{B} \preceq \sum_{i\in[n]} w_i^\star \mathbf{M}_i \preceq \kappa^\star \mathbf{B}, \tag{3}$$

the goal of matrix-dictionary approximation is to return weights $w \in \mathbb{R}_{\geq 0}^n$ such that

$$\mathbf{B} \preceq \sum_{i\in[n]} w_i \mathbf{M}_i \preceq (1+\epsilon)\kappa^\star \mathbf{B}. \tag{4}$$

When $\mathbf{B} = \mathbf{I}$, the problem in (3), (4) is a special case of (2) where each $\mathbf{M}_i = \mathbf{C}_i = \kappa^\star \mathbf{P}_i$; we call this the isotropic case. We further handle general $\mathbf{B}$, and demonstrate that our formulation captures many interesting applications. We refer to the problem in (3), (4) as *matrix-dictionary recovery*.

**Our results: matrix-dictionary recovery.**   Our results concerning (3) and (4) assume that the matrix-dictionary $\{\mathbf{M}_i\}_{i\in[n]}$ is "simple" in two respects. First, we assume that we have explicit factorizations

$$\mathbf{M}_i = \mathbf{V}_i \mathbf{V}_i^\top, \ \mathbf{V}_i \in \mathbb{R}^{d\times m}. \tag{5}$$

Our applications in Sections 2 and 3 satisfy this assumption with $m = 1$. Second, denoting $\mathcal{M}(w) := \sum_{i\in[n]} w_i \mathbf{M}_i$, we assume we can approximately solve systems in $\mathcal{M}(w) + \lambda \mathbf{I}$ for any $w \in \mathbb{R}_{\geq 0}^n$ and $\lambda \geq 0$. Concretely, for any $\epsilon > 0$, we assume there is a linear operator $\widetilde{\mathcal{M}}_{w,\lambda,\epsilon}$ which we can compute and apply in $\mathcal{T}_{\mathcal{M}}^{\mathrm{sol}} \cdot \log \frac{1}{\epsilon}$ time,[12] and that $\widetilde{\mathcal{M}}_{w,\lambda,\epsilon} \approx (\mathcal{M}(w) + \lambda \mathbf{I})^{-1}$ in that:

$$\left\| \widetilde{\mathcal{M}}_{w,\lambda,\epsilon} v - (\mathcal{M}(w) + \lambda \mathbf{I})^{-1} v \right\|_2 \leq \epsilon \left\| (\mathcal{M}(w) + \lambda \mathbf{I})^{-1} v \right\|_2 \text{ for all } v \in \mathbb{R}^d. \tag{6}$$

In this case, we say "we can solve in $\mathcal{M}$ to $\epsilon$-relative accuracy in $\mathcal{T}_{\mathcal{M}}^{\mathrm{sol}} \cdot \log \frac{1}{\epsilon}$ time." If $\mathcal{M}$ is a single matrix $\mathbf{M}$, we say "we can solve in $\mathbf{M}$ to $\epsilon$-relative accuracy in $\mathcal{T}_{\mathbf{M}}^{\mathrm{sol}} \cdot \log \frac{1}{\epsilon}$ time." Notably, for the matrix-dictionaries in our applications, e.g., diagonal 1-sparse matrices or edge Laplacians, such access to $\{\mathbf{M}_i\}_{i\in[n]}$ exists so we obtain end-to-end efficient algorithms. Ideally (for near-linear time algorithms), $\mathcal{T}_{\mathbf{M}}^{\mathrm{sol}}$ is roughly the total sparsity of $\{\mathbf{M}_i\}_{i\in[n]}$, which holds in all our applications.

Under these assumptions, we give the following novel meta-solvers for matrix-dictionary recovery.[13]

**Theorem 6** (Matrix dictionary recovery, isotropic case). *Given matrices $\{\mathbf{M}_i\}_{i\in[n]}$ with explicit factorizations* (5)*, such that* (3) *is feasible for $\mathbf{B} = \mathbf{I}$ and some $\kappa^\star \geq 1$, we can return weights $w \in \mathbb{R}_{\geq 0}^n$ satisfying* (4) *with probability $\geq 1 - \delta$ in time*

$$O\left( \mathcal{T}_{\mathrm{mv}}(\{\mathbf{V}_i\}_{i\in[n]}) \cdot (\kappa^\star)^{1.5} \cdot \mathrm{poly}\left( \frac{\log \frac{mnd\kappa^\star}{\delta}}{\epsilon} \right) \right).$$

Here $\mathcal{T}_{\mathrm{mv}}(\{\mathbf{V}_i\}_{i\in[n]})$ denotes the computational complexity of multiplying an arbitrary vector by *all* matrices in $\{\mathbf{V}_i\}_{i\in[n]}$. Notably, in the isotropic case $\mathbf{B} = \mathbf{I}$, Theorem 6 does not require solvers in the sense of (6). We next state our solver which handles the case of general $\mathbf{B}$, under access to (6).

---

[12]We use this notation because, if $\mathcal{T}_{\mathcal{M}}^{\mathrm{sol}}$ is the complexity of solving the system to constant error $c < 1$, then we can use an iterative refinement procedure to solve the system to accuracy $\epsilon$ in time $\mathcal{T}_{\mathcal{M}}^{\mathrm{sol}} \cdot \log \frac{1}{\epsilon}$ for any $\epsilon > 0$.

[13]We did not heavily optimize logarithmic factors and $\epsilon^{-1}$ dependences; for many applications (notably Theorems 1, 2), $\epsilon$ is a constant, so our runtimes are near-linear for a natural representation of the problem under the assumption (5). In several applications (e.g., Theorems 1 and 2) the most important parameter is the "relative condition number" $\kappa^\star$ in the promise (3), so we primarily optimized for the dependence on $\kappa^\star$.

**Theorem 7** (Matrix dictionary recovery, general case). *Given matrices $\{\mathbf{M}_i\}_{i \in [n]}$ with explicit factorizations (5), such that (3) is feasible for some $\kappa^\star \geq 1$ and we can solve in $\mathcal{M}$ to $\epsilon$ relative accuracy in $\mathcal{T}_\mathcal{M}^{sol} \cdot \log \frac{1}{\epsilon}$ time, and $\mathbf{B}$ satisfying*

$$\mathbf{B} \preceq \mathcal{M}(\mathbb{1}) \preceq \alpha \mathbf{B} \text{ and } \mathbf{I} \preceq \mathbf{B} \preceq \beta \mathbf{I}, \tag{7}$$

*we can return weights $w \in \mathbb{R}_{\geq 0}^n$ satisfying (4) with probability $\geq 1 - \delta$ in time*

$$O\left(\mathcal{T}_{tot} \cdot (\kappa^\star)^2 \cdot \text{poly}\left(\frac{\log \frac{mnd\kappa^\star \alpha \beta}{\delta}}{\epsilon}\right)\right), \text{ where } \mathcal{T}_{tot} := \mathcal{T}_{mv}\left(\{\mathbf{V}_i\}_{i \in [n]} \cup \{\mathbf{B}\}\right) + \mathcal{T}_\mathcal{M}^{sol}.$$

The first condition in (7) is no more general than assuming we have a "warm start" reweighting $w_0 \in \mathbb{R}_{\geq 0}^n$ (not necessarily $\mathbb{1}$) satisfying $\mathbf{B} \preceq \sum_{i \in [n]} [w_0]_i \mathbf{M}_i \preceq \alpha \mathbf{B}$, by exploiting scale invariance of the problem and setting $\mathbf{M}_i \leftarrow [w_0]_i \mathbf{M}_i$. The second bound in (7) is equivalent to $\kappa(\mathbf{B}) \leq \beta$ up to constant factors, since given a bound $\beta$, we can use the power method to shift the scale of $\mathbf{B}$ so it is spectrally larger than $\mathbf{I}$. The operation requires just a logarithmic number of matrix vector multiplications with $\mathbf{B}$, which does not impact the runtime in Theorem 7.

**Proof sketches of Theorems 6 and 7.** We defer full proofs of Theorems 6 and 7 to Section 3 of the supplement, but overview our techniques here. Our main workhorse is the following Algorithm 1, which solves a decision variant of the isotropic matrix-dictionary recovery problem (i.e., $\mathbf{B} = \mathbf{I}$), leveraging any subroutine $\mathcal{A}_{pack}$ for solving pure packing instances of (2) from the literature.[14]

We define our approximation notions (used in Lines 6 and 10 of Algorithm 1) in Section 2 of the supplement. In Section 3.1 of the supplement, we analyze correctness of Algorithm 1 using regret bounds for the matrix multiplicative weights framework [ZLO15], which Lines 4-12 are an instance of, and demonstrate tolerance to the stated approximations. Our proof shows that Algorithm 1 meets its output guarantees, which directly implies a solver for the matrix-dictionary recovery problem (3), (4) in the isotropic case $\mathbf{B} = \mathbf{I}$, provided we can efficiently implement the algorithm's steps.[15]

By carefully combining polynomial approximations to the exponential, Johnson-Lindenstrauss sketches, and the power method, we obtain the needed approximations in Lines 6 and 10 of Algorithm 1, which combined with our correctness proof yields Theorem 6. Our proof of Theorem 7 in Section 3.2 of the supplement builds upon Theorem 6 and recursive preconditioning, based on the observation that if we could efficiently apply $\mathbf{B}^{-\frac{1}{2}}$, setting $\mathbf{M}_i \leftarrow \mathbf{B}^{-\frac{1}{2}} \mathbf{M}_i \mathbf{B}^{-\frac{1}{2}}$ reduces to the isotropic case. We show that at a $\sqrt{\kappa^\star}$ overhead, we can use (6) to efficiently simulate $(\mathbf{B} + \lambda \mathbf{I})^{-\frac{1}{2}}$ for $\lambda$ values which are recursively halved, allowing use of Theorem 6 for each recursive call. This general type of adaptive regularization strategy, which we term a homotopy method, is reminiscent of techniques used by other recent works in the literature on numerical linear algebra and structured continuous optimization, such as [LMP13, KLM+14, BCLL18, AKPS19].

**Preconditioning applications.** Formal proofs of our applications in Section 2 are given in Section 5 of the supplement. Theorem 2 follows immediately from Theorem 6 with the dictionary $\mathbf{M}_i = a_i a_i^\top$, where $\{a_i\}_{i \in [n]}$ are rows of $\mathbf{A}$, which satisfies (5) with $m = 1$. Specifically, note for $w \in \mathbb{R}_{\geq 0}^n$,

$$\mathcal{M}(w) = \sum_{i \in [n]} w_i \mathbf{M}_i = \sum_{i \in [n]} w_i a_i a_i^\top = \mathbf{A}^\top \mathbf{W} \mathbf{A}.$$

A result analogous to Theorem 1, but depending quadratically on $\kappa_o^\star(\mathbf{K})$, follows from applying Theorem 7 with $n = d$, $\mathbf{M}_i = e_i e_i^\top$, $\kappa = \kappa_o^\star(\mathbf{K})$, and $\mathbf{B} = \frac{1}{\kappa} \mathbf{K}$ (i.e., using the dictionary of 1-sparse diagonal matrices to approximate $\mathbf{K}$, which satisfies (5) with $m = 1$ and (6) with $\mathcal{T}_\mathcal{M}^{sol} = d$). Compared to inner scaling, this application exchanges the role of the dictionary and the constraint matrix. Theorem 1 goes beyond this black-box use via a homotopy method for simulating access to matrix square roots (to reduce to the isotropic case), yielding an improved $(\kappa_o^\star(\mathbf{K}))^{1.5}$ dependence.

---

[14]For simplicity, we assume each matrix dictionary element's top eigenvalue is in a bounded range. We explicitly bound the cost of achieving this in applications (via rescaling by a constant-factor approximation to the top eigenvalue of each matrix using the power method, see Fact 3 of the supplement), and this does not dominate the runtime. The runtime bottleneck in all our applications is the cost of approximate packing SDP oracles in Line 7; this is an active research area and improvements therein would also reflect in our algorithm's runtime.

[15]We show how to search for $\kappa^\star$ in (3) using an incremental search over a small number of calls to Algorithm 1.

---

**Algorithm 1:** DecideStructuredMPC($\{\mathbf{M}_i\}_{i\in[n]}, \kappa, \mathcal{A}_{\text{pack}}, \delta, \epsilon$)

---

1 **Input:** $\{\mathbf{M}_i\}_{i\in[n]} \in \mathbb{S}_{\succeq \mathbf{0}}^d$ with $\lambda_{\max}(\mathbf{M}_i) \in [1, 2]$ for all $i \in [n]$, $\kappa > 1$, $\delta \in (0, 1)$, $\epsilon \in (0, 1)$,
   $\mathcal{A}_{\text{pack}}$ which on input $v \in \mathbb{R}_{\geq 0}^n$ returns $w \in \mathbb{R}_{\geq 0}^n$ with $\mathcal{M}(w) \preceq \mathbf{I}$ that $\frac{\epsilon}{10}$-multiplicatively
   approximates $\max_{w:\mathcal{M}(w)\preceq\mathbf{I}} v^\top w$ with probability $1 - \frac{\delta}{2T}$, $T = O(\frac{\kappa \log d}{\epsilon^2})$.

2 **Output:** With probability $\geq 1 - \delta$: "yes" or "no" is returned. If there exists $w \in \mathbb{R}_{\geq 0}^n$ with

$$\lambda_{\max}\left(\sum_{i\in[n]} w_i \mathbf{M}_i\right) \leq (1 - \epsilon)\kappa\lambda_{\min}\left(\sum_{i\in[n]} w_i \mathbf{M}_i\right), \tag{8}$$

the algorithm must return "yes." If "yes" is returned, a vector $w$ is given with

$$\lambda_{\max}\left(\sum_{i\in[n]} w_i \mathbf{M}_i\right) \leq (1 + \epsilon)\kappa\lambda_{\min}\left(\sum_{i\in[n]} w_i \mathbf{M}_i\right). \tag{9}$$

3 $\eta \leftarrow \frac{\epsilon}{10\kappa}, T \leftarrow \left\lceil \frac{10 \log d}{\eta\epsilon} \right\rceil, \mathbf{Y}_0 \leftarrow \frac{1}{d}\mathbf{I}, \mathbf{S}_0 \leftarrow \mathbf{0}$

4 **for** $0 \leq t < T$ **do**

5  $\quad \mathbf{Y}_t \leftarrow \frac{\exp(\mathbf{S}_t)}{\operatorname{Tr}\exp(\mathbf{S}_t)}$

6  $\quad v_t \leftarrow$ entrywise $(\frac{\epsilon}{10}, \frac{\epsilon}{10\kappa n})$-approximations to $\{\langle \mathbf{M}_i, \mathbf{Y}_t \rangle\}_{i\in[n]}$, with probability $\geq 1 - \frac{\delta}{4T}$

7  $\quad x_t \leftarrow \mathcal{A}_{\text{pack}}(\kappa v_t)$

8  $\quad$ **if** $\kappa \langle x_t, v_t \rangle < 1 - \frac{\epsilon}{5}$ **then return** "no"

9  $\quad \mathbf{S}_{t+1} \leftarrow \mathbf{S}_t - \eta\kappa\mathcal{M}(x_t)$

10 $\quad \tau \leftarrow \frac{\log d}{\epsilon}$-additive approximation to $\lambda_{\min}(-\mathbf{S}_{t+1})$, with probability $\geq 1 - \frac{\delta}{4T}$

11 $\quad$ **if** $\tau \geq \frac{12 \log d}{\epsilon}$ **then return** ("yes", $\bar{x}$) for $\bar{x} := \frac{1}{t+1}\sum_{0 \leq s \leq t} x_s$

12 **end**

13 **return** ("yes", $\bar{x}$) for $\bar{x} := \frac{1}{T}\sum_{0 \leq t < T} x_t$

---

Our applications in Section 3 are deferred to Section 4 of the supplement. Robust linear system solvers for perturbed variants of structured matrix families follow directly from Theorem 7, taking the matrix dictionary to be a suitable basis for the relevant non-perturbed structured family, which naturally satisfy (6). As an example, to prove Theorem 3, we use Theorem 7 with the dictionary of all $b_e b_e^\top$ for edges $e$ of a complete graph, where $b_e$ is the associated incidence vector; the access (6) is then afforded by known Laplacian solvers. Finally, Theorems 4 and 5 follow by combining Theorems 6, 7 with homotopy techniques, alongside structural facts about M-matrices and Laplacians.

**Further work.** A natural open question is if, e.g., for outer scaling, the $\kappa_o^\star(\mathbf{K})$ dependence in Theorem 1 can be reduced further, ideally to $\sqrt{\kappa_o^\star(\mathbf{K})}$. This would match the most efficient solvers in $\mathbf{K}$ under diagonal rescaling, *if the best known outer scaling was known in advance*. Towards this goal, we prove in Appendix D of the supplement that if a width-independent variant of Theorem 6 is developed, it can achieve such improved runtimes for Theorem 1 (with an analogous improvement for Theorem 2). We also give generalizations of this improvement to finding rescalings which minimize natural average notions of conditioning, under existence of such a conjectured solver.

## 5    Additional related work

**Matrix-dictionary recovery.** Our algorithm for Theorem 6 is based on matrix multiplicative weights [WK06, AK07, AHK12], a popular meta-algorithm for approximately solving SDPs, with carefully chosen gain matrices formed by using packing SDP solvers as a black box. In this sense, it is an efficient reduction from structured SDP instances of the form (3), (4) to pure packing instances.

Similar ideas were previously used in [LS17] (repurposed in [CG18]) for solving graph-structured matrix-dictionary recovery problems. Our Theorems 6 and 7 improve upon these results both in generality (prior works only handled $\mathbf{B} = \mathbf{I}$, and $\kappa^\star = 1 + \epsilon$ for sufficiently small $\epsilon$) and efficiency (our reduction calls a packing solver $\approx \log d$ times for constant $\epsilon, \kappa^\star$, while [LS17] used $\approx \log^2 d$

calls). Perhaps the most direct analog of Theorem 6 is Theorem 3.1 of [CG18], which builds upon the proof of Lemma 3.5 of [LS17] (but lifts the sparsity constraint). The primary qualitative difference with Theorem 6 is that Theorem 3.1 of [CG18] only handles the case where the optimal rescaling $\kappa^\star$ is in $[1, 1.1]$, whereas we handle general $\kappa^\star$. This restriction is important in the proof technique of [CG18], as their approach relies on bounding the change in potential functions based on the matrix exponential of dictionary linear combinations (e.g., the Taylor expansions in their Lemma B.1), which scales poorly with large $\kappa^\star$. Moreover, our method is a natural application of the MMW framework, and is arguably simpler. This simplicity is useful in diagonal scaling applications, as it allows us to obtain a tighter characterization of our $\kappa^\star$ dependence, the primary quantity of interest. Finally, to our knowledge Theorem 7 (which handles general constraint matrices $\mathbf{B}$, crucial for our applications in Theorems 3, 4, and 5) has no analog in prior work, which focused on the isotropic case.

**Semi-random models.** The semi-random noise model we introduce in Section 2 for linear system solving, presented in more detail and formality in Section 6 of the supplement, follows a line of noise models originating in [BS95]. A semi-random model consists of an (unknown) planted instance which a classical algorithm performs well against, augmented by additional information given by a "monotone" or "helpful" adversary masking the planted instance. Conceptually, when an algorithm fails given this "helpful" information, it may have overfit to its generative assumptions. This model has been studied in various statistical settings [Jer92, FK00, FK01, MPW16, MMV12]. Of particular relevance to our work, which studies robustness to semi-random noise in the context of fast algorithms (as opposed to the distinction between polynomial-time algorithms and computational intractability) is [CG18], which developed an algorithm for semi-random matrix completion.

### Acknowledgments

AS was supported in part by a Microsoft Research Faculty Fellowship, NSF CAREER Award CCF-1844855, NSF Grant CCF-1955039, a PayPal research award, and a Sloan Research Fellowship. KS was supported by a Stanford Data Science Scholarship and a Dantzig-Lieberman Operations Research Fellowship. KT was supported by a Google Ph.D. Fellowship, a Simons-Berkeley VMware Research Fellowship, a Microsoft Research Faculty Fellowship, NSF CAREER Award CCF-1844855, NSF Grant CCF-1955039, and a PayPal research award.

We would like to thank Huishuai Zhang for his contributions to an earlier version of this project, Moses Charikar and Yin Tat Lee for helpful conversations, and anonymous reviewers for feedback on earlier variations of this paper.

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
