# Structured Semidefinite Programming
# for Recovering Structured Preconditioners[*]

Arun Jambulapati[†]    Jerry Li[‡]    Christopher Musco[§]    Kirankumar Shiragur[¶]

Aaron Sidford[‖]    Kevin Tian[**]

## Abstract

We develop a general framework for finding approximately-optimal preconditioners for solving linear systems. Leveraging this framework we obtain improved runtimes for fundamental preconditioning and linear system solving problems including the following.

- **Diagonal preconditioning.** We give an algorithm which, given positive definite $\mathbf{K} \in \mathbb{R}^{d \times d}$ with $\mathrm{nnz}(\mathbf{K})$ nonzero entries, computes an $\epsilon$-optimal diagonal preconditioner in time $\widetilde{O}(\mathrm{nnz}(\mathbf{K}) \cdot \mathrm{poly}(\kappa^\star, \epsilon^{-1}))$, where $\kappa^\star$ is the optimal condition number of the rescaled matrix.

- **Structured linear systems.** We give an algorithm which, given $\mathbf{M} \in \mathbb{R}^{d \times d}$ that is either the pseudoinverse of a graph Laplacian matrix or a constant spectral approximation of one, solves linear systems in $\mathbf{M}$ in $\widetilde{O}(d^2)$ time.

Our diagonal preconditioning results improve state-of-the-art runtimes of $\Omega(d^{3.5})$ attained by general-purpose semidefinite programming, and our solvers improve state-of-the-art runtimes of $\Omega(d^\omega)$ where $\omega > 2.3$ is the current matrix multiplication constant. We attain our results via new algorithms for a class of semidefinite programs (SDPs) we call *matrix-dictionary approximation SDPs*, which we leverage to solve an associated problem we call *matrix-dictionary recovery*.

---

[*]This paper is a merge of two unpublished works by subsets of the authors, [JSS18] and [JLM+21], available on arXiv, and is intended to replace them.

[†]Simons Institute, `jmblpati@berkeley.edu`. Work completed at Stanford and the University of Washington.

[‡]Microsoft Research, `jerrl@microsoft.com`.

[§]New York University, `cmusco@nyu.edu`.

[¶]Broad Institute of MIT and Harvard, `shiragur@stanford.edu`. Work completed at Stanford.

[‖]Stanford University, `sidford@stanford.edu`.

[**]University of Texas at Austin, `kjtian@cs.utexas.edu`. Work completed at Stanford and Microsoft Research.

# Contents

# 1 Introduction

Preconditioning is a fundamental primitive in the theory and practice of numerical linear algebra, optimization, and data science. Broadly, its goal is to improve conditioning properties (e.g., the range of eigenvalues) of a matrix $\mathbf{M}$ by finding another matrix $\mathbf{N}$ which approximates the inverse of $\mathbf{M}$ and is more efficient to construct and apply than computing $\mathbf{M}^{-1}$. This strategy underpins a variety of popular, recently-developed tools, such as adaptive gradient methods for machine learning (e.g., Adagrad and Adam [DHS11, KB15]), and near-linear time solvers for combinatorially-structured matrices (e.g. graph Laplacians [ST04]). Despite widespread practical adoption of such techniques, there is a surprising lack of provably efficient algorithms for preconditioning.

Our work introduces a new tool, *matrix-dictionary recovery*, and leverages it to obtain the first near-linear time algorithms for several structured preconditioning problems in well-studied applications. Informally, the problem we study is as follows (see Section 1.3 for the formal definition).

> *Given a matrix $\mathbf{M}$ and a "matrix-dictionary" $\{\mathbf{M}_i\}$, find the best preconditioner*
> $$\mathbf{N} = \sum_i w_i \mathbf{M}_i \text{ of } \mathbf{M} \text{ expressible as a nonnegative linear combination of } \{\mathbf{M}_i\}. \tag{1}$$

We develop general-purpose solvers for the problem (1). We further apply these solvers to obtain state-of-the-art algorithms for fundamental tasks such as preconditioning linear systems and regression, and approximately recovering structured matrices, including the following results.

- **Diagonal preconditioning.** We consider the classical numerical linear algebra problem of *diagonal preconditioning* [vdS69]. Given $\mathbf{K} \in \mathbb{S}_{\succ \mathbf{0}}^d$, the goal is to find a diagonal $\mathbf{W} \in \mathbb{S}_{\succ \mathbf{0}}^d$ minimizing the condition number of $\mathbf{W}^{\frac{1}{2}} \mathbf{K} \mathbf{W}^{\frac{1}{2}}$. Theorem 1 obtains the first near-linear time algorithms for this problem when the optimal condition number of the rescaled matrix is small.

- **Semi-random regression.** We consider a related problem, motivated by semi-random noise models, which takes full-rank $\mathbf{A} \in \mathbb{R}^{n \times d}$ with $n \geq d$ and seeks $\mathbf{W} \in \mathbb{S}_{\succ \mathbf{0}}^n$ minimizing the condition number of $\mathbf{A}^\top \mathbf{W} \mathbf{A}$. Theorem 2 gives the first near-linear time algorithm for this problem, and applications of it reduce risk bounds for statistical linear regression.

- **Structured linear systems.** We robustify Laplacian system solvers, e.g., [ST04], to obtain near-linear time solvers for systems in dense matrices well-approximated spectrally by Laplacians in Theorem 3. We also give new near-linear time solvers for several families of structured matrices, e.g., dense inverse Laplacians and M-matrices,[1] in Theorems 4 and 5.

For the preconditioning problems considered in Theorems 1, 2, and 3, we give the first runtimes faster than a generic SDP solver, for which state-of-the-art runtimes [JKL+20, HJS+22] are highly superlinear ($\Omega(d^{3.5})$ for diagonal preconditioning and $\Omega(d^{2\omega})$ for approximating Laplacians, where $d$ is the matrix dimension and $\omega > 2.3$ is the current matrix multiplication constant [AW21]). For the corresponding linear system solving problems in each case, as well as in Theorems 4 and 5, the prior state-of-the-art was to treat the linear system as generic and ran in $\Omega(d^\omega)$ time.

We survey these results in Section 1.1 and 1.2, highlighting how the problems they study can be viewed as instances of (1). We then discuss the general matrix-dictionary recovery problem we study in more detail, and state the guarantees of our solvers, in Section 1.3. Finally, we compare our framework to related algorithms and give a more thorough runtime comparison in Section 1.4.

---

[1] Inverse M-matrices are necessarily dense, see Lemma 33 in Appendix B.

## 1.1 Diagonal preconditioning

When solving linear systems via iterative methods, one of the most popular preconditioning strategies is to use a diagonal matrix. This strategy is appealing because diagonal matrices can be applied and inverted quickly. Determining the best diagonal preconditioner is a classical numerical linear algebra problem studied since the 1950s [FS55, vdS69, PG90]. In the context of (1), a diagonal preconditioner is a nonnegative linear combination of the matrix-dictionary consisting of

$$e_i e_i^\top \text{ where } e_i \text{ is the } i^{\text{th}} \text{ basis vector.} \tag{2}$$

Leveraging this viewpoint, we study two natural instantiations of diagonal preconditioning.

**Outer scaling.** One formulation of the optimal diagonal preconditioning problem, which we refer to as *outer scaling*, asks to optimally reduce the condition number of positive definite $\mathbf{K} \in \mathbb{R}^{d \times d}$ with a diagonal matrix $\mathbf{W}$, i.e., return diagonal $\mathbf{W} = \mathbf{diag}(w)$ for $w \in \mathbb{R}_{>0}^d$ such that[2]

$$\kappa(\mathbf{W}^{\frac{1}{2}} \mathbf{K} \mathbf{W}^{\frac{1}{2}}) \approx \kappa_o^\star(\mathbf{K}) := \min_{\text{diagonal } \mathbf{W} \succ \mathbf{0}} \kappa(\mathbf{W}^{\frac{1}{2}} \mathbf{K} \mathbf{W}^{\frac{1}{2}}).$$

Given $\mathbf{W}$, a solution to $\mathbf{K}x = b$ can be obtained by solving the better-conditioned $\mathbf{W}^{\frac{1}{2}} \mathbf{K} \mathbf{W}^{\frac{1}{2}} y = \mathbf{W}^{\frac{1}{2}} b$ and returning $x = \mathbf{W}^{\frac{1}{2}} y$. The optimal $\mathbf{W}$ can be obtained via a semidefinite program (SDP) [QYZ20], but the computational cost of general-purpose SDP solvers outweighs benefits for solving linear systems. Outer scaling is poorly understood algorithmically; prior to our work, even attaining a constant-factor approximation to $\kappa_o^\star(\mathbf{K})$ without a generic SDP solver was unknown.

This state of affairs has resulted in the widespread use of studying heuristics for constructing $\mathbf{W}$, such as *Jacobi preconditioning* [vdS69, GR89] and *matrix scaling* [AZLOW17, CMTV17a, GO18]. The former was notably highlighted by Adagrad [DHS11], which used Jacobi preconditioning to improve the computational costs of their method. However, both heuristcs have clear drawbacks both from a theoretical and practical perspective. Prior to our work the best approximation guarantee known for Jacobi preconditioning was a result of van der Sluis [vdS69, GR89], which shows the Jacobi preconditioner is an $m$-factor approximation to the optimal preconditioning problem where $m \leq d$ is the maximum number of non-zeros in any row of $\mathbf{K}$: in dense matrices this scales linearly in the problem dimension and can be much larger than $\kappa_o^\star(\mathbf{K})$. We review and slightly strengthen this result in Appendix C. We also prove a new *dimension-independent* baseline result of independent interest: the Jacobi preconditioner always obtains condition number no worse than $(\kappa_o^\star(\mathbf{K}))^2$. Unfortunately, we exhibit a simple family of matrices showing this characterization of the Jacobi preconditioner quality is tight, dashing hopes they can solve the outer scaling problem near-optimally. On the other hand, while it is

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

[5]Our informal results suppress precise dependences on approximation and log factors as the only informality.

$\kappa(\mathbf{A}^\top \mathbf{A}) \gg \kappa(\mathbf{A}_g^\top \mathbf{A}_g)$. This can occur when rows are sampled from heterogeneous sources. Perhaps counterintuitively, by giving additional consistent data, the adversary can arbitrarily hinder the cost of iterative methods. This failure can be interpreted as being due to overfitting to generative assumptions (e.g., sampling rows from a well-conditioned covariance, instead of a mixture): standard iterative methods assume too much structure, where ideally they would use as little as information-theoretically possible.

Our inner scaling methods can be viewed as "robustifying" linear system solving to such semi-random noise models (by finding $\mathbf{W}$ yielding a rescaled condition number comparable or better than the indicator of the rows of $\mathbf{A}_g$, which are not known a priori). In Section 6, we demonstrate applications of inner scaling in reducing the mean-squared error risk in statistical regression settings encompassing our semi-random noise model, where the observations $b$ are corrupted by (homoskedastic or heteroskedastic) noise. In all settings, our preconditioning algorithms yield computational gains, improved risk bounds, or both, by factors of roughly $\kappa(\mathbf{A}^\top \mathbf{A})/\kappa_i^\star(\mathbf{A})$.

## 1.2 Robust linear algebra for structured matrices

Over the past decade, the theoretical computer science and numerical linear algebra communities have dedicated substantial effort to developing solvers for regression problems in various families of combinatorially-structured matrices. Perhaps the most prominent example is [ST04], which gave a near-linear time solver for linear systems in graph Laplacian matrices.[6] A long line of exciting work has obtained improved solvers for these systems [KMP10, KMP11, KOSZ13, LS13, CKM+14, PS14, KLP+16, KS16, JS21], which have been used to improve the runtimes for a wide variety of graph-structured problems, including maximum flow [CKM+11, Mad13, LS14], sampling random spanning trees [KM09, MST15, DKP+17, Sch18], graph clustering [ST04, OV11, OSV12], and more [DS08, KRSS15, CMSV17, CMTV17b]. Additionally, efficient linear system solvers have been developed for solving systems in other types of structured matrices, e.g., block diagonally dominant systems [KLP+16], M-matrices [AJSS19], and directed Laplacians [CKP+16, CKP+17, CKK+18].

**Perturbations of structured matrices.** Despite the importance of these families of matrices with combinatorial structure, the solvers developed in prior work are in some ways quite brittle. In particular, there are several simple classes of matrices closely related to Laplacians for which the best-known runtimes for solving linear systems are achieved by ignoring the structure of the problem, and using generic matrix multiplication techniques as a black box. Perhaps the simplest example is solving systems in *perturbed Laplacians*, i.e., matrices which admit constant-factor approximations by a Laplacian matrix, but which are not Laplacians themselves. This situation can arise when a Laplacian is used to approximate a physical phenomenon [BHV08]. As an illustration of the techniques we develop, we give the following perturbed Laplacian solver.

**Theorem 3** (Perturbed Laplacian solver, informal, see Theorem 10). *Let $\mathbf{M} \succeq \mathbf{0} \in \mathbb{R}^{n \times n}$ be such that there exists an (unknown) Laplacian $\mathbf{L}$ with $\mathbf{M} \preceq \mathbf{L} \preceq \kappa^\star \mathbf{M}$, and that $\mathbf{L}$ corresponds to a graph with edge weights between $w_{\min}$ and $w_{\max}$, with $\frac{w_{\max}}{w_{\min}} \leq U$. For any $\delta \in (0, 1)$ and $\epsilon > 0$, there is an algorithm recovering a Laplacian $\mathbf{L}'$ with $\mathbf{M} \preceq \mathbf{L}' \preceq (1 + \epsilon)\kappa^\star \mathbf{M}$ with probability $\geq 1 - \delta$ in time*

$$O\left(n^2 \cdot (\kappa^\star)^2 \cdot \mathrm{poly}\left(\frac{\log \frac{n\kappa^\star U}{\delta}}{\epsilon}\right)\right).$$

*Consequently, there is an algorithm for solving linear systems in $\mathbf{M}$ to $\epsilon$-relative accuracy with probability $\geq 1 - \delta$, in time $O(n^2 \cdot (\kappa^\star)^2 \cdot \mathrm{polylog}\left(\frac{n\kappa^\star U}{\delta\epsilon}\right)).$[7]*

---

[6]We formally define the structured families of matrices we study in Section 4.

[7]See (8) and the following discussion for the definition of solving to relative accuracy.

Theorem 3 can be viewed as solving a preconditioner construction problem, where we know there exists a Laplacian matrix $\mathbf{L}$ which spectrally resembles $\mathbf{M}$, and wish to efficiently recover a Laplacian with similar guarantees. Our matrix-dictionary recovery framework captures the setting of Theorem 3 by leveraging the matrix-dictionary consisting of the $O(n^2)$ matrices

$$b_e b_e^\top \in \mathbb{R}^{n \times n},$$

where $b_e$ is the 2-sparse signed vector corresponding to an edge in an $n$-vertex graph (see Section 2). The conceptual message of Theorem 3 is that near-linear time solvers for Laplacians robustly extend through our preconditioning framework to efficiently solve matrices approximated by Laplacians. Beyond this specific application, our framework could be used to solve perturbed generalizations of future families of structured matrices.

**Recovery of structured matrices.** In addition to directly spectrally approximating and solving in matrices which are well-approximated by preconditioners with diagonal or combinatorial structure, our framework also yields solvers for new families of matrices. We show that our preconditioning techniques can be used in conjunction with properties of graph-structured matrices to provide solvers and spectral approximations for *inverse M-matrices* and *Laplacian pseudoinverses*. Recovering Laplacians from their pseudoinverses and solving linear systems in the Laplacian pseudoinverse arise when trying to fit a graph to data or recover a graph from effective resistances, a natural distance measure (see [HMMT18] for motivation and discussion of related problems). More broadly, the problem of solving linear systems in inverse symmetric M-matrices is prevalent and corresponds to statistical inference problems involving distributions that are multivariate totally positive of order 2 (MTP$_2$) [KR83, SH14, FLS$^+$17]. Our main results are the following.

**Theorem 4** (M-matrix recovery and inverse M-matrix solver, informal, see Theorem 11). *Let $\mathbf{M}$ be the inverse of an unknown invertible symmetric M-matrix, let $\kappa$ be an upper bound on its condition number, and let $U$ be the ratio of the largest to smallest entries of $\mathbf{M}\mathbb{1}$.*[8] *For any $\delta \in (0,1)$ and $\epsilon > 0$, there is an algorithm recovering a $(1+\epsilon)$-spectral approximation to $\mathbf{M}^{-1}$ in time*

$$O\left(n^2 \cdot \mathrm{poly}\left(\frac{\log \frac{n\kappa U}{\delta}}{\epsilon}\right)\right).$$

*Consequently, there is an algorithm for solving linear systems in $\mathbf{M}$ to $\epsilon$-relative accuracy with probability $\geq 1 - \delta$, in time $O(n^2 \cdot \mathrm{polylog}\left(\frac{n\kappa U}{\delta\epsilon}\right))$.*

**Theorem 5** (Laplacian recovery and Laplacian pseudoinverse solver, informal, see Theorem 12). *Let $\mathbf{M}$ be the pseudoinverse of unknown Laplacian $\mathbf{L}$, and that $\mathbf{L}$ corresponds to a graph with edge weights between $w_{\min}$ and $w_{\max}$, with $\frac{w_{\max}}{w_{\min}} \leq U$. For any $\delta \in (0,1)$ and $\epsilon > 0$, there is an algorithm recovering a Laplacian $\mathbf{L}'$ with $\mathbf{M}^\dagger \preceq \mathbf{L}' \preceq (1+\epsilon)\mathbf{M}^\dagger$ in time*

$$O\left(n^2 \cdot \mathrm{poly}\left(\frac{\log \frac{nU}{\delta}}{\epsilon}\right)\right).$$

*Consequently, there is an algorithm for solving linear systems in $\mathbf{M}$ to $\epsilon$-relative accuracy with probability $\geq 1 - \delta$, in time $O(n^2 \cdot \mathrm{polylog}\left(\frac{nU}{\delta\epsilon}\right))$.*

Theorems 4 and 5 are perhaps a surprising demonstration of the utility of our techniques: just because a matrix family is well-approximated by structured preconditioners, it is not a priori clear

---

[8]By Lemma 33 in Appendix B, the vector $\mathbf{M}\mathbb{1}$ is entrywise positive.

that their inverses also are. However, we show that by applying recursive preconditioning tools in conjunction with our recovery methods, we can obtain analogous results for these inverse families. These results add to the extensive list of combinatorially-structured matrix families admitting efficient linear algebra primitives. We view our approach as a proof-of-concept of further implications in designing near-linear time system solvers for structured families via algorithms for (1).

## 1.3 Our framework: matrix-dictionary recovery

Our general strategy for matrix-dictionary recovery, i.e., recovering preconditioners in the sense of (1), is via applications of a new custom approximate solver we develop for a family of structured SDPs. SDPs are fundamental optimization problems that have been the source of extensive study for decades [VB96], with numerous applications across operations research and theoretical computer science [GW95], statistical modeling [WSV00, GM12], and machine learning [RSL18]. Though there have been recent advances in solving general SDPs (e.g., [JKL+20, HJS+22] and references therein), the current state-of-the-art solvers have superlinear runtimes, prohibitive in large-scale applications. Consequently, there has been extensive research on designing faster approximate SDP solvers under different assumptions [KV05, WK06, AK07, BBN13, GHM15, AL17, CDST19].

We now provide context for our solver for structured "matrix-dictionary approximation" SDPs, and state our main results on solving them.

**Positive SDPs.** One prominent class of structured SDPs are what we refer to as *positive SDPs*, namely SDPs in which the cost and constraint matrices are all positive semidefinite (PSD), a type of structure present in many important applications [GW95, ARV09, JJUW11, LS17, CG18, CDG19, CFB19, CMY20]. This problem generalizes positive linear programming (also referred to as "mixed packing-covering linear programs"), a well-studied problem over the past several decades [LN93, PST95, You01, MRWZ16b, AO19]. It was recently shown that a prominent special case of positive SDPs known as *packing SDPs* can be solved in nearly-linear time [ALO16, PTZ16], a fact that has had numerous applications in robust learning and estimation [CG18, CDG19, CFB19, CMY20] as well as in combinatorial optimization [LS17]. However, extending known packing SDP solvers to broader classes of positive SDPs, e.g., covering or mixed packing-covering SDPs has been elusive [JY12, JLL+20], and is a key open problem in the algorithmic theory of structured optimization.[9] We use the term positive SDP in this paper to refer to the fully general mixed packing-covering SDP problem, parameterized by "packing" and "covering" matrices $\{\mathbf{P}_i\}_{i \in [n]}, \mathbf{P}, \{\mathbf{C}_i\}_{i \in [n]}, \mathbf{C} \in \mathbb{S}_{\succeq \mathbf{0}}^d$, and asks to find[10] the smallest $\mu > 0$ such that there exists $w \in \mathbb{R}_{\geq 0}^n$ with

$$\sum_{i \in [n]} w_i \mathbf{P}_i \preceq \mu \mathbf{P}, \ \sum_{i \in [n]} w_i \mathbf{C}_i \succeq \mathbf{C}. \tag{3}$$

By redefining $\mathbf{P}_i \leftarrow \frac{1}{\mu} \mathbf{P}^{-\frac{1}{2}} \mathbf{P}_i \mathbf{P}^{-\frac{1}{2}}$ and $\mathbf{C}_i \leftarrow \mathbf{C}^{-\frac{1}{2}} \mathbf{C}_i \mathbf{C}^{-\frac{1}{2}}$ for all $i \in [n]$, the optimization problem in (3) is equivalent to testing whether there exists $w \in \mathbb{R}_{\geq 0}^n$ such that

$$\sum_{i \in [n]} w_i \mathbf{P}_i \preceq \sum_{i \in [n]} w_i \mathbf{C}_i. \tag{4}$$

The formulation (4) was studied by [JY12, JLL+20], and an important open problem in structured convex programming is designing a "width-independent" solver for testing the feasibility of (4) up to a $1 + \epsilon$ factor (i.e., testing whether (4) is approximately feasible with an iteration count polynomial

---

[9]A faster solver for general positive (mixed packing-covering) SDPs was claimed in [JLL+20], but an error was later discovered in that work, as is recorded in the most recent arXiv version [JLL+21].

[10]This is the optimization variant; the corresponding decision variant asks to test if for a given $\mu$, (3) is feasible.

in $\epsilon^{-1}$ and polylogarithmic in other problem parameters), or solving for the optimal $\mu$ in (3) to this approximation factor. Up to now, such width-independent solvers have remained elusive beyond pure packing SDPs [ALO16, PTZ16, JLT20], even for basic extensions such as pure covering.

**Matrix-dictionary approximation SDPs.** We develop an efficient solver for specializations of (3) and (4) where the packing and covering matrices $\{\mathbf{P}_i\}_{i \in [n]}, \{\mathbf{C}_i\}_{i \in [n]}$, as well as the constraints $\mathbf{P}$, $\mathbf{C}$, are multiples of each other. As we will see, this structured family of SDPs, which we call *matrix-dictionary approximation SDPs*, is highly effective for capturing the forms of approximation required by preconditioning problems. Many of our preconditioning results follow as careful applications of the matrix-dictionary approximation SDP solvers we give in Theorem 6 and Theorem 7.

We develop efficient algorithms for the following special case of (3), (4), the main meta-problem we study. Given a set of matrices (a "matrix-dictionary") $\{\mathbf{M}_i\}_{i \in [n]} \in \mathbb{S}^d_{\succeq \mathbf{0}}$, a constraint matrix $\mathbf{B}$, and a tolerance parameter $\epsilon \in (0,1)$, such that there exists a feasible set of weights $w^\star \in \mathbb{R}^n_{\geq 0}$ with

$$\mathbf{B} \preceq \sum_{i \in [n]} w_i^\star \mathbf{M}_i \preceq \kappa^\star \mathbf{B}, \tag{5}$$

for some unknown $\kappa^\star \geq 1$, we wish to return a set of weights $w \in \mathbb{R}^n_{\geq 0}$ such that

$$\mathbf{B} \preceq \sum_{i \in [n]} w_i \mathbf{M}_i \preceq (1 + \epsilon)\kappa^\star \mathbf{B}. \tag{6}$$

While this "matrix-dictionary recovery" problem is a restricted case of (3), as we demonstrate, it is already expressive enough to capture many interesting applications.

Our results concerning the problem (5) and (6) assume that the matrix-dictionary $\{\mathbf{M}_i\}_{i \in [n]}$ is "simple" in two respects. First, we assume that we have an explicit factorization of each $\mathbf{M}_i$ as

$$\mathbf{M}_i = \mathbf{V}_i \mathbf{V}_i^\top, \ \mathbf{V}_i \in \mathbb{R}^{d \times m}. \tag{7}$$

Our applications in Sections 1.1 and 1.2 satisfy this assumption with $m = 1$. Second, denoting $\mathcal{M}(w) := \sum_{i \in [n]} w_i \mathbf{M}_i$, we assume that we can approximately solve systems in $\mathcal{M}(w) + \lambda \mathbf{I}$ for any $w \in \mathbb{R}^n_{\geq 0}$ and $\lambda \geq 0$. Concretely, for any $\epsilon > 0$, we assume there is a linear operator $\widetilde{\mathcal{M}}_{w,\lambda,\epsilon}$ which we can compute and apply in $\mathcal{T}^{\mathrm{sol}}_{\mathcal{M}} \cdot \log \frac{1}{\epsilon}$ time,[11] and that $\widetilde{\mathcal{M}}_{w,\lambda,\epsilon} \approx (\mathcal{M}(w) + \lambda \mathbf{I})^{-1}$ in that:

$$\left\| \widetilde{\mathcal{M}}_{w,\lambda,\epsilon} v - (\mathcal{M}(w) + \lambda \mathbf{I})^{-1} v \right\|_2 \leq \epsilon \left\| (\mathcal{M}(w) + \lambda \mathbf{I})^{-1} v \right\|_2 \text{ for all } v \in \mathbb{R}^d. \tag{8}$$

In this case, we say "we can solve in $\mathcal{M}$ to $\epsilon$-relative accuracy in $\mathcal{T}^{\mathrm{sol}}_{\mathcal{M}} \cdot \log \frac{1}{\epsilon}$ time." If $\mathcal{M}$ is a single matrix $\mathbf{M}$, we say "we can solve in $\mathbf{M}$ to $\epsilon$-relative accuracy in $\mathcal{T}^{\mathrm{sol}}_{\mathbf{M}} \cdot \log \frac{1}{\epsilon}$ time." Notably, for the matrix-dictionaries in our applications, e.g., diagonal 1-sparse matrices or edge Laplacians, such access to $\{\mathbf{M}_i\}_{i \in [n]}$ exists so we obtain end-to-end efficient algorithms. Ideally (for near-linear time algorithms), $\mathcal{T}^{\mathrm{sol}}_{\mathbf{M}}$ is roughly the total sparsity of $\{\mathbf{M}_i\}_{i \in [n]}$, which holds in all our applications.

Under these assumptions, we prove the following main claims in Section 3. We did not heavily optimize logarithmic factors and the dependence on $\epsilon^{-1}$, as many of these factors are inherited from subroutines in prior work. For many applications, it suffices to set $\epsilon$ to be a sufficiently small constant, so our runtimes are nearly-linear for a natural representation of the problem under access to efficient solvers for the dictionary $\mathcal{M}$. In several applications (e.g., Theorems 1 and 2) the most

---

[11]We use this notation because, if $\mathcal{T}^{\mathrm{sol}}_{\mathcal{M}}$ is the complexity of solving the system to constant error $c < 1$, then we can use an iterative refinement procedure to solve the system to accuracy $\epsilon$ in time $\mathcal{T}^{\mathrm{sol}}_{\mathcal{M}} \cdot \log \frac{1}{\epsilon}$ for any $\epsilon > 0$.

important parameter is the "relative condition number" $\kappa$, so we primarily optimized for $\kappa$.

**Theorem 6** (Matrix dictionary recovery, isotropic case, informal, see Theorem 8)**.** *Given matrices* $\{\mathbf{M}_i\}_{i\in[n]}$ *with explicit factorizations* (7)*, such that* (5) *is feasible for* $\mathbf{B} = \mathbf{I}$ *and some* $\kappa^\star \geq 1$*, we can return weights* $w \in \mathbb{R}^n_{\geq 0}$ *satisfying* (6) *with probability* $\geq 1 - \delta$ *in time*

$$O\left(\mathcal{T}_{\mathrm{mv}}(\{\mathbf{V}_i\}_{i\in[n]}) \cdot (\kappa^\star)^{1.5} \cdot \mathrm{poly}\left(\frac{\log \frac{mnd\kappa^\star}{\delta}}{\epsilon}\right)\right).$$

Here $\mathcal{T}_{\mathrm{mv}}(\{\mathbf{V}_i\}_{i\in[n]})$ denotes the computational complexity of multiplying an arbitrary vector by *all* matrices in $\{\mathbf{V}_i\}_{i\in[n]}$. As an example of the utility of Theorem 6, letting the rows of $\mathbf{A} \in \mathbb{R}^{n\times d}$ be denoted $\{a_i\}_{i\in[n]}$, a direct application with $\mathbf{V}_i \leftarrow a_i$, $\mathbf{M}_i \leftarrow a_i a_i^\top$ results in Theorem 2, our result on inner scaling diagonal preconditioners. We next handle the case of general $\mathbf{B}$.

**Theorem 7** (Matrix dictionary recovery, general case, informal, see Theorem 9)**.** *Given matrices* $\{\mathbf{M}_i\}_{i\in[n]}$ *with explicit factorizations* (7)*, such that* (5) *is feasible for some* $\kappa^\star \geq 1$ *and we can solve in* $\mathcal{M}$ *to* $\epsilon$ *relative accuracy in* $\mathcal{T}^{\mathrm{sol}}_{\mathcal{M}} \cdot \log \frac{1}{\epsilon}$ *time, and* $\mathbf{B}$ *satisfying*

$$\mathbf{B} \preceq \mathcal{M}(\mathbb{1}) \preceq \alpha\mathbf{B} \text{ and } \mathbf{I} \preceq \mathbf{B} \preceq \beta\mathbf{I}, \tag{9}$$

*we can return weights* $w \in \mathbb{R}^n_{\geq 0}$ *satisfying* (6) *with probability* $\geq 1 - \delta$ *in time*

$$O\left(\left(\mathcal{T}_{\mathrm{mv}}\left(\{\mathbf{V}_i\}_{i\in[n]} \cup \{\mathbf{B}\}\right) + \mathcal{T}^{\mathrm{sol}}_{\mathcal{M}}\right) \cdot (\kappa^\star)^2 \cdot \mathrm{poly}\left(\frac{\log \frac{mnd\kappa^\star\alpha\beta}{\delta}}{\epsilon}\right)\right).$$

The first condition in (9) is no more general than assuming we have a "warm start" reweighting $w_0 \in \mathbb{R}^n_{\geq 0}$ (not necessarily $\mathbb{1}$) satisfying $\mathbf{B} \preceq \sum_{i\in[n]}[w_0]_i \mathbf{M}_i \preceq \alpha\mathbf{B}$, by exploiting scale invariance of the problem and setting $\mathbf{M}_i \leftarrow [w_0]_i \mathbf{M}_i$. The second bound in (9) is equivalent to $\kappa(\mathbf{B}) \leq \beta$ (see Section 2) up to constant factors, since given a bound $\beta$, we can use the power method (cf. Fact 3) to shift the scale of $\mathbf{B}$ so it is spectrally larger than $\mathbf{I}$ (i.e., estimating the largest eigenvalue and shifting it to be $\Omega(\beta)$). The operation requires just a logarithmic number of matrix vector multiplications with $\mathbf{B}$, which does not impact the runtime in Theorem 7.

Several of our preconditioning results go beyond black-box applications of Theorems 6 and 7. For example, a result analogous to Theorem 1 but depending quadratically on $\kappa^\star_o(\mathbf{K}))$ can be obtained by directly applying Theorem 7 with $n = d$, $\mathbf{M}_i = e_i e_i^\top$, $\kappa = \kappa^\star_o(\mathbf{K})$, and $\mathbf{B} = \frac{1}{\kappa}\mathbf{K}$ (i.e., using the dictionary of 1-sparse diagonal matrices to approximate $\mathbf{K}$). We obtain an improved $(\kappa^\star_o(\mathbf{K}))^{1.5}$ dependence via another homotopy method (similar to the one used for our SDP solver in Theorem 7), which allows us to efficiently compute matrix-vector products with a symmetric square-root of $\mathbf{K}$. Access to the square root allows us to reduce the iteration complexity of our SDP solver.

**Further work.** A natural open question is if, e.g., for outer scaling, the $\kappa^\star_o(\mathbf{K})$ dependence in Theorem 1 can be reduced further, ideally to $\sqrt{\kappa^\star_o(\mathbf{K})}$. This would match the most efficient solvers in $\mathbf{K}$ under diagonal rescaling, *if the best known outer scaling was known in advance*. Towards this goal, we prove in Appendix D that if a width-independent variant of Theorem 6 is developed, it can achieve such improved runtimes for Theorem 1 (with an analogous improvement for Theorem 2). We also give generalizations of this improvement to finding rescalings which minimize natural average notions of conditioning, under existence of such a conjectured solver.

## 1.4 Comparison to prior work

**Runtime implications.** For all the problems we study (enumerated in Sections 1.1 and 1.2), our methods are (to our knowledge) the first in the literature to run in nearly-linear time in the sparsities of the constraint matrices, with polynomial dependence on the *optimal* conditioning.

For example, consider our results (Theorems 1 and 2) on computing diagonal preconditioners. Beyond that which is obtainable by black-box using general SDP solvers, we are not aware of any other claimed runtime in the literature. Directly using state-of-the-art SDP solvers [JKL+20, HJS+22] incurs substantial overhead $\Omega(n^\omega \sqrt{d} + nd^{2.5})$ or $\Omega(n^\omega + d^{4.5} + n^2\sqrt{d})$, where $\omega < 2.372$ is the current matrix multiplication constant [Wil12, Gal14, AW21, DWZ23, WXXZ23]. For outer scaling, where $n = d$, this implies an $\Omega(d^{3.5})$ runtime; for other applications, e.g., preconditioning $d \times d$ perturbed Laplacians where $n = d^2$, the runtime is $\Omega(d^{2\omega})$. Applying state-of-the-art approximate SDP solvers (rather than our custom ones, i.e., Theorems 6 and 7) appears to yield runtimes $\Omega(\mathrm{nnz}(\mathbf{A}) \cdot d^{2.5})$, as described in Appendix E.2 of [LSTZ20]. This is in contrast with our Theorems 1, 2 which achieve $\widetilde{O}\left(\mathrm{nnz}(\mathbf{A}) \cdot (\kappa^\star)^{1.5}\right)$. Hence, we improve existing tools by $\mathrm{poly}(d)$ factors in the main regime of interest where the optimal rescaled condition number $\kappa^\star$ is small. Concurrent to our work, [QGH+22] gave algorithms for constructing optimal diagonal preconditioners using interior point methods for SDPs, which run in at least the superlinear times discussed previously.

Similar speedups hold for our results on solving matrix-dictionary recovery for graph-structured matrices (Theorems 3, 4, and 5). Further, for key matrices in each of these cases (e.g., constant-factor spectral approximations of Laplacians, inverse M-matrices, and Laplacian pseudoinverses) we obtain $\widetilde{O}(n^2)$ time algorithms for solving linear systems in these matrices to inverse polynomial accuracy. This runtime is near-linear when the input is dense and in each case when the input is dense the state-of-the-art prior methods were to run general linear system solvers using $O(n^\omega)$ time.

**Matrix-dictionary recovery.** Our algorithm for Theorem 6 is based on matrix multiplicative weights [WK06, AK07, AHK12], a popular meta-algorithm for approximately solving SDPs, with carefully chosen gain matrices formed by using packing SDP solvers as a black box. In this sense, it is an efficient reduction from structured SDP instances of the form (5), (6) to pure packing instances.

Similar ideas were previously used in [LS17] (repurposed in [CG18]) for solving graph-structured matrix-dictionary recovery problems. Our Theorems 6 and 7 improve upon these results both in generality (prior works only handled $\mathbf{B} = \mathbf{I}$, and $\kappa^\star = 1+\epsilon$ for sufficiently small $\epsilon$) and efficiency (our reduction calls a packing solver $\approx \log d$ times for constant $\epsilon, \kappa^\star$, while [LS17] used $\approx \log^2 d$ calls). Perhaps the most direct analog of Theorem 6 is Theorem 3.1 of [CG18], which builds upon the proof of Lemma 3.5 of [LS17] (but lifts the sparsity constraint). The primary qualitative difference with Theorem 6 is that Theorem 3.1 of [CG18] only handles the case where the optimal rescaling $\kappa^\star$ is in $[1, 1.1]$, whereas we handle general $\kappa^\star$. This restriction is important in the proof technique of [CG18], as their approach relies on bounding the change in potential functions based on the matrix exponential of dictionary linear combinations (e.g., the Taylor expansions in their Lemma B.1), which scales poorly with large $\kappa^\star$. Moreover, our method is a natural application of the MMW framework, and is arguably simpler. This simplicity is useful in diagonal scaling applications, as it allows us to obtain a tighter characterization of our $\kappa^\star$ dependence, the primary quantity of interest.

Finally, to our knowledge Theorem 7 (which handles general constraint matrices $\mathbf{B}$, crucial for our applications in Theorems 3, 4, and 5) has no analog in prior work, which focused on the isotropic case. The algorithm we develop to prove Theorem 7 is based on combining Theorem 6 with a multi-level iterative preconditioning scheme we refer to as a homotopy method. In particular, our algorithm for Theorem 6 recursively calls Theorem 6 and preconditioned linear system solvers as black boxes, to provide near-optimal reweightings $\mathcal{M}(w)$ which approximate $\mathbf{B} + \lambda\mathbf{I}$ for various values of $\lambda$. We then combine our access to linear system solvers in $\mathcal{M}(w)$ with efficient rational

approximations to various matrix functions, yielding our overall algorithm. This homotopy method framework is reminiscent of techniques used by other recent works in the literature on numerical linear algebra and structured continuous optimization, such as [LMP13, KLM$^+$14, BCLL18, AKPS19].

**Semi-random models.** The semi-random noise model we introduce in Section 1.1 for linear system solving, presented in more detail and formality in Section 6, follows a line of noise models originating in [BS95]. A semi-random model consists of an (unknown) planted instance which a classical algorithm performs well against, augmented by additional information given by a "monotone" or "helpful" adversary masking the planted instance. Conceptually, when an algorithm fails given this "helpful" information, it may have overfit to its generative assumptions. This model has been studied in various statistical settings [Jer92, FK00, FK01, MPW16, MMV12]. Of particular relevance to our work, which studies robustness to semi-random noise in the context of fast algorithms (as opposed to the distinction between polynomial-time algorithms and computational intractability) is [CG18], which developed an algorithm for semi-random matrix completion.

## 1.5 Organization

We give preliminaries and the notation used throughout the paper in Section 2. We prove our main results on efficiently solving matrix-dictionary approximation SDPs, Theorems 6 and 7, in Section 3. As relatively direct demonstrations of the utility of our solvers, we next present our results on solving in perturbed Laplacians and inverse matrices with combinatorial structure, i.e., Theorems 3, 4, and 5, in Section 4. We give our results on outer and inner scaling variants of diagonal preconditioning, i.e., Theorems 1 and 2, in Section 5. Finally, we present the implications of our inner scaling solver for semi-random statistical linear regression in Section 6.

Various proofs throughout the paper are deferred to Appendices A and B. We present our results on Jacobi preconditioning in Appendix C, and our improvements to our diagonal preconditioning results (assuming a width-independent positive SDP solver) in Appendix D.

## 2 Preliminaries

**General notation.** We let $[n] := \{1, 2, \cdots, n\}$. Applied to a vector, $\|\cdot\|_p$ is the $\ell_p$ norm. Applied to a matrix, $\|\cdot\|_2$ is overloaded to mean the $\ell_2$ operator norm. $\mathcal{N}(\mu, \boldsymbol{\Sigma})$ denotes the multivariate Gaussian with specified mean and covariance. $\Delta^n$ is the simplex in $n$ dimensions (the subset of $\mathbb{R}^n_{\geq 0}$ with unit $\ell_1$ norm). We use $\widetilde{O}$ to hide polylogarithmic factors in problem conditioning, dimensions, the target accuracy, and the failure probability. We say $\alpha \in \mathbb{R}$ is an $(\epsilon, \delta)$-approximation to $\beta \in \mathbb{R}$ if $\alpha = (1+\epsilon')\beta + \delta'$, for $|\epsilon'| \leq \epsilon$, $|\delta'| \leq \delta$. An $(\epsilon, 0)$-approximation is an "$\epsilon$-multiplicative approximation" and a $(0, \delta)$-approximation is a "$\delta$-additive approximation". We let $\mathcal{N}(\mu, \boldsymbol{\Sigma})$ denote the multivariate Gaussian distribution of specified mean and covariance.

**Matrices.** Throughout, matrices are denoted in boldface. We use nnz($\mathbf{A}$) to denote the number of nonzero entries of a matrix $\mathbf{A}$. The set of $d \times d$ symmetric matrices is denoted $\mathbb{S}^d$, and the positive semidefinite and definite cones are $\mathbb{S}^d_{\succeq \mathbf{0}}$ and $\mathbb{S}^d_{\succ \mathbf{0}}$ respectively. For $\mathbf{A} \in \mathbb{S}^d$, let $\lambda_{\max}(\mathbf{A})$, $\lambda_{\min}(\mathbf{A})$, and Tr($\mathbf{A}$) denote the largest magnitude eigenvalue, smallest eigenvalue, and trace. For $\mathbf{A} \in \mathbb{S}^d_{\succ \mathbf{0}}$, let $\kappa(\mathbf{M}) := \frac{\lambda_{\max}(\mathbf{M})}{\lambda_{\min}(\mathbf{M})}$ denote the condition number. We let Im($\mathbf{A}$) refer to the image of $\mathbf{A}$, and use $\mathbf{A}^\dagger$ to denote the pseudoinverse of $\mathbf{A} \in \mathbb{S}^d_{\succeq \mathbf{0}}$. The inner product between matrices $\mathbf{M}, \mathbf{N} \in \mathbb{S}^d$ is the trace product, $\langle \mathbf{M}, \mathbf{N} \rangle := \text{Tr}(\mathbf{M}\mathbf{N}) = \sum_{i,j \in [d]} \mathbf{M}_{ij} \mathbf{N}_{ij}$. We use the Loewner order on $\mathbb{S}^d$: $\mathbf{M} \preceq \mathbf{N}$ if and only if $\mathbf{N} - \mathbf{M} \in \mathbb{S}^d_{\succeq \mathbf{0}}$. $\mathbf{I}$ is the identity of appropriate dimension when clear. $\mathbf{diag}(w)$ for $w \in \mathbb{R}^n$ is the diagonal matrix with diagonal entries $w$. For $\mathbf{M} \in \mathbb{S}^d_{\succ \mathbf{0}}$, $\|v\|_{\mathbf{M}} := \sqrt{v^\top \mathbf{M} v}$. For $\mathbf{M} \in \mathbb{S}^d$ with eigendecomposition $\mathbf{V}^\top \boldsymbol{\Lambda} \mathbf{V}$, $\exp(\mathbf{M}) := \mathbf{V}^\top \exp(\boldsymbol{\Lambda}) \mathbf{V}$, where $\exp(\boldsymbol{\Lambda})$ is applies entrywise to the diagonal. Similarly for $\mathbf{M} = \mathbf{V}^\top \boldsymbol{\Lambda} \mathbf{V} \in \mathbb{S}^d_{\succeq \mathbf{0}}$, $\mathbf{M}^{\frac{1}{2}} := \mathbf{V}^\top \boldsymbol{\Lambda}^{\frac{1}{2}} \mathbf{V}$.

We denote the rows and columns of $\mathbf{A} \in \mathbb{R}^{n \times d}$ by $\mathbf{A}_{i:}$ for $i \in [n]$ and $\mathbf{A}_{:j}$ for $j \in [d]$ respectively. Finally, $\mathcal{T}_{\mathrm{mv}}(\mathbf{M})$ denotes the time it takes to multiply a vector $v$ by $\mathbf{M}$. We similarly denote the total cost of vector multiplication through a set $\{\mathbf{M}_i\}_{i \in [n]}$ by $\mathcal{T}_{\mathrm{mv}}(\{\mathbf{M}_i\}_{i \in [n]})$. We assume that $\mathcal{T}_{\mathrm{mv}}(\mathbf{M}) = \Omega(d)$ for any $d \times d$ matrix, as that time is generally required to write the output.

When discussing a graph on $n$ vertices, the elements of $V$, consider an edge $e = (u, v)$ for $u, v \in V$. We let $b_e \in \mathbb{R}^n$ denote the 2-sparse vector with a 1 in index $u$ and a $-1$ in index $v$.

# 3 Efficient matrix-dictionary recovery

In this section, we develop general solvers for the types of structured "mixed packing-covering" problems defined in Section 1.3, which we collectively call matrix-dictionary approximation SDPs.

In Section 3.1, we solve a basic version of this problem where $\mathbf{B} = \mathbf{I}$, i.e., the constraints are multiples of the identity. In Section 3.2, we give a more general solver able to handle arbitrary constraints, whose runtime depends polylogarithmically on the conditioning of said constraints. Our main results Theorems 8 and 9 are proven at the ends of Sections 3.1 and 3.2.

## 3.1 Identity constraints

In this section, we consider the special case of the problem (5), (6) in which $\mathbf{B} = \mathbf{I}$. To solve this problem, we first develop a framework for solving the decision variant of the problem (5), (6). Given a set of matrices $\{\mathbf{M}_i\}_{i \in [n]} \in \mathbb{S}_{\succeq \mathbf{0}}^d$ and a parameter $\kappa \geq 1$, we wish to determine

$$\text{does there exist } w \in \mathbb{R}_{\geq 0}^n \text{ such that } \lambda_{\max}\left(\sum_{i \in [n]} w_i \mathbf{M}_i\right) \leq \kappa \lambda_{\min}\left(\sum_{i \in [n]} w_i \mathbf{M}_i\right)? \qquad (10)$$

We note that the problem (10) is a special case of the more general mixed packing-covering semidefinite programming problem defined in [JLL+20], with packing matrices $\{\mathbf{M}_i\}_{i \in [n]}$ and covering matrices $\{\kappa \mathbf{M}_i\}_{i \in [n]}$. We define an $\epsilon$-*approximate tester* for the decision problem (10) to be an algorithm which returns "yes" whenever (10) is feasible for the value $(1 - \epsilon)\kappa$ (along with weights $w \in \mathbb{R}_{\geq 0}^n$ certifying this feasibility), and "no" whenever it is infeasible for the value $(1 + \epsilon)\kappa$ (and can return either answer in the middle range). After developing such a tester, we apply it to solve the (approximate) optimization variant (5), (6) by incrementally searching for the optimal $\kappa$.

To develop an approximate tester for (10), we require access to an algorithm for solving the optimization variant of a pure packing SDP,

$$\text{OPT}(v) := \max_{w \in \mathbb{R}_{\geq 0}^n \,:\, \sum_{i \in [n]} w_i \mathbf{M}_i \preceq \mathbf{I}} v^\top w. \qquad (11)$$

The algorithm is based on combining a solver for the testing variant of (11) by [JLT20] with a binary search. We state its guarantees as Proposition 1, and defer a proof to Appendix A.

**Proposition 1.** *Let* $\text{OPT}_+$ *and* $\text{OPT}_-$ *be known upper and lower bounds on* $\text{OPT}(v)$ *as in* (11). *There is an algorithm,* $\mathcal{A}_{\mathrm{pack}}$, *which succeeds with probability* $\geq 1 - \delta$, *whose runtime is*

$$O\left(\mathcal{T}_{\mathrm{mv}}\left(\{\mathbf{M}_i\}_{i \in [n]}\right) \cdot \frac{\log^2(ndT(\delta\epsilon)^{-1})\log^2 d}{\epsilon^5}\right) \cdot T \text{ for } T = O\left(\log\log \frac{\text{OPT}_+}{\text{OPT}_-} + \log \frac{1}{\epsilon}\right),$$

*and returns an* $\epsilon$-*multiplicative approximation to* $\text{OPT}(v)$, *and* $w$ *attaining this approximation.*

We require one additional tool, a regret analysis of matrix multiplicative weights from [ZLO15].

**Proposition 2** (Theorem 3.1, [ZLO15]). *Consider a sequence of gain matrices $\{\mathbf{G}_t\}_{0 \le t < T} \subset \mathbb{S}_{\succeq \mathbf{0}}^d$, which all satisfy for step size $\eta > 0$, $\|\eta \mathbf{G}_t\|_2 \le 1$. Then iteratively defining (from $\mathbf{S}_0 := \mathbf{0}$)*

$$\mathbf{Y}_t := \frac{\exp(\mathbf{S}_t)}{\operatorname{Tr}\exp(\mathbf{S}_t)}, \ \mathbf{S}_{t+1} := \mathbf{S}_t - \eta \mathbf{G}_t,$$

*we have the bound for any $\mathbf{U} \in \mathbb{S}_{\succeq \mathbf{0}}^d$ with $\operatorname{Tr}(\mathbf{U}) = 1$,*

$$\frac{1}{T}\sum_{0 \le t < T} \langle \mathbf{G}_t, \mathbf{Y}_t - \mathbf{U}\rangle \le \frac{\log d}{\eta T} + \frac{1}{T}\sum_{t \in [T]} \eta \|\mathbf{G}_t\|_2 \langle \mathbf{G}_t, \mathbf{Y}_t\rangle.$$

Finally, we are ready to state our $\epsilon$-approximate tester for the decision problem (10) as Algorithm 1. For simplicity in its analysis, we assume each matrix dictionary element's top eigenvalue is in a bounded range. We explicitly bound the cost of achieving this assumption in our applications (which can be achieved via rescaling by a constant-factor approximation to the top eigenvalue of each matrix using the power method, see Fact 3), and this does not dominate the runtime. The runtime bottleneck in all our applications is the cost of approximate packing SDP oracles in Line 7; this is an active research area and improvements therein would also reflect in our algorithm's runtime.

**Lemma 1.** *Algorithm 1 meets its output guarantees (as specified on Line 2).*

*Proof.* Throughout, assume all calls to $\mathcal{A}_{\mathrm{pack}}$ and the computation of approximations as given by Lines 6 and 13 succeed. By union bounding over $T$ iterations, this gives the failure probability.

We first show that if the algorithm terminates on Line 15, it is always correct. By the definition of $\mathcal{A}_{\mathrm{pack}}$, all $\mathbf{G}_t \preceq \kappa \mathbf{I}$, so throughout the algorithm, $-\mathbf{S}_{t+1} \preceq \eta \kappa T \mathbf{I} \preceq \frac{11\kappa \log d}{\epsilon}$. If the check on Line 14 passes, we must have $-\mathbf{S}_{t+1} \succeq \frac{11 \log d}{\epsilon}\mathbf{I}$, and hence the matrix $-\frac{1}{t+1}\mathbf{S}_{t+1}$ has condition number at most $\kappa$. The conclusion follows since the reweighting $\bar{x}$ induces $\mathbf{S}_{t+1}$.

We next prove correctness in the "no" case. Suppose the problem (12) is feasible; we show that the check in Line 9 will never pass (so the algorithm never returns "no"). Let $v_t^\star$ be the vector which is entrywise exactly $\{\langle \mathbf{M}_i, \mathbf{Y}_t\rangle\}_{i \in [n]}$, and let $v_t'$ be a $\frac{\epsilon}{10}$-multiplicative approximation to $v_t^\star$ such that $v_t$ is an entrywise $\frac{\epsilon}{10n}$-additive approximation to $v_t'$. By definition, it is clear $\mathrm{OPT}(\kappa v_t') \ge (1 - \frac{\epsilon}{10})\mathrm{OPT}(\kappa v_t^\star)$. Moreover, by the assumption that all $\lambda_{\max}(\mathbf{M}_i) \ge 1$, all $w_i \le 1$ in the feasible region of the problem (11). Hence, the combined additive error incurred by the approximation $\langle \kappa v_t, w\rangle$ to $\langle \kappa v_t', w\rangle$ for any feasible $w$ is $\frac{\epsilon}{10}$. All told, by the guarantee of $\mathcal{A}_{\mathrm{pack}}$,

$$\kappa\langle v_t, x_t\rangle \ge \left(1 - \frac{\epsilon}{10}\right)^2 \mathrm{OPT}(\kappa v_t^\star) - \frac{\epsilon}{10}, \ \text{where } \mathrm{OPT}(\kappa v_t^\star) = \max_{\substack{\sum_{i \in [n]} w_i \mathbf{M}_i \preceq \mathbf{I} \\ w \in \mathbb{R}_{\ge 0}^n}} \kappa\left\langle \mathbf{Y}_t, \sum_{i \in [n]} w_i \mathbf{M}_i\right\rangle. \quad (14)$$

However, by feasibility of (12) and scale invariance, there exists a $w \in \mathbb{R}_{\ge 0}^n$ with $\sum_{i \in [n]} w_i \mathbf{M}_i \preceq \mathbf{I}$ and $(1 - \epsilon)\kappa \sum_{i \in [n]} w_i \mathbf{M}_i \succeq \mathbf{I}$. Since $\mathbf{Y}_t$ has trace 1, this certifies $\mathrm{OPT}(\kappa v_t^\star) \ge \frac{1}{1-\epsilon}$, and thus

$$\kappa\langle v_t, x_t\rangle \ge \left(1 - \frac{\epsilon}{10}\right)^2 \cdot \frac{1}{1-\epsilon} - \frac{\epsilon}{10} > 1 - \frac{\epsilon}{5}.$$

Hence, whenever the algorithm returns "no" it is correct. Assume for the remainder of the proof that "yes" is returned on Line 18. Next, we observe that whenever $\mathcal{A}$ succeeds on iteration $t$,

**Algorithm 1** DecideStructuredMPC($\{\mathbf{M}_i\}_{i\in[n]}, \kappa, \mathcal{A}_{\text{pack}}, \delta, \epsilon$)

1: **Input:** $\{\mathbf{M}_i\}_{i\in[n]} \in \mathbb{S}_{\succeq\mathbf{0}}^{d\times d}$ such that $1 \leq \lambda_{\max}(\mathbf{M}_i) \leq 2$ for all $i \in [n]$, $\kappa > 1$, $\mathcal{A}_{\text{pack}}$ which on input $v \in \mathbb{R}_{\geq 0}^n$ returns $w \in \mathbb{R}_{\geq 0}^n$ satisfying (recalling definition (11))

$$\sum_{i\in[n]} w_i\mathbf{M}_i \preceq \mathbf{I}, \; v^\top w \geq \left(1 - \frac{\epsilon}{10}\right)\text{OPT}(v), \text{ with probability} \geq 1 - \frac{\delta}{2T} \text{ for some } T = O\left(\frac{\kappa\log d}{\epsilon^2}\right),$$

failure probability $\delta \in (0,1)$, tolerance $\epsilon \in (0,1)$

2: **Output:** With probability $\geq 1 - \delta$: "yes" or "no" is returned. The algorithm must return "yes" if there exists $w \in \mathbb{R}_{\geq 0}^n$ with

$$\lambda_{\max}\left(\sum_{i\in[n]} w_i\mathbf{M}_i\right) \leq (1-\epsilon)\kappa\lambda_{\min}\left(\sum_{i\in[n]} w_i\mathbf{M}_i\right), \tag{12}$$

and if "yes" is returned, a vector $w$ is given with

$$\lambda_{\max}\left(\sum_{i\in[n]} w_i\mathbf{M}_i\right) \leq (1+\epsilon)\kappa\lambda_{\min}\left(\sum_{i\in[n]} w_i\mathbf{M}_i\right). \tag{13}$$

3: $\eta \leftarrow \frac{\epsilon}{10\kappa}$, $T \leftarrow \left\lceil\frac{10\log d}{\eta\epsilon}\right\rceil$, $\mathbf{Y}_0 \leftarrow \frac{1}{d}\mathbf{I}$, $\mathbf{S}_0 \leftarrow \mathbf{0}$
4: **for** $0 \leq t < T$ **do**
5: $\quad \mathbf{Y}_t \leftarrow \frac{\exp(\mathbf{S}_t)}{\text{Tr}\exp(\mathbf{S}_t)}$
6: $\quad v_t \leftarrow$ entrywise nonnegative $(\frac{\epsilon}{10}, \frac{\epsilon}{10\kappa n})$-approximations to $\{\langle\mathbf{M}_i, \mathbf{Y}_t\rangle\}_{i\in[n]}$, with probability $\geq 1 - \frac{\delta}{4T}$
7: $\quad x_t \leftarrow \mathcal{A}_{\text{pack}}(\kappa v_t)$
8: $\quad \mathbf{G}_t \leftarrow \kappa\sum_{i\in[n]}[x_t]_i\mathbf{M}_i$
9: $\quad$ **if** $\kappa\langle x_t, v_t\rangle < 1 - \frac{\epsilon}{5}$ **then**
10: $\quad\quad$ **return** "no"
11: $\quad$ **end if**
12: $\quad \mathbf{S}_{t+1} \leftarrow \mathbf{S}_t - \eta\mathbf{G}_t$
13: $\quad \tau \leftarrow \frac{\log d}{\epsilon}$-additive approximation to $\lambda_{\min}(-\mathbf{S}_{t+1})$, with probability $\geq 1 - \frac{\delta}{4T}$
14: $\quad$ **if** $\tau \geq \frac{12\log d}{\epsilon}$ **then**
15: $\quad\quad$ **return** ("yes", $\bar{x}$) for $\bar{x} := \frac{1}{t+1}\sum_{0\leq s\leq t} x_s$
16: $\quad$ **end if**
17: **end for**
18: **return** ("yes", $\bar{x}$) for $\bar{x} := \frac{1}{T}\sum_{0\leq t<T} x_t$

---

$\sum_{i\in[n]}[x_t]_i\mathbf{M}_i \preceq \mathbf{I}$, and hence in every iteration we have $\|\mathbf{G}_t\|_2 \leq \kappa$. Proposition 2 then gives

$$\frac{1}{T}\sum_{0\leq t<T}\langle\mathbf{G}_t, \mathbf{Y}_t - \mathbf{U}\rangle \leq \frac{\log d}{\eta T} + \frac{1}{T}\sum_{t\in[T]}\eta\|\mathbf{G}_t\|_2\langle\mathbf{G}_t, \mathbf{Y}_t\rangle, \text{ for all } \mathbf{U} \in \mathbb{S}_{\succeq\mathbf{0}}^d \text{ with } \text{Tr}(\mathbf{U}) = 1.$$

Rearranging the above display, using $\eta \left\| \mathbf{G}_t \right\|_2 \leq \frac{\epsilon}{10}$, and minimizing over $\mathbf{U}$ yields

$$\lambda_{\min}\left(\frac{1}{T}\sum_{0\leq t<T}\mathbf{G}_t\right) \geq \frac{1-\frac{\epsilon}{10}}{T}\sum_{0\leq t<T}\langle\mathbf{G}_t,\mathbf{Y}_t\rangle - \frac{\log d}{\eta T} \geq \frac{1-\frac{\epsilon}{10}}{T}\sum_{0\leq t<T}\langle\mathbf{G}_t,\mathbf{Y}_t\rangle - \frac{\epsilon}{10}.$$

The last inequality used the definition of $T$. However, by definition of $v_t$, we have for all $0 \leq t < T$,

$$\langle\mathbf{Y}_t,\mathbf{G}_t\rangle = \kappa\sum_{i\in[n]}[x_t]_i\langle\mathbf{M}_i,\mathbf{Y}_t\rangle \geq \left(1-\frac{\epsilon}{10}\right)\kappa\langle x_t,v_t\rangle \geq \left(1-\frac{\epsilon}{10}\right)\left(1-\frac{\epsilon}{5}\right) \geq 1-\frac{3\epsilon}{10}. \qquad (15)$$

The second-to-last inequality used that Line 9 did not pass. Combining the previous two displays,

$$\kappa\lambda_{\min}\left(\sum_{i\in[n]}\bar{x}_i\mathbf{M}_i\right) = \lambda_{\min}\left(\frac{1}{T}\sum_{0\leq t<T}\mathbf{G}_t\right) \geq \left(1-\frac{\epsilon}{10}\right)\left(1-\frac{3\epsilon}{10}\right) - \frac{\epsilon}{10} \geq 1-\frac{\epsilon}{2}.$$

On the other hand, since all $0 \leq t < T$ have $\sum_{i\in[n]}[x_t]_i\mathbf{M}_i \preceq \mathbf{I}$, by convexity $\sum_{i\in[n]}\bar{x}_i\mathbf{M}_i \preceq \mathbf{I}$. Combining these two guarantees and $(1+\epsilon)(1-\frac{\epsilon}{2}) \geq 1$ shows $\bar{x}$ is correct for the "yes" case. $\qquad\square$

We bound the runtime complexities of Lines 6, 7, and 13 of Algorithm 1 in the following sections.

### 3.1.1 Approximating inner products

In this section, we bound the complexity of Line 6 of Algorithm 1. We will use the following two standard helper results on random projections and approximation theory.

**Fact 1** (Johnson-Lindenstrauss [DG03]). *For $0 \leq \epsilon \leq 1$, let $k = \Theta\left(\frac{1}{\epsilon^2}\log\frac{d}{\delta}\right)$ for an appropriate constant. For $\mathbf{Q} \in \mathbb{R}^{k\times d}$ with independent uniformly random unit vector rows in $\mathbb{R}^d$ scaled down by $\frac{1}{\sqrt{k}}$, with probability $\geq 1-\delta$ for any fixed $v \in \mathbb{R}^d$,*

$$(1-\epsilon)\left\|\mathbf{Q}v\right\|_2^2 \leq \left\|v\right\|_2^2 \leq (1+\epsilon)\left\|\mathbf{Q}v\right\|_2^2.$$

**Fact 2** (Polynomial approximation of exp [SV14], Theorem 4.1). *Let $\mathbf{M} \in \mathbb{S}_{\succeq\mathbf{0}}^d$ have $\mathbf{M} \preceq R\mathbf{I}$. Then for any $\delta > 0$, there is an explicit polynomial $p$ of degree $O(\sqrt{R\log\frac{1}{\delta}} + \log^2\frac{1}{\delta})$ with*

$$\exp(-\mathbf{M}) - \delta\mathbf{I} \preceq p(\mathbf{M}) \preceq \exp(-\mathbf{M}) + \delta\mathbf{I}.$$

We also state a simple corollary of Fact 2.

**Corollary 1.** *Given $R > 1$, $\mathbf{M} \in \mathbb{S}_{\succeq\mathbf{0}}^d$, and $\kappa$ with $\mathbf{M} \preceq \kappa\mathbf{I}$, we can compute a degree-$O(\sqrt{\kappa R} + R)$ polynomial $p$ such that for $\mathbf{P} = p(\mathbf{M})$,*

$$\exp\left(-\mathbf{M}\right) - \exp\left(-R\right)\mathbf{I} \preceq \mathbf{P} \preceq \exp\left(-\mathbf{M}\right) + \exp\left(-R\right)\mathbf{I}.$$

Using these tools, we next demonstrate that we can efficiently approximate the trace of a negative exponential of a bounded matrix, and quadratic forms through it.

**Lemma 2.** *Given $\mathbf{M} \in \mathbb{S}_{\succeq\mathbf{0}}^d$, $R, \kappa, \epsilon > 0$ such that $\lambda_{\min}(\mathbf{M}) \leq R$ and $\lambda_{\max}(\mathbf{M}) \leq \kappa R$, $\delta \in (0,1)$, we can compute an $\epsilon$-multiplicative approximation to $\operatorname{Tr}\exp(-\mathbf{M})$ with probability $\geq 1-\delta$ in time*

$$O\left(\mathcal{T}_{\mathrm{mv}}(\mathbf{M})\cdot\sqrt{\kappa R}\cdot\frac{\log\frac{d}{\delta}}{\epsilon^2}\right).$$

*Proof.* First, with probability at least $1 - \delta$, choosing $k = O(\frac{1}{\epsilon^2} \log \frac{d}{\delta})$ in Fact 1 and taking a union bound guarantees that for all rows $j \in [d]$, we have

$$\left\| \mathbf{Q} \left[ \exp \left( -\frac{1}{2}\mathbf{M} \right) \right]_{j:} \right\|_2^2 \text{ is a } \frac{\epsilon}{3}\text{-multiplicative approximation of } \left\| \left[ \exp \left( -\frac{1}{2}\mathbf{M} \right) \right]_{j:} \right\|_2^2.$$

Condition on this event in the remainder of the proof. The definition

$$\operatorname{Tr} \exp(-\mathbf{M}) = \sum_{j \in [d]} \left\| \left[ \exp \left( -\frac{1}{2}\mathbf{M} \right) \right]_{j:} \right\|_2^2,$$

and the sequence of equalities

$$\sum_{j \in [d]} \left\| \mathbf{Q} \left[ \exp \left( -\frac{1}{2}\mathbf{M} \right) \right]_{j:} \right\|_2^2 = \operatorname{Tr} \left( \exp \left( -\frac{1}{2}\mathbf{M} \right) \mathbf{Q}^\top \mathbf{Q} \exp \left( -\frac{1}{2}\mathbf{M} \right) \right)$$

$$= \operatorname{Tr} \left( \mathbf{Q} \exp \left( -\mathbf{M} \right) \mathbf{Q}^\top \right) = \sum_{\ell \in [k]} \left\| \exp \left( -\frac{1}{2}\mathbf{M} \right) \mathbf{Q}_{\ell:} \right\|_2^2,$$

implies that it suffices to obtain a $\frac{\epsilon}{3}$-multiplicative approximation to the last sum in the above display. Since $\operatorname{Tr} \exp(-\mathbf{M}) \geq \exp(-R)$ by the assumption on $\lambda_{\min}(\mathbf{M})$, it then suffices to approximate each term $\left\| \exp(-\frac{1}{2}\mathbf{M})\mathbf{Q}_{\ell:} \right\|_2^2$ to an additive $\frac{\epsilon}{3k} \exp(-R)$. For simplicity, fix some $\ell \in [k]$ and denote $q := \mathbf{Q}_{\ell:}$; recall $\|q\|_2^2 = \frac{1}{k}$ from the definition of $\mathbf{Q}$ in Fact 1.

By rescaling, it suffices to demonstrate that on any unit vector $q \in \mathbb{R}^d$, we can approximate $\left\| \exp(-\frac{1}{2}\mathbf{M})q \right\|_2^2$ to an additive $\frac{\epsilon}{3} \exp(-R)$. To this end, we note that (after shifting the definition of $R$ by a constant) Corollary 1 provides a matrix $\mathbf{P}$ with $\mathcal{T}_{\mathrm{mv}}(\mathbf{P}) = O(\mathcal{T}_{\mathrm{mv}}(\mathbf{M}) \cdot \sqrt{\kappa}R)$ and

$$\exp \left( -\mathbf{M} \right) - \frac{\epsilon}{3} \exp \left( -R \right) \mathbf{I} \preceq \mathbf{P} \preceq \exp \left( -\mathbf{M} \right) + \frac{\epsilon}{3} \exp \left( -R \right) \mathbf{I},$$

which exactly meets our requirements by taking quadratic forms. The runtime follows from the cost of applying $\mathbf{P}$ to each of the $k = O(\frac{1}{\epsilon^2} \log \frac{d}{\delta})$ rows of $\mathbf{Q}$. $\qquad \square$

**Lemma 3.** *Given $\mathbf{M} \in \mathbb{S}_{\geq 0}^d$ and $\kappa$ with $\mathbf{M} \preceq \kappa\mathbf{I}$, $1 > c > 0$, $\delta \in (0, 1)$, $\epsilon > 0$, and a set of matrices $\{\mathbf{M}_i\}_{i \in [n]}$ with decompositions of the form (7) and $1 \leq \lambda_{\max}(\mathbf{M}_i) \leq 2$ for all $i \in [n]$, we can compute $(\epsilon, c)$-approximations to all $\{\langle \mathbf{M}_i, \exp(-\mathbf{M}) \rangle\}_{i \in [n]}$, with probability $\geq 1 - \delta$ in time*

$$O \left( \mathcal{T}_{\mathrm{mv}} \left( \mathbf{M}, \{\mathbf{V}_i\}_{i \in [n]} \right) \cdot \sqrt{\kappa} \log \frac{c}{m} \cdot \frac{\log \frac{mn}{\delta}}{\epsilon^2} \right).$$

*Proof.* First, observe that for all $i \in [n]$, letting $\{v_j^{(i)}\}_{j \in [m]}$ be columns of $\mathbf{V}_i \in \mathbb{R}^{d \times m}$, we have

$$\langle \mathbf{M}_i, \exp(-\mathbf{M}) \rangle = \sum_{j \in [m]} \left( v_j^{(i)} \right)^\top \exp(-\mathbf{M}) \left( v_j^{(i)} \right).$$

Hence, to provide an $(\epsilon, c)$ approximation to $\langle \mathbf{M}_i, \exp(-\mathbf{M}) \rangle$ it suffices to provide, for all $i \in [n]$, $j \in [m]$, an $(\epsilon, \frac{c}{m})$-approximation to $\left( v_j^{(i)} \right)^\top \exp(-\mathbf{M}) \left( v_j^{(i)} \right)$. As in the proof of Lemma 2, by

taking a union bound it suffices to sample a $\mathbf{Q} \in \mathbb{R}^{k \times d}$ for $k = O(\frac{1}{\epsilon^2} \log \frac{mn}{\delta})$ and instead compute all $\|\mathbf{Q} \exp(-\frac{1}{2}\mathbf{M})v_j^{(i)}\|_2^2$ to additive error $\frac{c}{m}$. We will instead show how to approximate, for arbitrary vectors $q, v$ with norm at most 1,

$$\left\langle q, \exp\left(-\frac{1}{2}\mathbf{M}\right) v \right\rangle^2 \quad \text{to additive error } \frac{c}{2m}.$$

By letting $q$ range over rows of $\mathbf{Q}$ renormalized by $\sqrt{k}$, and scaling all $v_j^{(i)}$ by a factor of $\sqrt{2}$, this yields the desired result. To this end, consider using $\langle q, \mathbf{P}v \rangle^2$ for some $\mathbf{P}$ with $-\frac{c}{6m}\mathbf{I} \preceq \mathbf{P} - \exp(-\frac{1}{2}\mathbf{M}) \preceq \frac{c}{6m}\mathbf{I}$. Letting the difference matrix be $\mathbf{D} := \mathbf{P} - \exp(-\frac{1}{2}\mathbf{M})$, we compute

$$\left(q^\top \exp\left(-\frac{1}{2}\mathbf{M}\right) v\right)^2 - \left(q^\top \mathbf{P}v\right)^2 = 2\left(q^\top \exp\left(-\frac{1}{2}\mathbf{M}\right) v\right)\left(q^\top \mathbf{D}v\right) + \left(q^\top \mathbf{D}v\right)^2$$

$$\leq 2\|\mathbf{D}\|_2 + \|\mathbf{D}\|_2^2 \leq \frac{c}{2m}.$$

We used $q$ and $v$ have $\ell_2$ norm at most 1, $\exp(-\frac{1}{2}\mathbf{M}) \preceq \mathbf{I}$, and $\|\mathbf{D}\|_2 \leq \frac{c}{6m} \leq 1$. Hence, $\langle q, \mathbf{P}v \rangle^2$ is a valid approximation. The requisite $\mathbf{P}$ is given by Corollary 1 with $\mathcal{T}_{\mathrm{mv}}(\mathbf{P}) = O(\mathcal{T}_{\mathrm{mv}}(\mathbf{M}) \cdot \sqrt{\kappa} \log \frac{c}{m})$.

Finally, the runtime follows from first applying $\mathbf{P}$ to rows of $\mathbf{Q}$ to explicitly form $\widetilde{\mathbf{Q}}$ with $k$ rows, and then computing all $\|\widetilde{\mathbf{Q}}v_j^{(i)}\|_2^2$ for all $i \in [n]$, $j \in [m]$. □

Combining Lemmas 2 and 3, we bound the cost of Line 6 in Algorithm 1.

**Lemma 4.** *We can implement Line 6 of Algorithm 1 in time*

$$O\left(\mathcal{T}_{\mathrm{mv}}\left(\{\mathbf{V}_i\}_{i \in [n]}\right) \cdot \sqrt{\kappa} \cdot \frac{\log^3(\frac{mnd\kappa}{\delta\epsilon})}{\epsilon^3}\right).$$

*Proof.* Since $-\mathbf{S}_t$ is an explicit linear combination of $\{\mathbf{M}_i\}_{i \in [n]}$, we have $\mathcal{T}_{\mathrm{mv}}(\mathbf{S}_t) = O(\mathcal{T}_{\mathrm{mv}}(\{\mathbf{V}_i\}_{i \in [n]}))$. We first obtain a $\frac{\epsilon}{30}$ approximation to the denominator in Line 6 within the required time by applying Lemma 2 with $\kappa \leftarrow O(\kappa)$, $R \leftarrow O(\frac{\log d}{\epsilon})$, $\epsilon \leftarrow \frac{\epsilon}{30}$, and adjusting the failure probability by $O(T)$. The bound on $\lambda_{\min}(-\mathbf{S}_t)$ comes from the check on Line 13 and the algorithm yields the bound on $\lambda_{\max}(-\mathbf{S}_t)$. Next, we obtain a $(\frac{\epsilon}{30}, \frac{\epsilon}{30\kappa n})$ approximation to each numerator in Line 6 within the required time by using Lemma 3 with $\kappa \leftarrow O(\frac{\kappa \log d}{\epsilon})$ and adjusting constants appropriately. Combining these approximations to the numerators and denominator yields the result. □

### 3.1.2 Implementing a packing oracle

In this section, we bound the complexity of Line 7 of Algorithm 1 by using Proposition 1.

**Lemma 5.** *We can implement Line 7 of Algorithm 1 in time*

$$O\left(\mathcal{T}_{\mathrm{mv}}\left(\{\mathbf{V}_i\}_{i \in [n]}\right) \cdot \frac{\log^5(\frac{nd\kappa}{\delta\epsilon})}{\epsilon^5}\right).$$

*Proof.* First, we observe that the proof of the "no" case in Lemma 1 demonstrates that in all calls to $\mathcal{A}$, we can set our lower bound $\mathrm{OPT}_- = 1 - O(\epsilon)$, since the binary search of Proposition 1 will never need to check smaller values to determine whether the test on Line 9 passes. On the other hand, the definition of $\mathrm{OPT}(\kappa v_t^\star)$ in (14), as well as $\mathrm{OPT}(\kappa v_t) \leq (1 + \frac{\epsilon}{10})\mathrm{OPT}(\kappa v_t^\star)$ by the multiplicative approximation guarantee, shows that it suffices to set $\mathrm{OPT}_+ \leq (1 + O(\epsilon))\kappa$.

We will use the algorithm of Proposition 1 as $\mathcal{A}_{\text{pack}}$ in Algorithm 1. In our setting, we argued $\frac{\text{OPT}_+}{\text{OPT}_-} = O(\kappa)$, giving the desired runtime bound via Proposition 1. $\qquad\square$

### 3.1.3 Approximating the smallest eigenvalue

In this section, we bound the complexity of Line 13 of Algorithm 1, which asks to approximate the smallest eigenvalue of a matrix $\mathbf{M}$ to additive error. At a high level, our strategy is to use the power method on the negative exponential $\exp(-\mathbf{M})$, which we approximate to additive error via Corollary 1. We first state a guarantee on the classical power method from [MM15], which approximates the top eigenspace (see also [RST09, HMT11] for earlier analyses of the power method).

**Fact 3** (Theorem 1, [MM15]). *For any $\delta \in (0, 1)$ and $\mathbf{M} \in \mathbb{S}_{\geq 0}^d$, there is an algorithm, $\mathsf{Power}(\mathbf{M}, \delta)$, which returns with probability at least $1 - \delta$ a value $V$ such that $\lambda_{\max}(\mathbf{M}) \geq V \geq 0.9\lambda_{\max}(\mathbf{M})$. The algorithm runs in time $O(\mathcal{T}_{\text{mv}}(\mathbf{M}) \log \frac{d}{\delta})$, and is performed as follows:*

1. *Let $u \in \mathbb{R}^d$ be a random unit vector.*
2. *For some $\Delta = O(\log \frac{d}{\delta})$, let $v \leftarrow \frac{\mathbf{M}^\Delta u}{\|\mathbf{M}^\Delta u\|_2}$.*
3. *Return $\|\mathbf{M}v\|_2$.*

**Lemma 6.** *We can implement Line 13 of Algorithm 1 in time*

$$O\left(\mathcal{T}_{\text{mv}}\left(\{\mathbf{V}_i\}_{i \in [n]}\right) \cdot \sqrt{\kappa} \cdot \frac{\log^2(\frac{d\kappa}{\delta\epsilon})}{\epsilon}\right).$$

*Proof.* Throughout, denote $\mathbf{M} := -\mathbf{S}_{t+1}$, $L := \lambda_{\min}(\mathbf{M})$ and $R := \frac{\log d}{\epsilon}$, and note that (assuming all previous calls succeeded), we must have $L \leq 14R$ since the previous iteration had $L \leq 13R$ and $\mathbf{S}_t$ is changing by an $O(\epsilon)$-spectrally bounded matrix each iteration. It suffices to obtain $V$ with

$$0.9(\exp(-L) - \exp(-20R)) \leq V \leq \exp(-L), \tag{16}$$

and then return $-\log(V)$, to obtain an $R$-additive approximation to $L$. To see this, it is immediate that $-\log(V) \geq L$ from the above display. Moreover, for the given ranges of $L$ and $R$, it is clear

$$\exp(-20R + L) \leq \exp(-6R) \leq 1 - \frac{3}{2R}$$

$$\implies \exp(-L) - \exp(-20R) \geq \exp(-L) \cdot \frac{3}{2R}$$

$$\implies \exp(-L) - \exp(-20R) \geq \exp\left(-L - \frac{2R}{3}\right)$$

$$\implies \log\left(\frac{1}{\exp(-L) - \exp(-20R)}\right) \leq L + \frac{2R}{3}.$$

Combining with

$$-\log(V) \leq \log\left(\frac{1}{\exp(-L) - \exp(-20R)}\right) + \log\frac{10}{9} \leq \log\left(\frac{1}{\exp(-L) - \exp(-20R)}\right) + \frac{R}{3}$$

yields the claim. It hence suffices to provide $V$ satisfying (16) in the requisite time. To do so, we first use Corollary 1 with $\kappa \leftarrow O(\frac{\kappa \log d}{\epsilon})$ to produce $\mathbf{P}$ with $\mathcal{T}_{\text{mv}}(\mathbf{P}) = O(\mathcal{T}_{\text{mv}}(\mathbf{M}) \cdot \sqrt{\kappa} \cdot \frac{\log d}{\epsilon})$ and

$$\exp(-\mathbf{M}) - \frac{1}{2}\exp(-20R)\mathbf{I} \preceq \mathbf{P} \preceq \exp(-\mathbf{M}) + \frac{1}{2}\exp(-20R)\mathbf{I}.$$

The conclusion follows by applying Fact 3 to $\mathbf{P} - \frac{1}{2}\exp(-20R)\mathbf{I}$ and adjusting $\delta$. $\qquad\square$

### 3.1.4 Runtime of the optimization variant

Finally, we put these pieces together to solve the optimization variant of (5), (6). We begin by stating the runtime of Algorithm 1, which follows from combining Lemmas 4, 5, and 6.

**Corollary 2.** *Algorithm 1 can be implemented in time*

$$O\left(\mathcal{T}_{\mathrm{mv}}(\{\mathbf{V}_i\}) \cdot \kappa^{1.5} \cdot \frac{\log^6(\frac{mnd\kappa}{\delta\epsilon})}{\epsilon^7}\right).$$

**Theorem 8.** *Given matrices $\{\mathbf{M}_i\}_{i\in[n]}$ with explicit factorizations (7), such that (5) is feasible for $\mathbf{B} = \mathbf{I}$ and some $\kappa \geq 1$, we can return weights $w \in \mathbb{R}^n_{\geq 0}$ satisfying (6) with probability $\geq 1 - \delta$ in time*

$$O\left(\mathcal{T}_{\mathrm{mv}}(\{\mathbf{V}_i\}_{i\in[n]}) \cdot \kappa^{1.5} \cdot \frac{\log^7(\frac{mnd\kappa}{\delta\epsilon})}{\epsilon^7}\right).$$

*Proof.* First, to guarantee all $\mathbf{M}_i$ satisfy $1 \leq \lambda_{\max}(\mathbf{M}_i) \leq 2$, we exploit the scale-invariance of the problem (5), (6) and rescale each matrix by a 2-approximation to its largest eigenvalue. This can be done with Fact 3 and does not bottleneck the runtime.

Next, we perform an incremental search on $\kappa$ initialized at 1, and increasing in multiples of 2. By determining the first guess of $\kappa$ such that Algorithm 1 returns "yes," we obtain a 2-approximation to the optimal $\kappa$ at a $\log\kappa$ overhead from Corollary 2. We then can binary search at multiples of $1 + O(\epsilon)$ amongst the multiplicative range of 2 to obtain the required multiplicative approximation, at a $\log\frac{1}{\epsilon}$ overhead from Corollary 2, yielding the overall runtime. $\qquad\square$

## 3.2 General constraints

In this section, we consider a more general setting in which there is a constraint matrix $\mathbf{B}$ in the problem (5), (6) which we have matrix-vector product access to, but we cannot efficiently invert $\mathbf{B}$. We show that we can obtain a runtime similar to that in Theorem 8 (up to logarithmic factors and one factor of $\sqrt{\kappa}$), with an overhead depending polylogarithmically on $\alpha$ and $\beta$ defined in (9) and restated here for convenience:

$$\mathbf{B} \preceq \mathcal{M}(\mathbb{1}) \preceq \alpha\mathbf{B}, \ \mathbf{I} \preceq \mathbf{B} \preceq \beta\mathbf{I}.$$

### 3.2.1 Homotopy method preliminaries

Our algorithm will be a "homotopy method" which iteratively finds reweightings of the $\{\mathbf{M}_i\}_{i\in[n]}$ which $(1+\epsilon)\kappa$-approximate $\mathbf{B}+\lambda\mathcal{M}(\mathbb{1})$, for a sequence of $\lambda$ values. More specifically, we first bound the required range of $\lambda$ via two simple observations.

**Lemma 7.** *Let $\lambda \geq \frac{1}{\epsilon}$. Then,*

$$\mathbf{B} + \lambda\mathcal{M}(\mathbb{1}) \preceq \sum_{i\in[n]}(1+\lambda)\mathbf{M}_i \preceq (1+\epsilon)\left(\mathbf{B} + \lambda\mathcal{M}(\mathbb{1})\right).$$

*Proof.* The first inequality is immediate from (9). The second follows from $\mathbf{B} \succeq \mathbf{0}$. $\qquad\square$

**Lemma 8.** *Let $\lambda \le \frac{\epsilon\kappa}{2\alpha}$, and let $w \in \mathbb{R}_{\ge 0}^n$ satisfy*

$$\mathbf{B} + \lambda\mathcal{M}(\mathbb{1}) \preceq \sum_{i \in [n]} w_i\mathbf{M}_i \preceq (1 + \epsilon)\kappa\left(\mathbf{B} + \lambda\mathcal{M}(\mathbb{1})\right).$$

*Then, the same $w$ satisfies*

$$\mathbf{B} \preceq \sum_{i \in [n]} w_i\mathbf{M}_i \preceq (1 + 2\epsilon)\kappa\mathbf{B}.$$

*Proof.* The first inequality is immediate. The second is equivalent to

$$\epsilon\kappa\mathbf{B} \succeq (1 + \epsilon)\lambda\mathcal{M}(\mathbb{1})$$

which follows from $\epsilon\kappa \ge (1 + \epsilon)\lambda\alpha$ and (9). $\qquad\square$

Our homotopy method is driven by the observation that if (5) is feasible for a value of $\kappa$, then it is also feasible for the same $\kappa$ when $\mathbf{B}$ is replaced with $\mathbf{B} + \lambda\mathcal{M}(\mathbb{1})$ for any $\lambda \ge 0$.

**Lemma 9.** *For any $\lambda \ge 0$, if (5) is feasible for $\kappa \ge 1$, there also exists $w_\lambda^\star \in \mathbb{R}_{\ge 0}^n$ with*

$$\mathbf{B} + \lambda\mathcal{M}(\mathbb{1}) \preceq \sum_{i \in [n]} [w_\lambda^\star]_i\mathbf{M}_i \preceq \kappa\left(\mathbf{B} + \lambda\mathcal{M}(\mathbb{1})\right).$$

*Proof.* It suffices to choose $w_\lambda^\star = w^\star + \lambda\mathbb{1}$ where $w^\star$ is feasible for (5). $\qquad\square$

To this end, our algorithm will proceed in $K$ phases, where $K = \lceil \log_2 \frac{2\alpha}{\epsilon^2\kappa} \rceil$, setting

$$\lambda^{(0)} = \frac{1}{\epsilon}, \; \lambda^{(k)} = \frac{\lambda^{(0)}}{2^k} \text{ for all } 0 \le k \le K.$$

In each phase $k$ for $0 \le k \le K$, we will solve the problem (5), (6) for $\mathbf{B} \leftarrow \mathbf{B} + \lambda^{(k)}\mathcal{M}(\mathbb{1})$. To do so, we will use the fact that from the previous phase we have access to a matrix which is a $3\kappa$-spectral approximation to $\mathbf{B}$ which we can efficiently invert.

**Lemma 10.** *Suppose for some $\lambda \ge 0$, $\kappa \ge 1$, and $0 < \epsilon \le \frac{1}{2}$, we have $w \in \mathbb{R}_{\ge 0}^n$ such that*

$$\mathbf{B} + \lambda\mathcal{M}(\mathbb{1}) \preceq \sum_{i \in [n]} w_i\mathbf{M}_i \preceq (1 + \epsilon)\kappa(\mathbf{B} + \lambda\mathcal{M}(\mathbb{1})).$$

*Then defining $\mathbf{D} := \mathcal{M}(w)$, we have*

$$\mathbf{B} + \frac{\lambda}{2}\mathcal{M}(\mathbb{1}) \preceq \mathbf{D} \preceq 3\kappa\left(\mathbf{B} + \frac{\lambda}{2}\mathcal{M}(\mathbb{1})\right).$$

*Proof.* This is immediate from the assumption and

$$\mathbf{B} + \lambda\mathcal{M}(\mathbb{1}) \preceq 2\left(\mathbf{B} + \frac{\lambda}{2}\mathcal{M}(\mathbb{1})\right).$$

$\qquad\square$

Now, Lemma 7 shows we can access weights satisfying (6) for $\mathbf{B} \leftarrow \mathbf{B} + \lambda^{(0)}\mathcal{M}(\mathbb{1})$, and Lemma 8 shows if we can iteratively compute weights satisfying (6) for each $\mathbf{B} + \lambda^{(k)}\mathcal{M}(\mathbb{1})$, $0 \le k \le K$, then we

solve the original problem up to a constant factor in $\epsilon$. Moreover, Lemma 9 implies that the problem (5) is feasible for all phases assuming it is feasible with some value of $\kappa$ for the original problem. Finally, Lemma 10 shows that we start each phase with a matrix $\mathcal{M}(w)$ which we can efficiently invert using the linear operator in (8), which is a $3\kappa$-spectral approximation to the constraint matrix. We solve the self-contained subproblem of providing approximate inverse square root access (when granted a preconditioner) in the following section, by using rational approximations to the square root, and an appropriate call to Theorem 8.

### 3.2.2 Inverse square root approximations with a preconditioner

In this section, we show how to efficiently approximate the inverse square root of some $\mathbf{B} \in \mathbb{S}_{\succ \mathbf{0}}^d$ given access to a matrix $\mathcal{M}(w) \in \mathbb{S}_{\succ \mathbf{0}}^d$ satisfying

$$\mathbf{B} \preceq \mathcal{M}(w) \preceq \kappa \mathbf{B}$$

and supporting efficient inverse access in the form of (8). For brevity in this section we will use $\widetilde{\mathcal{M}}_{\lambda,\epsilon}$ to denote the linear operator guaranteeing (8) for $\mathcal{M}(w) + \lambda \mathbf{I}$. Given this access, we will develop a subroutine for efficiently applying a linear operator $\mathbf{R} \in \mathbb{S}_{\succ \mathbf{0}}^d$ such that

$$\left\| (\mathbf{R} - \mathbf{B}^{-\frac{1}{2}})v \right\|_2 \leq \epsilon \left\| \mathbf{B}^{-\frac{1}{2}}v \right\|_2 \text{ for all } v \in \mathbb{R}^d. \tag{17}$$

We will also use $\mathcal{T}_{\mathrm{mv}}$ to denote $\mathcal{T}_{\mathrm{mv}}(\mathbf{B}) + \mathcal{T}_{\mathrm{mv}}(\mathcal{M}(w)) + \mathcal{T}_{\mathcal{M}}^{\mathrm{sol}}$ for brevity in this section, where $\mathcal{T}_{\mathcal{M}}^{\mathrm{sol}}$ is the cost of solving a system in $\mathcal{M}(w)$ to constant accuracy (see (8)). Our starting point is the following result in the literature on preconditioned accelerated gradient descent, which shows we can efficiently solve linear systems in combinations of $\mathbf{B}$ and $\mathbf{I}$.

**Lemma 11.** *Given any $\lambda \geq 0$ and $\epsilon > 0$, we can compute a linear operator $\widehat{\mathcal{M}}_{\lambda,\epsilon}$ such that*

$$\left\| (\widehat{\mathcal{M}}_{\lambda,\epsilon} - (\mathbf{B} + \lambda \mathbf{I})^{-1})v \right\|_2 \leq \epsilon \left\| (\mathbf{B} + \lambda \mathbf{I})^{-1}v \right\|_2 \text{ for all } v \in \mathbb{R}^d,$$

*and*

$$\mathcal{T}_{\mathrm{mv}}(\widehat{\mathcal{M}}_{\lambda,\epsilon}) = O\left( \mathcal{T}_{\mathrm{mv}} \cdot \sqrt{\kappa} \log \kappa \log \frac{1}{\epsilon} \right).$$

*Proof.* This follows from Theorem 4.4 in [JS21], the fact that $\mathcal{M}(w) + \lambda \mathbf{I}$ is a $\kappa$-spectral approximation to $\mathbf{B} + \lambda \mathbf{I}$, and our assumed linear operator $\widetilde{\mathcal{M}}_{\lambda,(10\kappa)^{-1}}$ which solves linear systems in $\mathcal{M}(w) + \lambda \mathbf{I}$ to relative error $\frac{1}{10\kappa}$ which can be applied in time $O(\mathcal{T}_{\mathrm{mv}} \cdot \log \kappa)$ (the assumption (8)). $\square$

We hence can apply and (approximately) invert matrices of the form $\mathbf{B} + \lambda \mathbf{I}$. We leverage this fact to approximate inverse square roots using the following approximation result.

**Proposition 3** (Lemma 13, [JS19]). *For any matrix $\mathbf{B}$ and $\epsilon \in (0,1)$, there is a rational function $r(\mathbf{B})$ of degree $L = O(\log \frac{1}{\epsilon} \log \kappa(\mathbf{B}))$ such that for all vectors $v \in \mathbb{R}^d$,*

$$\left\| (r(\mathbf{B}) - \mathbf{B}^{-\frac{1}{2}})v \right\|_2 \leq \epsilon \left\| \mathbf{B}^{-\frac{1}{2}}v \right\|_2.$$

*The rational function $r(\mathbf{B})$ has the form, for $\{\lambda_\ell, \nu_\ell\}_{\ell \in [L]} \cup \{\nu_{L+1}\} \subset \mathbb{R}_{\geq 0}$ computable in $O(L)$ time,*

$$r(\mathbf{B}) = \left( \prod_{\ell \in [L]} (\mathbf{B} + \lambda_\ell \mathbf{I}) \right) \left( \prod_{\ell \in [L+1]} (\mathbf{B} + \nu_\ell \mathbf{I})^{-1} \right). \tag{18}$$

We note that Proposition 3 as it is stated in [JS19] is for the square root (not the inverse square root), but these are equivalent since all terms in the rational approximation commute. We show how to use Lemma 11 to efficiently approximate the inverse denominator in (18).

**Lemma 12.** *For any vector $v \in \mathbb{R}^d$, $\{\nu_\ell\}_{\ell \in [L+1]} \subset \mathbb{R}_{\geq 0}$ and $\epsilon > 0$, we can compute a linear operator $\widehat{\mathbf{D}}_\epsilon$ such that for the denominator of* (18), *denoted*

$$\mathbf{D} := \prod_{\ell \in [L+1]} (\mathbf{B} + \nu_\ell \mathbf{I}),$$

*we have*

$$\left\| (\widehat{\mathbf{D}}_\epsilon - \mathbf{D}^{-1}) v \right\|_2 \leq \epsilon \left\| \mathbf{D}^{-1} v \right\|_2,$$

*and*

$$\mathcal{T}_{\mathrm{mv}}(\widehat{\mathbf{D}}_\epsilon) = O\left( \mathcal{T}_{\mathrm{mv}} \cdot \sqrt{\kappa} \cdot L^2 \log(\kappa) \log\left( \frac{\kappa L \cdot \kappa(\mathbf{B})}{\epsilon} \right) \right).$$

*Proof.* We give our linear operator $\widehat{\mathbf{D}}_\epsilon$ in the form of an algorithm, which applies a sequence of linear operators to $v$ in the allotted time. Denote $\mathbf{B}_\ell := \mathbf{B} + \nu_\ell \mathbf{I}$ for all $\ell \in [L+1]$. We also define

$$\mathbf{\Pi}_\ell := \prod_{i \in [\ell]} \mathbf{B}_i^{-1} \text{ for all } \ell \in [L+1], \text{ and } \mathbf{\Pi}_0 := \mathbf{I}.$$

We define a sequence of vectors $\{v_\ell\}_{0 \leq \ell \leq L+1}$ as follows: let $v_0 := v$, and for all $\ell \in [L+1]$ define

$$v_\ell \leftarrow \widehat{\mathcal{M}}_{\nu_{L+2-\ell}, \Delta} v_{\ell-1} \text{ for } \Delta := \frac{\epsilon}{3(L+1)\kappa(\mathbf{B})^{2(L+1)}}.$$

Here we use notation from Lemma 11. In particular, for the given $\Delta$, we have for all $\ell \in [L+1]$,

$$\left\| v_\ell - \mathbf{B}_{L+2-\ell}^{-1} v_{\ell-1} \right\|_2 \leq \Delta \left\| v_{\ell-1} \right\|_2. \tag{19}$$

Our algorithm will simply return $v_{L+1}$, which is the application of a linear operator to $v$ within the claimed runtime via Lemma 11. We now prove correctness. We first prove a bound on the sizes of this iterate sequence. By the triangle inequality and (19), for each $\ell \in [L+1]$ we have

$$\|v_\ell\|_2 \leq \left\| v_\ell - \mathbf{B}_{L+2-\ell}^{-1} v_{\ell-1} \right\|_2 + \left\| \mathbf{B}_{L+2-\ell}^{-1} v_{\ell-1} \right\|_2 \leq (1 + \Delta) \left\| \mathbf{B}_{L+2-\ell}^{-1} \right\|_2 \|v_{\ell-1}\|_2.$$

Applying this bound inductively, we have that

$$\|v_\ell\|_2 \leq (1+\Delta)^\ell \left( \prod_{i \in [\ell]} \left\| \mathbf{B}_{L+2-\ell}^{-1} \right\|_2 \right) \|v_0\|_2 \leq 3 \left( \prod_{i \in [\ell]} \left\| \mathbf{B}_{L+2-\ell}^{-1} \right\|_2 \right) \|v_0\|_2. \tag{20}$$

Next, we expand by the triangle inequality that

$$\|v_{L+1} - \mathbf{\Pi}_{L+1}v_0\|_2 \leq \sum_{\ell \in [L+1]} \|\mathbf{\Pi}_{L+1-\ell}v_\ell - \mathbf{\Pi}_{L+2-\ell}v_{\ell-1}\|_2$$

$$\leq \sum_{\ell \in [L+1]} \|\mathbf{\Pi}_{L+1-\ell}\|_2 \|v_\ell - \mathbf{B}_{L+2-\ell}^{-1}v_{\ell-1}\|_2$$

$$\leq \sum_{\ell \in [L+1]} \Delta \|\mathbf{\Pi}_{L+2-\ell}\|_2 \|v_{\ell-1}\|_2$$

$$\leq \sum_{\ell \in [L+1]} 3\Delta \|\mathbf{\Pi}_{L+1}\|_2 \|v_0\|_2 \leq 3\Delta(L+1) \|\mathbf{\Pi}_{L+1}\|_2 \|v_0\|_2.$$

The second inequality used commutativity of all $\{\mathbf{B}_\ell\}_{\ell \in [L+1]}$ and the definition of $\mathbf{\Pi}_{L+2-\ell}$, the third used the guarantee (19), the fourth used (20) and that all $\{\mathbf{B}_\ell\}_{\ell \in [L+1]}$ have similarly ordered eigenvalues, and the last is straightforward. Finally, correctness of returning $v_{L+1}$ follows from $\mathbf{D} = \mathbf{\Pi}_{L+1}$ and the bound

$$\kappa(\mathbf{\Pi}_{L+1}) \leq \kappa(\mathbf{B})^{L+1} \implies 3\Delta(L+1) \|\mathbf{\Pi}_{L+1}\|_2 \|v_0\|_2 \leq \epsilon \|\mathbf{\Pi}_{L+1}v_0\|_2.$$

To see the first claim, every $\mathbf{B}_\ell$ clearly has $\kappa(\mathbf{B}_\ell) \leq \kappa(\mathbf{B})$, and we used the standard facts that for commuting $\mathbf{A}, \mathbf{B} \in \mathbb{S}_{\succ 0}^d$, $\kappa(\mathbf{A}) = \kappa(\mathbf{A}^{-1})$ and $\kappa(\mathbf{AB}) \leq \kappa(\mathbf{A})\kappa(\mathbf{B})$. $\qquad\square$

At this point, obtaining the desired (17) follows from a direct application of Lemma 12 and the fact that the numerator and denominator of (18) commute with each other and $\mathbf{B}$.

**Lemma 13.** *For any vector $v \in \mathbb{R}^d$ and $\epsilon > 0$, we can compute a linear operator $\mathbf{R}_\epsilon$ such that*

$$\left\|(\mathbf{R}_\epsilon - \mathbf{B}^{-\frac{1}{2}})v\right\|_2 \leq \epsilon \left\|\mathbf{B}^{-\frac{1}{2}}v\right\|_2,$$

*and*

$$\mathcal{T}_{\mathrm{mv}}(\mathbf{R}_\epsilon) = O\left(\mathcal{T}_{\mathrm{mv}} \cdot \sqrt{\kappa} \cdot \log^6\left(\frac{\kappa \cdot \kappa(\mathbf{B})}{\epsilon}\right)\right).$$

*Proof.* Let $\epsilon' \leftarrow \frac{\epsilon}{3}$, and let $\mathbf{D}$ and $\mathbf{N}$ be the (commuting) numerator and denominator of (18) for the rational function in Proposition 3 for the approximation factor $\epsilon'$. Let $u \leftarrow \mathbf{N}v$, which we can compute explicitly within the alloted runtime. We then have from Proposition 3 that

$$\left\|\mathbf{D}^{-1}u - \mathbf{B}^{-\frac{1}{2}}v\right\|_2 \leq \epsilon' \left\|\mathbf{B}^{-\frac{1}{2}}v\right\|_2.$$

Moreover by Lemma 12 we can compute a vector $w$ within the allotted runtime such that

$$\|\mathbf{D}^{-1}u - w\|_2 \leq \epsilon' \|\mathbf{D}^{-1}u\|_2 \leq \epsilon' \left(\left\|\mathbf{B}^{-\frac{1}{2}}v\right\|_2 + \left\|\mathbf{D}^{-1}u - \mathbf{B}^{-\frac{1}{2}}v\right\|_2\right) \leq 2\epsilon' \left\|\mathbf{B}^{-\frac{1}{2}}v\right\|_2.$$

The vector $w$ follows from applying an explicit linear operator to $v$ as desired. Finally, by combining the above two displays we have the desired approximation quality as well:

$$\left\|\mathbf{B}^{-\frac{1}{2}}v - w\right\|_2 \leq 3\epsilon' \left\|\mathbf{B}^{-\frac{1}{2}}v\right\|_2 = \epsilon \left\|\mathbf{B}^{-\frac{1}{2}}v\right\|_2.$$

$\qquad\square$

### 3.2.3 Implementing the homotopy method

In this section, we implement the homotopy method outlined in Section 3.2.1 by using the inverse square root access given by Lemma 13. We require the following helper result.

**Lemma 14.** *Suppose for some $\epsilon \in (0, \frac{1}{2})$, and $\mathbf{M}, \mathbf{N} \in \mathbb{S}_{\succ \mathbf{0}}^d$ such that $\kappa(\mathbf{N}) \leq \beta \leq \frac{1}{3\epsilon}$,*

$$-\epsilon \mathbf{N}^{-1} \preceq \mathbf{M}^{-1} - \mathbf{N}^{-1} \preceq \epsilon \mathbf{N}^{-1}.$$

*Then,*

$$-9\epsilon\beta \mathbf{N}^2 \preceq \mathbf{M}^2 - \mathbf{N}^2 \preceq 9\epsilon\beta \mathbf{N}^2.$$

*Proof.* The statement is scale-invariant, so suppose for simplicity that $\mathbf{I} \preceq \mathbf{N} \preceq \beta\mathbf{I}$. Also, by rearranging the assumed bound, we have $-3\epsilon\mathbf{N} \preceq \mathbf{M} - \mathbf{N} \preceq 3\epsilon\mathbf{N}$. Write $\mathbf{D} = \mathbf{M} - \mathbf{N}$ such that

$$-3\epsilon\beta\mathbf{I} \preceq -3\epsilon\mathbf{N} \preceq \mathbf{D} \preceq 3\epsilon\mathbf{N} \preceq 3\epsilon\beta\mathbf{I}.$$

We bound the quadratic form of $\mathbf{M}^2 - \mathbf{N}^2$ with an arbitrary vector $v \in \mathbb{R}^d$, such that $\|\mathbf{N}v\|_2 = 1$. Since all eigenvalues of $\mathbf{D}$ are in $[-3\epsilon\beta, 3\epsilon\beta]$, this implies

$$\|\mathbf{D}v\|_2^2 = v^\top \mathbf{D}^2 v \leq 9\epsilon^2\beta^2 \|v\|_2^2 \leq 9\epsilon^2\beta^2 v^\top \mathbf{N}^2 v \leq 9\epsilon^2\beta^2$$

where we use $\mathbf{N} \succeq \mathbf{I}$ and hence $\mathbf{N}^2 \succeq \mathbf{I}$. We conclude by the triangle inequality and Cauchy-Schwarz:

$$\left| v^\top (\mathbf{M}^2 - \mathbf{N}^2) v \right| = \left| v^\top (\mathbf{N}^2 - (\mathbf{N} + \mathbf{D})^2) v \right|$$
$$= \left| v^\top (\mathbf{ND} + \mathbf{DN} + \mathbf{D}^2) v \right|$$
$$\leq 2 \|\mathbf{D}v\|_2 \|\mathbf{N}v\|_2 + \|\mathbf{D}v\|_2^2 \leq 6\epsilon\beta + 9\epsilon^2\beta^2 \leq 9\epsilon\beta.$$

$\square$

We now state our main claim regarding solving (5), (6) with general $\mathbf{B}$.

**Theorem 9.** *Given matrices $\{\mathbf{M}_i\}_{i \in [n]}$ with explicit factorizations (7), such that (5) is feasible for some $\kappa \geq 1$ and we can solve linear systems in linear combinations of $\{\mathbf{M}_i\}_{i \in [n]}$ to $\epsilon$ relative accuracy in the sense of (8) in $\mathcal{T}_\mathcal{M}^{\mathrm{sol}} \cdot \log \frac{1}{\epsilon}$ time, and $\mathbf{B}$ satisfying*

$$\mathbf{B} \preceq \mathcal{M}(\mathbb{1}) \preceq \alpha\mathbf{B}, \ \mathbf{I} \preceq \mathbf{B} \preceq \beta\mathbf{I},$$

*we can return weights $w \in \mathbb{R}_{\geq 0}^n$ satisfying (6) with probability $\geq 1 - \delta$ in time*

$$O\left( \left(\mathcal{T}_{\mathrm{mv}}\left(\{\mathbf{V}_i\}_{i \in [n]} \cup \{\mathbf{B}\}\right) + \mathcal{T}_\mathcal{M}^{\mathrm{sol}}\right) \cdot \kappa^2 \cdot \frac{\log^{12}(\frac{mnd\kappa\beta}{\delta\epsilon})}{\epsilon^7} \cdot \log \frac{\alpha}{\epsilon} \right).$$

*Proof.* We follow Section 3.2.1, which shows that it suffices to solve the following problem $O(\log \frac{\alpha}{\epsilon})$ times. We have an instance of (5), (6) for the original value of $\kappa$, and some matrix $\mathbf{B}$ with $\kappa(\mathbf{B}) \leq \beta$ such that we know $w \in \mathbb{R}_{\geq 0}^n$ with $\mathbf{B} \preceq \mathcal{M}(w) \preceq 3\kappa\mathbf{B}$. We wish to compute $w'$ with

$$\mathbf{B} \preceq \mathcal{M}(w') \preceq (1 + \epsilon)\kappa\mathbf{B}. \tag{21}$$

To do so, we use Lemma 13, which yields a matrix $\mathbf{R}$ such that

$$-\frac{\epsilon}{45\beta}\mathbf{B}^{-\frac{1}{2}} \preceq \mathbf{R} - \mathbf{B}^{-\frac{1}{2}} \preceq \frac{\epsilon}{45\beta}\mathbf{B}^{-\frac{1}{2}} \text{ and } \mathcal{T}_{\mathrm{mv}}(\mathbf{R}) = O\left(\mathcal{T}_{\mathrm{mv}}\left(\{\mathbf{V}_i\}_{i\in[n]} \cup \{\mathbf{B}\}\right) \cdot \sqrt{\kappa} \cdot \log^6\left(\frac{\kappa\beta}{\epsilon}\right)\right).$$

By Lemma 14, this implies that

$$-\frac{\epsilon}{5}\mathbf{B} \preceq \mathbf{R}^{-2} - \mathbf{B} \preceq \frac{\epsilon}{5}\mathbf{B}. \tag{22}$$

Now, if we solve (5), (6) with $\mathbf{R}^{-2}$ in place of $\mathbf{B}$ to accuracy $1 + \frac{\epsilon}{5}$, since the optimal value of $\kappa$ has changed by at most a factor of $1 + \frac{\epsilon}{5}$, it suffices to compute $w'$ such that

$$\mathbf{R}^{-2} \preceq \mathcal{M}(w') \preceq \left(1 + \frac{3\epsilon}{5}\right)\kappa\mathbf{R}^{-2},$$

since combined with (22) this implies (21) up to rescaling $w'$. Finally, to compute the above reweighting it suffices to apply Theorem 8 with $\mathbf{M}_i \leftarrow \mathbf{R}\mathbf{M}_i\mathbf{R}$ for all $i \in [n]$. It is clear these matrices satisfy (7) with the decompositions given by $\mathbf{V}_i \leftarrow \mathbf{R}\mathbf{V}_i$, and we can apply these matrices in time $\mathcal{T}_{\mathrm{mv}}(\mathbf{R}) + \mathcal{T}_{\mathrm{mv}}(\mathbf{V}_i)$. It is straightforward to check that throughout the proof of Theorem 8, this replaces each cost of $\mathcal{T}_{\mathrm{mv}}(\{\mathbf{V}_i\}_{i\in[n]})$ with $\mathcal{T}_{\mathrm{mv}}(\{\mathbf{V}_i\}_{i\in[n]}) + \mathcal{T}_{\mathrm{mv}}(\mathbf{R})$, giving the desired runtime. $\square$

## 4 Graph structured systems

In this section, we provide several applications of Theorem 9, restated for convenience.

**Theorem 9.** *Given matrices $\{\mathbf{M}_i\}_{i\in[n]}$ with explicit factorizations (7), such that (5) is feasible for some $\kappa \geq 1$ and we can solve linear systems in linear combinations of $\{\mathbf{M}_i\}_{i\in[n]}$ to $\epsilon$ relative accuracy in the sense of (8) in $\mathcal{T}_{\mathcal{M}}^{\mathrm{sol}} \cdot \log\frac{1}{\epsilon}$ time, and $\mathbf{B}$ satisfying*

$$\mathbf{B} \preceq \mathcal{M}(\mathbb{1}) \preceq \alpha\mathbf{B}, \ \mathbf{I} \preceq \mathbf{B} \preceq \beta\mathbf{I},$$

*we can return weights $w \in \mathbb{R}_{\geq 0}^n$ satisfying (6) with probability $\geq 1 - \delta$ in time*

$$O\left(\left(\mathcal{T}_{\mathrm{mv}}\left(\{\mathbf{V}_i\}_{i\in[n]} \cup \{\mathbf{B}\}\right) + \mathcal{T}_{\mathcal{M}}^{\mathrm{sol}}\right) \cdot \kappa^2 \cdot \frac{\log^{12}(\frac{mnd\kappa\beta}{\delta\epsilon})}{\epsilon^7} \cdot \log\frac{\alpha}{\epsilon}\right).$$

We show how to use Theorem 9 in a relatively straightforward fashion to derive further recovery and solving results for several types of natural structured matrices. Our first result of the section, Theorem 10, serves as an example of how a black-box application of Theorem 9 can be used to solve linear systems in, and spectrally approximate, families of matrices which are well-approximated by a "simple" matrix dictionary such as graph Laplacians; we call these matrices "perturbed Laplacians."

Our other two results in this section, Theorems 11 and 12, extend the applicability of Theorem 9 by wrapping it in a recursive preconditioning outer loop. We use this strategy, combined with new (simple) structural insights on graph-structured families, to give nearly-linear time solvers and spectral approximations for inverse M-matrices and Laplacian pseudoinverses.

Below, we record several key definitions and preliminaries we will use in this section. We also state key structural tools we leverage to prove these results, and defer formal proofs to Appendix B.

**Preliminaries: graph structured systems.** We call a square matrix $\mathbf{A}$ a Z-matrix if $\mathbf{A}_{ij} \leq 0$ for all $i \neq j$. We call a matrix $\mathbf{L}$ a Laplacian if it is a symmetric Z-matrix with $\mathbf{L}\mathbb{1} = 0$. We call a square matrix $\mathbf{A} \in \mathbb{R}^{d\times d}$ diagonally dominant (DD) if $\mathbf{A}_{ii} \geq \sum_{j\neq i} \mathbf{A}_{ij}$ for all $i \in [d]$ and symmetric

diagonally dominant (SDD) if it is DD and symmetric. We call $\mathbf{M}$ an invertible $M$-matrix if $\mathbf{M} = s\mathbf{I} - \mathbf{A}$ where $s > 0$, $\mathbf{A} \in \mathbb{R}^{d \times d}_{\geq 0}$, and $\rho(\mathbf{A}) < s$ where $\rho$ is the spectral radius.

For an undirected graph $G = (V, E)$ with nonnegative edge weights $w \in \mathbb{R}^E_{\geq 0}$, its Laplacian $\mathbf{L}$ is defined as $\sum_{e \in E} w_e b_e b_e^\top$, where for an edge $e = (u, v)$ we let $b_e \in \mathbb{R}^V$ be 1 in the $u$ coordinate and $-1$ in the $v$ coordinate. It is well-known that all Laplacian matrices can be written in this way for some graph, and are SDD [ST04]. Finally, we let $\mathbf{L}_{K_n} := n\mathbf{I} - \mathbb{1}\mathbb{1}^\top$ be the Laplacian of the unweighted complete graph on $n$ vertices, which is a scalar multiple of the projection matrix of the image space of all SDD matrices, the subspace of $\mathbb{R}^V$ orthogonal to $\mathbb{1}$.

**Lemma 15.** *Let* $\mathbf{M}$ *be an invertible symmetric M-matrix. Let* $x = \mathbf{M}^{-1}\mathbb{1}$ *and define* $\mathbf{X} = \mathbf{diag}(x)$. *Then* $\mathbf{XMX}$ *is a SDD Z-matrix.*

**Lemma 16.** *Let* $\mathbf{A}$ *be an invertible SDD Z-matrix. For any* $\alpha \geq 0$, *the matrix* $\mathbf{B} = (\mathbf{A}^{-1} + \alpha\mathbf{I})^{-1}$ *is also an invertible SDD Z-matrix.*

We also will make extensive use of a standard notion of preconditioned (accelerated) gradient descent, which we state here (note that we used a special case of the following result as Lemma 10).

**Proposition 4** (Theorem 4.4, [JS21]). *Suppose for* $\kappa \geq 1$ *and* $\mathbf{A}, \mathbf{B} \in \mathbb{S}^d_{\succeq \mathbf{0}}$ *sharing a kernel,* $\mathbf{A} \preceq \mathbf{B} \preceq \kappa\mathbf{A}$, *and suppose in time* $\mathcal{T}^{\mathrm{sol}}_{\mathbf{A}} \log \frac{1}{\epsilon}$ *we can compute and apply a linear operator* $\mathcal{A}_\epsilon$ *with*

$$\left\| (\mathcal{A}_\epsilon - \mathbf{A}^\dagger)v \right\|_2 \leq \epsilon \left\| \mathbf{A}^\dagger v \right\|_2 \ \text{for all } v \in \mathrm{Im}(\mathbf{A}).$$

*Then, we can compute and apply a linear operator* $\mathcal{B}_\epsilon$ *in time* $(\mathcal{T}^{\mathrm{sol}}_{\mathbf{A}} + \mathcal{T}_{\mathrm{mv}}(\mathbf{B}))\sqrt{\kappa} \log \kappa \log \frac{1}{\epsilon}$ *with*

$$\left\| (\mathcal{B}_\epsilon - \mathbf{B}^\dagger)v \right\|_2 \leq \epsilon \left\| \mathbf{B}^\dagger v \right\|_2 \ \text{for all } v \in \mathrm{Im}(\mathbf{B}).$$

### 4.1 Perturbed Laplacian solver

In this section, we prove Theorem 10 through a direct application of Theorem 9.

**Theorem 10** (Perturbed Laplacian solver). *Let* $\mathbf{A} \succeq \mathbf{0} \in \mathbb{R}^{n \times n}$ *be such that there exists an (unknown) Laplacian* $\mathbf{L}$ *with* $\mathbf{A} \preceq \mathbf{L} \preceq \kappa\mathbf{A}$, *and that* $\mathbf{L}$ *corresponds to a graph with edge weights between* $w_{\min}$ *and* $w_{\max}$, *with* $\frac{w_{\max}}{w_{\min}} \leq U$. *For any* $\delta \in (0, 1)$ *and* $\epsilon > 0$, *there is an algorithm recovering a Laplacian* $\mathbf{L}'$ *with* $\mathbf{A} \preceq \mathbf{L}' \preceq (1 + \epsilon)\kappa\mathbf{A}$ *with probability* $\geq 1 - \delta$ *in time*

$$O\left( n^2 \cdot \kappa^2 \cdot \frac{\log^{14}(\frac{n\kappa U}{\delta\epsilon})}{\epsilon^7} \right).$$

*Consequently, there is an algorithm for solving linear systems in* $\mathbf{A}$ *to* $\epsilon$-*relative accuracy (see* (8)) *with probability* $\geq 1 - \delta$, *in time*

$$O\left( n^2 \cdot \kappa^2 \cdot \log^{14}\left(\frac{n\kappa U}{\delta}\right) + n^2 \, \mathrm{polyloglog}(n) \log\left(\frac{1}{\delta}\right) \cdot \sqrt{\kappa} \log \kappa \cdot \log\left(\frac{1}{\epsilon}\right) \right).$$

*Proof.* First, we note that $\mathbf{\Pi} := \frac{1}{n}\mathbf{L}_{K_n}$ is the projection matrix onto the orthogonal complement of $\mathbb{1}$, and that $\mathbf{\Pi}$ can be applied to any vector in $\mathbb{R}^n$ in $O(n)$ time. Next, consider the matrix dictionary $\mathcal{M}$ consisting of the matrices

$$\mathbf{M}_e := b_e b_e^\top \ \text{for all } e = (u, v) \in \binom{V}{2}, \tag{23}$$

where we follow the notation of Section 2. Because all unweighted graphs have $\text{poly}(n)$-bounded condition number restricted to the image of $\mathbf{\Pi}$ [Spi19], it is clear that we may set $\alpha = \beta = \text{poly}(n, U)$ in Theorem 9 with $\mathbf{B} \leftarrow \mathbf{A}$. Moreover, feasibility of (6) for this dictionary $\mathcal{M}$ and approximation factor $\kappa$ holds by assumption. The first result then follows directly from Theorem 9, where we ensure that all vectors we work with are orthogonal to $\mathbb{1}$ by appropriately applying $\mathbf{\Pi}$, and recall that we may take $\mathcal{T}_{\mathcal{M}}^{\text{sol}} = n^2 \cdot \text{polyloglog}(n)$ as was shown in [JS21].

The second result follows by combining the first result for any sufficiently small constant $\epsilon$, Proposition 4, and the solver of [JS21] for the resulting Laplacian $\mathbf{L}'$; we note this can be turned into a high-probability solver by a standard application of Markov at the cost of $\log \frac{1}{\delta}$ overhead. $\qquad \square$

## 4.2 M-matrix recovery and inverse M-matrix solver

In this section, we prove Theorem 11. We first provide a proof sketch. Since $\mathbf{A} = \mathbf{M}^{-1}$ for some invertible M-matrix $\mathbf{M}$, we can compute $x = \mathbf{A}\mathbb{1}$ in $O(n^2)$ time and set $\mathbf{X} = \mathbf{diag}(x)$. Any spectral approximation $\mathbf{N} \approx \mathbf{XMX}$ will have $\mathbf{X}^{-1}\mathbf{N}\mathbf{X}^{-1} \approx \mathbf{M}$ spectrally with the same approximation factor. Moreover, $\mathbf{XMX}$ is an SDD Z-matrix by Lemma 15, and hence we can approximate it using Theorem 8 if we can efficiently apply matrix-vector products through it. The key is to provide matrix-vector access to a spectral approximation to $\mathbf{XMX} = (\mathbf{X}^{-1}\mathbf{A}\mathbf{X}^{-1})^{-1}$. Our algorithm does so by recursively providing spectral approximations to

$$\mathbf{A}_i := \lambda_i \mathbf{I} + \mathbf{A}$$

for a sequence of nonnegative $\lambda_i$, using a homotopy method similar to the one used in Theorem 9.

**Theorem 11** (M-matrix recovery and inverse M-matrix solver). *Let $\mathbf{A}$ be the inverse of an unknown invertible symmetric M-matrix, let $\kappa$ be an upper bound on its condition number, and let $U$ be the ratio of the entries of $\mathbf{A}\mathbb{1}$. For any $\delta \in (0, 1)$ and $\epsilon > 0$, there is an algorithm recovering a $(1 + \epsilon)$-spectral approximation to $\mathbf{A}^{-1}$ in time*

$$O\left(n^2 \cdot \frac{\log^{13}\left(\frac{n\kappa U}{\delta\epsilon}\right) \log\left(\frac{\kappa U}{\epsilon}\right) \log \frac{1}{\epsilon}}{\epsilon^7}\right).$$

*Consequently, there is an algorithm for solving linear systems in $\mathbf{A}$ to $\epsilon$-relative accuracy (see (8)) with probability $\geq 1 - \delta$, in time*

$$O\left(n^2 \cdot \left(\log^{14}\left(\frac{n\kappa U}{\delta}\right) + \log \frac{1}{\epsilon}\right)\right).$$

*Proof.* We begin with the first statement. In this proof, we will define a sequence of matrices

$$\mathbf{B}_i = \left(\mathbf{X}^{-1}\mathbf{A}\mathbf{X}^{-1} + \lambda_i\mathbf{I}\right)^{-1}, \ \lambda_i = \frac{\lambda_0}{2^i}.$$

By Lemma 16, each $\mathbf{B}_i$ is an invertible SDD Z-matrix, and hence can be perfectly spectrally approximated by the dictionary consisting of all matrices in (23), and $\{e_i e_i^\top\}_{i \in [n]}$ (1-sparse nonnegative diagonal matrices). We refer to this joint dictionary by $\mathcal{M}$. By a proof strategy similar to that of Theorem 9, it is clear we may set $\lambda_0$ and $K = O(\log \frac{\kappa U}{\epsilon})$ such that $\lambda_0^{-1}\mathbf{I}$ is a constant-factor spectral approximation to $\mathbf{B}_0$, and $\mathbf{B}_K$ is a $(1 + \frac{\epsilon}{4})$-factor spectral approximation to $\mathbf{X}^{-1}\mathbf{A}\mathbf{X}^{-1}$.

For each $0 \leq k \leq K - 1$, our algorithm recursively computes an explicit linear operator $\widetilde{\mathbf{B}}_i$ which is a $O(1)$-spectral approximation to $\mathbf{B}_i$, with $\widetilde{\mathbf{B}}_0 \leftarrow \lambda_0^{-1}\mathbf{I}$. Then by applying Proposition 4, the fact that $\widetilde{\mathbf{B}}_i$ is a constant spectral approximation to $\mathbf{B}_{i+1}$ by a variant of Lemma 10, and the fact that

we have explicit matrix-vector access to $\mathbf{B}_{i+1}^{-1}$, we can efficiently provide matrix-vector access to $\mathcal{B}_{i+1}$, a $1 + \frac{\epsilon}{4}$-spectral approximation to $\mathbf{B}_{i+1}$ in time

$$O\left(\left(\mathcal{T}_{\mathrm{mv}}(\widetilde{\mathbf{B}}_i) + \mathcal{T}_{\mathrm{mv}}(\mathbf{A})\right) \cdot \log\frac{1}{\epsilon}\right).$$

It remains to show how to compute $\widetilde{\mathbf{B}}_i$ and apply it efficiently. To do so, it suffices to set $\widetilde{\mathbf{B}}_i$ to be a $1 + \frac{\epsilon}{2}$-spectral approximation to $\mathcal{B}_{i+1}$ using the dictionary $\mathcal{M}$, which is clearly achievable using Theorem 9 because $\mathcal{B}_{i+1}$ is a $1 + \frac{\epsilon}{4}$-spectral approximation to an SDD Z-matrix, which is perfectly approximable. Hence, we have $\mathcal{T}_{\mathrm{mv}}(\widetilde{\mathbf{B}}_i) = n^2$ because we have explicit access to it through our matrix dictionary. Repeating this strategy $K$ times yields the second conclusion; the bottleneck step in terms of runtime is the $K$ calls to Theorem 9. Here we note that we have solver access for linear combinations of the dictionary $\mathcal{M}$ by a well-known reduction from SDD solvers to Laplacian solvers (see e.g., [KOSZ13]), and we apply the solver of [JS21].

Finally, for the second conclusion, it suffices to use Proposition 4 with the preconditioner $\widetilde{\mathbf{B}}_K$ we computed earlier for constant $\epsilon$, along with our explicit accesses to $\mathbf{X}$, and $\mathbf{A}$. $\qquad\square$

## 4.3 Laplacian recovery and Laplacian pseudoinverse solver

In this section, we prove Theorem 12. Our algorithm for Theorem 12 is very similar to the one we derived in proving Theorem 11, where for a sequence of nonnegative $\lambda_i$, we provide spectral approximations to

$$\mathbf{B}_i := \mathbf{A}_i^\dagger, \text{ where } \mathbf{A}_i := \lambda_i\mathbf{\Pi} + \mathbf{A}, \tag{24}$$

and throughout this section we use $\mathbf{\Pi} := \frac{1}{n}\mathbf{L}_{K_n}$ to be the projection onto the image of all Laplacian matrices. We provide the following helper lemma in the connected graph case, proven in Appendix B; a straightforward observation reduces to the connected case without loss.

**Lemma 17.** *If $\mathbf{A}^\dagger$ is a Laplacian of a connected graph, then $(\mathbf{A} + \alpha\mathbf{L}_{K_n})^\dagger$ for any $\alpha > 0$ is a Laplacian matrix.*

**Theorem 12** (Laplacian recovery and Laplacian pseudoinverse solver). *Let $\mathbf{A}$ be the pseudoinverse of unknown Laplacian $\mathbf{L}$, and that $\mathbf{L}$ corresponds to a graph with edge weights between $w_{\min}$ and $w_{\max}$, with $\frac{w_{\max}}{w_{\min}} \le U$. For any $\delta \in (0, 1)$ and $\epsilon > 0$, there is an algorithm recovering a Laplacian $\mathbf{L}'$ with $\mathbf{A}^\dagger \preceq \mathbf{L}' \preceq (1 + \epsilon)\mathbf{A}^\dagger$ in time*

$$O\left(n^2 \cdot \frac{\log^{13}\left(\frac{nU}{\delta\epsilon}\right)\log\left(\frac{nU}{\epsilon}\right)\log\frac{1}{\epsilon}}{\epsilon^7}\right).$$

*Consequently, there is an algorithm for solving linear systems in $\mathbf{A}$ to $\epsilon$-relative accuracy (see (6)) with probability $\ge 1 - \delta$, in time*

$$O\left(n^2 \cdot \left(\log^{14}\left(\frac{nU}{\delta}\right) + \log\frac{1}{\epsilon}\right)\right).$$

*Proof.* First, we reduce to the connected graph case by noting that the connected components of the graph corresponding to $\mathbf{L}$ yield the same block structure in $\mathbf{A} = \mathbf{L}^\dagger$. In particular, whenever $i$ and $j$ lie in different connected components, then $\mathbf{A}_{ij}$ will be 0, so we can reduce to multiple instances of the connected component case by partitioning $\mathbf{A}$ appropriately in $O(n^2)$ time.

To handle the connected graph case, we use the strategy of Theorem 11 to recursively approximate a sequence of matrices (24). It suffices to perform this recursion $K = O(\log \frac{nU}{\epsilon})$ times, since (as was argued in proving Theorem 10) the conditioning of $\mathbf{A}$ is bounded by $\text{poly}(n, U)$. The remainder of the proof is identical to Theorem 11, again using the solver of [JS21]. $\qquad\square$

## 5 Diagonal scaling

In this section, we provide a collection of results concerning the following two diagonal scaling problems, which we refer to as "inner scaling" and "outer scaling". In the inner scaling problem, we are given a full rank $n \times d$ matrix $\mathbf{A}$ with $n \geq d$, and the goal is to find an $n \times n$ nonnegative diagonal matrix $\mathbf{W}$ that minimizes or approximately minimizes $\kappa(\mathbf{A}^\top \mathbf{W} \mathbf{A})$. We refer to the optimal value of this problem by

$$\kappa_i^\star(\mathbf{A}) := \min_{\text{diagonal } \mathbf{W} \succeq \mathbf{0}} \kappa\left(\mathbf{A}^\top \mathbf{W} \mathbf{A}\right).$$

Similarly, in the outer scaling problem, we are given a full rank $d \times d$ matrix $\mathbf{K}$, and the goal is to find a $d \times d$ nonnegative diagonal matrix $\mathbf{W}$ that (approximately) minimizes $\kappa(\mathbf{W}^{\frac{1}{2}} \mathbf{K} \mathbf{W}^{\frac{1}{2}})$. The optimal value is denoted

$$\kappa_o^\star(\mathbf{K}) := \min_{\text{diagonal } \mathbf{W} \succeq \mathbf{0}} \kappa\left(\mathbf{W}^{\frac{1}{2}} \mathbf{K} \mathbf{W}^{\frac{1}{2}}\right).$$

In Appendix C, we give a simple, new result concerning a classic heuristic for solving the outer scaling problem. Our main technical contributions are fast algorithms for obtaining diagonal reweightings admitting constant-factor approximations to the optimal rescaled condition numbers $\kappa_i^\star$ and $\kappa_o^\star$. We provide the former result in Section 5.1 by a direct application of Theorem 8. A weaker version of the latter follows immediately from applying Theorem 9, but we obtain a strengthening by exploiting the structure of the outer scaling problem more carefully in Section 5.2. We state our main scaling results here for convenience, and defer proofs to respective sections.

**Theorem 13.** *Let $\epsilon > 0$ be a fixed constant. There is an algorithm, which given full-rank $\mathbf{A} \in \mathbb{R}^{n \times d}$ for $n \geq d$ computes $w \in \mathbb{R}_{\geq 0}^n$ such that $\kappa(\mathbf{A}^\top \mathbf{W} \mathbf{A}) \leq (1 + \epsilon)\kappa_i^\star(\mathbf{A})$ with probability $\geq 1 - \delta$ in time*

$$O\left(\mathcal{T}_{\text{mv}}(\mathbf{A}) \cdot (\kappa_i^\star(\mathbf{A}))^{1.5} \cdot \log^7\left(\frac{n\kappa_i^\star(\mathbf{A})}{\delta}\right)\right).$$

**Theorem 14.** *Let $\epsilon > 0$ be a fixed constant.[12] There is an algorithm, which given full-rank $\mathbf{K} \in \mathbb{S}_{\succ \mathbf{0}}^d$ computes $w \in \mathbb{R}_{\geq 0}^d$ such that $\kappa(\mathbf{W}^{\frac{1}{2}} \mathbf{K} \mathbf{W}^{\frac{1}{2}}) \leq (1 + \epsilon)\kappa_o^\star(\mathbf{K})$ with probability $\geq 1 - \delta$ in time*

$$O\left(\mathcal{T}_{\text{mv}}(\mathbf{K}) \cdot (\kappa_o^\star(\mathbf{K}))^{1.5} \cdot \log^8\left(\frac{d\kappa_o^\star(\mathbf{K})}{\delta}\right)\right).$$

### 5.1 Inner scaling

We prove Theorem 13 on computing constant-factor optimal inner scalings.

*Proof of Theorem 13.* We apply Theorem 8 with $\kappa = \kappa_i^\star(\mathbf{A})$ and $\mathbf{M}_i = a_i a_i^\top$ for all $i \in [n]$, where rows of $\mathbf{A}$ are denoted $\{a_i\}_{i \in [n]}$. The definition of $\kappa_i^\star(\mathbf{A})$ and $\mathbf{A}^\top \mathbf{W} \mathbf{A} = \sum_{i \in [n]} w_i a_i a_i^\top$ (where $\mathbf{W} = \mathbf{diag}(w)$) implies that (5) is feasible for these parameters: namely, there exists $w^\star \in \mathbb{R}_{\geq 0}^n$

---

[12]We do not focus on the $\epsilon$ dependence and instead take it to be constant since, in applications involving solving linear systems, there is little advantage to obtaining better than a two factor approximation (i.e., setting $\epsilon = 1$).

such that

$$\mathbf{I} \preceq \sum_{i \in [n]} w_i \mathbf{M}_i \preceq \kappa \mathbf{I}.$$

Moreover, factorizations (7) hold with $\mathbf{V}_i = a_i$ and $m = 1$. Note that $\mathcal{T}_{\mathrm{mv}}(\{\mathbf{V}_i\}_{i \in [n]}) = O(\mathrm{nnz}(\mathbf{A}))$ since this is the cost of multiplication through all rows of $\mathbf{A}$. $\qquad\square$

## 5.2 Outer scaling

In this section, we prove a result on computing constant-factor optimal outer scalings of a matrix $\mathbf{K} \in \mathbb{S}_{\succ \mathbf{0}}^d$. We first remark that we can obtain a result analogous to Theorem 13, but which scales quadratically in the $\kappa_o^\star(\mathbf{K})$, straightforwardly by applying Theorem 9 with $n = d$, $\mathbf{M}_i = e_i e_i^\top$ for all $i \in [d]$, $\kappa = \kappa_o^\star(\mathbf{K})$, and $\mathbf{B} = \frac{1}{\kappa} \mathbf{K}$. This is because the definition of $\kappa_o^\star$ implies there exists $w^\star \in \mathbb{R}_{\geq 0}^d$ with $\mathbf{I} \preceq (\mathbf{W}^\star)^{-\frac{1}{2}} \mathbf{K} (\mathbf{W}^\star)^{-\frac{1}{2}} \preceq \kappa \mathbf{I}$, where $\mathbf{W}^\star = \mathbf{diag}(w^\star)$ and where this $w^\star$ is the entrywise inverse of the optimal outer scaling attaining $\kappa_o^\star(\mathbf{K})$. This then implies

$$\frac{1}{\kappa} \mathbf{K} \preceq \sum_{i \in [d]} w_i^\star \mathbf{M}_i \preceq \mathbf{K},$$

since $\sum_{i \in [d]} w_i^\star \mathbf{M}_i = \mathbf{W}^\star$ and hence Theorem 9 applies, obtaining a runtime for the outer scaling problem of roughly $\mathcal{T}_{\mathrm{mv}}(\mathbf{K}) \cdot \kappa_o^\star(\mathbf{K})^2$ (up to logarithmic factors).

We give an alternative approach in this section which obtains a runtime of roughly $\mathcal{T}_{\mathrm{mv}}(\mathbf{K}) \cdot \kappa_o^\star(\mathbf{K})^{1.5}$, matching Theorem 13 up to logarithmic factors. Our approach is to define

$$\mathbf{A} := \mathbf{K}^{\frac{1}{2}}$$

to be the positive definite square root of $\mathbf{K}$; we use this notation throughout the section, and let $\{a_i\}_{i \in [d]}$ be the rows of $\mathbf{A}$. We cannot explicitly access $\mathbf{A}$, but if we could, directly applying Theorem 13 suffices because $\kappa(\mathbf{W}^{\frac{1}{2}} \mathbf{K} \mathbf{W}^{\frac{1}{2}}) = \kappa(\mathbf{W}^{\frac{1}{2}} \mathbf{A}^2 \mathbf{W}^{\frac{1}{2}}) = \kappa(\mathbf{A} \mathbf{W} \mathbf{A})$ for any nonnegative diagonal $\mathbf{W} \in \mathbb{R}^{d \times d}$. We show that by using a homotopy method similar to the one employed in Section 3.2, we can implement this strategy with only a polylogarithmic runtime overhead. At a high level, the improvement from Theorem 9 is because we have explicit access to $\mathbf{K} = \mathbf{A}^2$. By exploiting cancellations in polynomial approximations we can improve the cost of iterations of Algorithm 1 from roughly $\kappa$ (where one factor of $\sqrt{\kappa}$ comes from the cost of rational approximations to square roots in Section 3.2, and the other comes from the degree of polynomials), to roughly $\sqrt{\kappa}$.

Finally, throughout this section we will assume $\kappa(\mathbf{A}) \leq \kappa_o^\star(\mathbf{K})$, which is without loss of generality by rescaling based on the diagonal (Jacobi preconditioning), as we show in Appendix C. Also, for notational convenience we will fix a matrix $\mathbf{K} \in \mathbb{S}_{\succ \mathbf{0}}^d$ and denote $\kappa^\star := \kappa_o^\star(\mathbf{K})$.

### 5.2.1 Preliminaries

We first state a number of preliminary results which will be used in building our outer scaling method. We begin with a polynomial approximation to the square root, proven in Appendix A. It yields a corollary regarding approximating matrix-vector products with a matrix square root.

**Fact 4** (Polynomial approximation of $\sqrt{\cdot}$). *Let* $\mathbf{M} \in \mathbb{S}_{\succ \mathbf{0}}^d$ *have* $\mu \mathbf{I} \preceq \mathbf{M} \preceq \kappa \mu \mathbf{I}$ *where* $\mu$ *is known. Then for any* $\delta \in (0, 1)$, *there is an explicit polynomial* $p$ *of degree* $O(\sqrt{\kappa} \log \frac{\kappa}{\delta})$ *with*

$$(1 - \delta) \mathbf{M}^{\frac{1}{2}} \preceq p(\mathbf{M}) \preceq (1 + \delta) \mathbf{M}^{\frac{1}{2}}.$$

**Corollary 3.** *For any vector $b \in \mathbb{R}^d$, $\delta, \epsilon \in (0,1)$, and $\mathbf{M} \in \mathbb{S}^d_{\succ \mathbf{0}}$ with $\kappa(\mathbf{M}) \leq \kappa$, with probability $\geq 1 - \delta$ we can compute $u \in \mathbb{R}^d$ such that*

$$\left\| u - \mathbf{M}^{\frac{1}{2}} b \right\|_2 \leq \epsilon \left\| \mathbf{M}^{\frac{1}{2}} b \right\|_2 \quad \text{in time } O\left( \mathcal{T}_{\mathrm{mv}}(\mathbf{M}) \cdot \left( \sqrt{\kappa} \log \frac{\kappa}{\epsilon} + \log \frac{d}{\delta} \right) \right).$$

*Proof.* First, we compute a 2-approximation to $\mu$ in Fact 4 within the runtime budget using the power method (Fact 3), since $\kappa$ is given. This will only affect parameters in the remainder of the proof by constant factors. If $u = \mathbf{P}b$ for commuting $\mathbf{P}$ and $\mathbf{M}$, our requirement is equivalent to

$$-\epsilon^2 \mathbf{M} \preceq \left( \mathbf{P} - \mathbf{M}^{\frac{1}{2}} \right)^2 \preceq \epsilon^2 \mathbf{M}.$$

Since square roots are operator monotone (by the Löwner-Heinz inequality), this is true iff

$$-\epsilon \mathbf{M}^{\frac{1}{2}} \preceq \mathbf{P} - \mathbf{M}^{\frac{1}{2}} \preceq \epsilon \mathbf{M}^{\frac{1}{2}},$$

and such a $\mathbf{P}$ which is applicable within the runtime budget is given by Fact 4. $\qquad \square$

We next demonstrate two applications of Corollary 3 in estimating applications of products involving $\mathbf{A} = \mathbf{K}^{\frac{1}{2}}$, where we can only explicitly access $\mathbf{K}$. We will use the following standard fact about operator norms, whose proof is deferred to Appendix A.

**Lemma 18.** *Let $\mathbf{B} \in \mathbb{R}^{d \times d}$ and let $\mathbf{A} \in \mathbb{S}^d_{\succ \mathbf{0}}$. Then $\min \left( \|\mathbf{AB}\|_2, \|\mathbf{BA}\|_2 \right) \geq \frac{1}{\kappa(\mathbf{A})} \|\mathbf{B}\|_2 \|\mathbf{A}\|_2$.*

First, we discuss the application of a bounded-degree polynomial in $\mathbf{AWA}$ to a uniformly random unit vector, where $\mathbf{W}$ is an explicit nonnegative diagonal matrix.

**Lemma 19.** *Let $u \in \mathbb{R}^d$ be a uniformly random unit vector, let $\mathbf{K} \in \mathbb{S}^d_{\succ \mathbf{0}}$ such that $\mathbf{A} := \mathbf{K}^{\frac{1}{2}}$ and $\kappa(\mathbf{K}) \leq \kappa$, and let $\mathbf{P}$ be a degree-$\Delta$ polynomial in $\mathbf{AWA}$ for an explicit diagonal $\mathbf{W} \in \mathbb{S}^d_{\succeq \mathbf{0}}$. For $\delta, \epsilon \in (0,1)$, with probability $\geq 1 - \delta$ we can compute $w \in \mathbb{R}^d$ so $\|w - \mathbf{P}u\|_2 \leq \epsilon \|\mathbf{P}u\|_2$ in time*

$$O\left( \mathcal{T}_{\mathrm{mv}}(\mathbf{K}) \cdot \left( \Delta + \sqrt{\kappa} \log \frac{d\kappa}{\delta \epsilon} \right) \right).$$

*Proof.* We can write $\mathbf{P} = \mathbf{ANA}$ for some matrix $\mathbf{N}$ which is a degree-$O(\Delta)$ polynomial in $\mathbf{K}$ and $\mathbf{W}$, which we have explicit access to. Standard concentration bounds show that with probability at least $1 - \delta$, for some $N = \mathrm{poly}(d, \delta^{-1})$, $\|\mathbf{P}u\|_2 \geq \frac{1}{N} \|\mathbf{P}\|_2$. Condition on this event for the remainder of the proof, such that it suffices to obtain additive accuracy $\frac{\epsilon}{N} \|\mathbf{P}\|_2$. By two applications of Lemma 18, we have

$$\|\mathbf{ANA}\|_2 \geq \frac{1}{\kappa} \|\mathbf{A}\|_2^2 \|\mathbf{N}\|_2. \tag{25}$$

Our algorithm is as follows: for $\epsilon' \leftarrow \frac{\epsilon}{3N\kappa}$, compute $v$ such that $\|v - \mathbf{A}u\|_2 \leq \epsilon' \|\mathbf{A}\|_2 \|u\|_2$ using Corollary 3, explicitly apply $\mathbf{N}$, and then compute $w$ such that $\|w - \mathbf{AN}v\|_2 \leq \epsilon' \|\mathbf{A}\|_2 \|\mathbf{N}v\|_2$; the runtime of this algorithm clearly fits in the runtime budget. The desired approximation is via

$$
\begin{aligned}
\|w - \mathbf{ANA}u\|_2 &\leq \|w - \mathbf{AN}v\|_2 + \|\mathbf{AN}v - \mathbf{ANA}u\|_2 \\
&\leq \epsilon' \|\mathbf{A}\|_2 \|\mathbf{N}\|_2 \|v\|_2 + \|\mathbf{AN}v - \mathbf{ANA}u\|_2 \\
&\leq 2\epsilon' \|\mathbf{A}\|_2^2 \|\mathbf{N}\|_2 + \|\mathbf{A}\|_2 \|\mathbf{N}\|_2 \|v - \mathbf{A}u\|_2 \\
&\leq 2\epsilon' \|\mathbf{A}\|_2^2 \|\mathbf{N}\|_2 + \epsilon' \|\mathbf{A}\|_2^2 \|\mathbf{N}\|_2 \\
&\leq 3\epsilon' \kappa \|\mathbf{ANA}\|_2 = \frac{\epsilon}{N} \|\mathbf{P}\|_2.
\end{aligned}
$$

The third inequality used $\|v\|_2 \leq \|\mathbf{A}u\|_2 + \epsilon \|\mathbf{A}\|_2 \|u\|_2 \leq (1 + \epsilon) \|\mathbf{A}\|_2 \leq 2 \|\mathbf{A}\|_2$. $\qquad\square$

We give a similar guarantee for random bilinear forms through $\mathbf{A}$ involving an explicit vector.

**Lemma 20.** *Let $u \in \mathbb{R}^d$ be a uniformly random unit vector, let $\mathbf{K} \in \mathbb{S}_{\succ 0}^d$ such that $\mathbf{A} := \mathbf{K}^{\frac{1}{2}}$ and $\kappa(\mathbf{K}) \leq \kappa$, and let $v \in \mathbb{R}^d$. For $\delta, \epsilon \in (0, 1)$, with probability $\geq 1 - \delta$ we can compute $w \in \mathbb{R}^d$ so $\langle w, v \rangle$ is an $\epsilon$-multiplicative approximation to $u^\top \mathbf{A} v$ in time*

$$O\left( \mathcal{T}_{\mathrm{mv}}(\mathbf{K}) \cdot \sqrt{\kappa} \log \frac{d\kappa}{\delta\epsilon} \right).$$

*Proof.* As in Lemma 19, for some $N = \mathrm{poly}(d, \delta^{-1})$ it suffices to give a $\frac{\epsilon}{N} \|\mathbf{A}v\|_2$-additive approximation. For $\epsilon' \leftarrow \frac{\epsilon}{N\sqrt{\kappa}}$, we apply Corollary 3 to obtain $w$ such that $\|w - \mathbf{A}u\|_2 \leq \epsilon' \|\mathbf{A}u\|_2$, which fits within the runtime budget. Correctness follows from

$$|\langle \mathbf{A}u - w, v \rangle| \leq \|\mathbf{A}u - w\|_2 \|v\|_2 \leq \epsilon' \|\mathbf{A}\|_2 \|v\|_2 \leq \epsilon'\sqrt{\kappa} \|\mathbf{A}v\|_2 \leq \frac{\epsilon}{N} \|\mathbf{A}v\|_2.$$

$\qquad\square$

### 5.2.2 Implementing Algorithm 1 implicitly

In this section, we bound the complexity of Algorithm 1 in the following setting. Throughout this section denote $\mathbf{M}_i = a_i a_i^\top$ for all $i \in [d]$, where $\mathbf{A} \in \mathbb{S}_{\succ 0}^d$ has rows $\{a_i\}_{i \in [d]}$, and $\mathbf{A}^2 = \mathbf{K}$. We assume that

$$\kappa(\mathbf{K}) \leq \kappa_{\mathrm{scale}} := 3\kappa^\star,$$

and we wish to compute a reweighting $w \in \mathbb{R}_{\geq 0}^d$ such that

$$\kappa\left( \sum_{i \in [d]} w_i a_i a_i^\top \right) = \kappa\left( \mathbf{W}^{\frac{1}{2}} \mathbf{K} \mathbf{W}^{\frac{1}{2}} \right) \leq (1 + \epsilon)\kappa^\star,$$

assuming there exists a reweighting $w^\star \in \mathbb{R}_{\geq 0}^d$ such that above problem is feasible with conditioning $\kappa^\star$. In other words, we assume we start with a matrix whose conditioning is within a 3-factor of the optimum after rescaling, and wish to obtain a $1 + \epsilon$-approximation to the optimum. We show in the next section how to use a homotopy method to reduce the outer scaling problem to this setting.

Our strategy is to apply the method of Theorem 13. To deal with the fact that we cannot explictly access the matrix $\mathbf{A}$, we give a custom analysis of the costs of Lines 6, 7, and 13 under implicit access in this section, and prove a variant of Theorem 13 for this specific setting.

**Estimating the smallest eigenvalue implicitly.** We begin by discussing implicit implementation of Line 13. Our strategy combines the approach of Lemma 6 (applying the power method to the negative exponential), with Lemma 19 since to handle products through random vectors.

**Lemma 21.** *Given $\delta \in (0, 1)$, constant $\epsilon > 0$, $\mathbf{K} \in \mathbb{S}_{\succ 0}^d$ such that $\mathbf{A} := \mathbf{K}^{\frac{1}{2}}$ and $\kappa(\mathbf{K}) \leq \kappa_{\mathrm{scale}}$, and diagonal $\mathbf{W} \in \mathbb{S}_{\succeq 0}^d$ such that $\mathbf{M} := \mathbf{A}\mathbf{W}\mathbf{A} \preceq O(\kappa_{\mathrm{scale}} \log d)\mathbf{I}$, we can compute a $O(\log d)$-additive approximation to $\lambda_{\min}(\mathbf{M})$ with probability $\geq 1 - \delta$ in time*

$$O\left( \mathcal{T}_{\mathrm{mv}}(\mathbf{K}) \cdot \sqrt{\kappa_{\mathrm{scale}}} \cdot \log^2 \frac{d\kappa_{\mathrm{scale}}}{\delta} \right).$$

*Proof.* The proof of Lemma 6 implies it suffices to compute a 0.2-multiplicative approximation to the largest eigenvalue of $\mathbf{P}$, a degree-$\Delta = O(\sqrt{\kappa_{\mathrm{scale}}} \log d)$ polynomial in $\mathbf{M}$. Moreover, letting

$\Delta' = O(\log \frac{d}{\delta})$ be the degree given by Fact 3 with $\delta \leftarrow \frac{\delta}{3}$, the statement of the algorithm in Fact 3 shows it suffices to compute for a uniformly random unit vector $u$,

$$\left\|\mathbf{P}^{\Delta} u\right\|_2 \text{ and } \left\|\mathbf{P}^{\Delta+1} u\right\|_2 \text{ to multiplicative accuracy } \frac{1}{30}.$$

We demonstrate how to compute $\left\|\mathbf{P}^{\Delta} u\right\|_2$ to this multiplicative accuracy with probability at least $1 - \frac{\delta}{3}$; the computation of $\left\|\mathbf{P}^{\Delta+1} u\right\|_2$ is identical, and the failure probability follows from a union bound over these three random events. Since $\mathbf{P}^{\Delta}$ is a degree-$O(\Delta\Delta') = O(\sqrt{\kappa_{\text{scale}}} \log d \log \frac{d}{\delta})$ polynomial in $\mathbf{AWA}$, the conclusion follows from Lemma 19. $\qquad\square$

**Estimating inner products with a negative exponential implicitly.** We next discuss implicit implementation of Line 6. In particular, we give variants of Lemmas 2 and 3 which are tolerant to implicit approximate access of matrix-vector products.

**Lemma 22.** *Given $\delta \in (0, 1)$, constant $\epsilon > 0$, $\mathbf{K} \in \mathbb{S}_{\succ \mathbf{0}}^d$ such that $\mathbf{A} := \mathbf{K}^{\frac{1}{2}}$ and $\kappa(\mathbf{K}) \leq \kappa_{\text{scale}}$, and diagonal $\mathbf{W} \in \mathbb{S}_{\succeq \mathbf{0}}^d$ such that $\mathbf{M} := \mathbf{AWA} \preceq O(\kappa_{\text{scale}} \log d)\mathbf{I}$ and $\lambda_{\min}(\mathbf{M}) = O(\log d)$, we can compute an $\epsilon$-multiplicative approximation to $\operatorname{Tr} \exp(-\mathbf{M})$ with probability $\geq 1 - \delta$ in time*

$$O\left(\mathcal{T}_{\text{mv}}(\mathbf{K}) \cdot \sqrt{\kappa_{\text{scale}}} \cdot \log^2 \frac{d\kappa_{\text{scale}}}{\delta}\right).$$

*Proof.* The proof of Lemma 2 shows it suffices to compute $k = O(\log \frac{d}{\delta})$ times, an $\frac{\epsilon}{3} \exp(-R)$-additive approximation to $u^\top \mathbf{P} u$ where $R = O(\log d)$, for uniformly random unit $u$ and $\mathbf{P}$, a degree-$\Delta = O(\sqrt{\kappa_{\text{scale}}} \log d)$-polynomial in $\mathbf{M}$ with $\|\mathbf{P}\|_2 \leq \|\exp(-\mathbf{M})\|_2 + \frac{\epsilon}{3} \exp(-R) \leq \frac{4}{3} \exp(-R)$. Applying Lemma 19 with $\epsilon \leftarrow \frac{\epsilon}{4}$ to compute $w$, an approximation to $\mathbf{P}u$, the approximation follows:

$$|\langle w, u \rangle - \langle \mathbf{P}u, u \rangle| \leq \|w - \mathbf{P}u\|_2 \leq \frac{\epsilon}{4} \|\mathbf{P}\|_2 \leq \frac{\epsilon}{3} \exp(-R).$$

The runtime follows from the cost of applying Lemma 19 to all $k$ random unit vectors. $\qquad\square$

**Lemma 23.** *Given $\delta \in (0, 1)$, constant $\epsilon > 0$, $\mathbf{K} \in \mathbb{S}_{\succ \mathbf{0}}^d$ such that $\mathbf{A} := \mathbf{K}^{\frac{1}{2}}$ and $\kappa(\mathbf{K}) \leq \kappa_{\text{scale}}$, and diagonal $\mathbf{W} \in \mathbb{S}_{\succeq \mathbf{0}}^d$ such that $\mathbf{M} := \mathbf{AWA} \preceq O(\kappa_{\text{scale}} \log d)\mathbf{I}$, we can compute $(\epsilon, O(\frac{1}{\kappa_{\text{scale}}d}))$-approximations to all*

$$\left\{\left\langle a_i a_i^\top, \exp(-\mathbf{M}) \right\rangle\right\}_{i \in [d]},$$

*with probability $\geq 1 - \delta$ in time*

$$O\left(\mathcal{T}_{\text{mv}}(\mathbf{K}) \cdot \sqrt{\kappa_{\text{scale}}} \cdot \log^2 \frac{d\kappa_{\text{scale}}}{\delta}\right).$$

*Proof.* The proof of Lemma 3 implies it suffices to compute $k = O(\log \frac{d}{\delta})$ times, for each $i \in [d]$, the quantity $\langle u, \mathbf{P} a_i \rangle$ to multiplicative error $\frac{\epsilon}{2}$, for uniformly random unit vector $u$ and $\mathbf{P}$, a degree-$\Delta = O(\sqrt{\kappa_{\text{scale}}} \log(\kappa_{\text{scale}}d))$-polynomial in $\mathbf{M}$. Next, note that since $a_i = \mathbf{A}e_i$ and $\mathbf{P} = \mathbf{ANA}$ for $\mathbf{N}$ an explicit degree-$O(\Delta)$ polynomial in $\mathbf{K}$ and $\mathbf{W}$, we have $\langle u, \mathbf{P} a_i \rangle = u^\top \mathbf{A} (\mathbf{NK}e_i)$. We can approximate this by some $\langle w, \mathbf{NK}e_i \rangle$ via Lemma 20 to the desired accuracy. The runtime comes from applying Lemma 20 $k$ times, multiplying each of the resulting vectors $w$ by $\mathbf{KN}$ and stacking them to form a $k \times d$ matrix $\widetilde{\mathbf{Q}}$, and then computing all $\|\widetilde{\mathbf{Q}}e_i\|_2$ for $i \in [d]$. $\qquad\square$

**Implementing a packing oracle implicitly.** Finally, we discuss implementation of Line 7 of Algorithm 1. The requirement of Line 7 is a multiplicative approximation (and a witnessing reweighting) to the optimization problem

$$\max_{\substack{\sum_{i\in[d]} w_i \mathbf{M}_i \preceq \mathbf{I} \\ w\in\mathbb{R}^d_{\geq 0}}} v^\top w.$$

Here, $v$ is explicitly given by an implementation of Line 6 of the algorithm, but we do not have $\{\mathbf{M}_i\}_{i\in[d]}$ explicitly. To implement this step implicitly, we recall the approximation requirements of the solver of Proposition 1, as stated in [JLT20]. We remark that the approximation tolerance is stated for the decision problem tester of [JLT20] (Proposition 5); once the tester is implicitly implemented, the same reduction as described in Appendix A yields an analog to Proposition 1.

**Corollary 4** (Approximation tolerance of Proposition 1, Theorem 5, [JLT20]). *Let $\epsilon > 0$ be a fixed constant. The runtime of Proposition 1 is due to $T = O(\log(\frac{d}{\delta})\log d \cdot \log\log \frac{\mathrm{OPT}_+}{\mathrm{OPT}_-})$ iterations, each of which requires $O(1)$ vector operations and $O(\epsilon)$-multiplicative approximations to*

$$\mathrm{Tr}\left(\mathbf{M}^p\right), \ \left\{\left\langle \mathbf{A}_i, \mathbf{M}^{p-1}\right\rangle\right\}_{i\in[d]} \ \text{for } \mathbf{M} := \sum_{i\in[d]} w_i \mathbf{M}_i \text{ for an explicitly given } w \in \mathbb{R}^d_{\geq 0}, \qquad (26)$$

*where $p = O(\log d) \in \mathbb{N}$ is odd, and $S\mathbf{I} \preceq \mathbf{M} \preceq R\mathbf{I}$, for $R = O(\log d)$ and $S = \mathrm{poly}(\frac{1}{nd}, \kappa((\sum_{i\in[n]}\mathbf{M}_i))^{-1})$.*

We remark that the lower bound $S$ comes from the fact that the initial matrix of the [JLT20] solver is a bounded scaling of $\sum_{i\in[n]}\mathbf{M}_i$, and the iterate matrices are monotone in Loewner order. We now demonstrate how to use Lemmas 19 and 20 to approximate all quantities in (26). Throughout the following discussion, we specialize to the case where each $\mathbf{M}_i = a_i a_i^\top$, so $\mathbf{M}$ in (26) will always have the form $\mathbf{M} = \mathbf{AWA}$ for diagonal $\mathbf{W} \in \mathbb{S}^d_{\succeq \mathbf{0}}$, and $S = \mathrm{poly}((d\kappa_{\mathrm{scale}})^{-1})$.

**Lemma 24.** *Given $\delta \in (0,1)$, constant $\epsilon > 0$, $\mathbf{K} \in \mathbb{S}^d_{\succ \mathbf{0}}$ such that $\mathbf{A} := \mathbf{K}^{\frac{1}{2}}$ and $\kappa(\mathbf{K}) \leq \kappa_{\mathrm{scale}}$, and diagonal $\mathbf{W} \in \mathbb{S}^d_{\succeq \mathbf{0}}$ such that $S\mathbf{I} \preceq \mathbf{M} := \mathbf{AWA} \preceq O(\log d)\mathbf{I}$ where $S = \mathrm{poly}((d\kappa_{\mathrm{scale}})^{-1})$, we can compute an $\epsilon$-multiplicative approximation to $\mathrm{Tr}(\mathbf{M}^p)$ for integer $p$ in time*

$$O\left(\mathcal{T}_{\mathrm{mv}}(\mathbf{K}) \cdot \left(p + \sqrt{\kappa_{\mathrm{scale}}}\log\frac{d\kappa_{\mathrm{scale}}}{\delta}\right) \cdot \log\frac{d}{\delta}\right).$$

*Proof.* As in Lemma 22, it suffices to compute $k = O(\log\frac{d}{\delta})$ times, an $\frac{\epsilon}{N}S^p$-additive approximation to $u^\top \mathbf{M}^p u$, for uniformly random unit vector $u$ and $N = \mathrm{poly}(d, \delta^{-1})$. By applying Lemma 19 with accuracy $\epsilon' \leftarrow \frac{\epsilon S^p}{NR^p}$ to obtain $w$, an approximation to $\mathbf{M}^p u$, we have the desired

$$|\langle u, \mathbf{M}^p u\rangle - \langle u, w\rangle| \leq \|\mathbf{M}^p u - w\|_2 \leq \epsilon' \|\mathbf{M}^p u\|_2 \leq \epsilon' R^p \leq \frac{\epsilon}{N}S^p.$$

The runtime follows from $k$ applications of Lemma 19 to the specified accuracy level. $\qquad\square$

**Lemma 25.** *Given $\delta \in (0,1)$, constant $\epsilon > 0$, $\mathbf{K} \in \mathbb{S}^d_{\succ \mathbf{0}}$ such that $\mathbf{A} := \mathbf{K}^{\frac{1}{2}}$ and $\kappa(\mathbf{K}) \leq \kappa_{\mathrm{scale}}$, and diagonal $\mathbf{W} \in \mathbb{S}^d_{\succeq \mathbf{0}}$ such that $\mathbf{M} := \mathbf{AWA} \preceq O(\log d)\mathbf{I}$, we can compute an $\epsilon$-multiplicative approximation to all*

$$\left\{\left\langle a_i a_i^\top, \mathbf{M}^{p-1}\right\rangle\right\}_{i\in[d]} \ \text{where } \{a_i\}_{i\in[d]} \text{ are rows of } \mathbf{A},$$

*where $p$ is an odd integer, with probability $\geq 1 - \delta$ in time*

$$O\left(\mathcal{T}_{\mathrm{mv}}(\mathbf{K}) \cdot \left(p + \sqrt{\kappa_{\mathrm{scale}}} \log \frac{d\kappa_{\mathrm{scale}}}{\delta}\right) \cdot \log \frac{d}{\delta}\right).$$

*Proof.* First, observe that for all $i \in [d]$ it is the case that

$$\left\langle a_i a_i^\top, \mathbf{M}^{p-1} \right\rangle = (\mathbf{A} e_i)^\top \mathbf{M}^{p-1} (\mathbf{A} e_i) \geq S^{p-1} \|\mathbf{A}\|_2^2 \kappa_{\mathrm{scale}}^{-2}.$$

Letting $r = \frac{1}{2}(p-1)$ and following Lemma 23 and the above calculation, it suffices to show how to compute $k = O(\log \frac{d}{\delta})$ times, for each $i \in [d]$, the quantity $\langle u, \mathbf{M}^r a_i \rangle$ to multiplicative error $\frac{\epsilon}{2}$, for uniformly random unit vector $u$ and $N = \mathrm{poly}(d, \delta^{-1})$. As in Lemma 23, each such inner product is $u^\top \mathbf{A}(\mathbf{N}\mathbf{K} e_i)$ for $\mathbf{N}$ an explicit degree-$O(p)$ polynomial in $\mathbf{K}$ and $\mathbf{W}$. The runtime follows from applying Lemma 20 $k$ times and following the runtime analysis of Lemma 23. $\qquad\square$

**Putting it all together.** Finally, we state our main result of this section, regarding rescaling well-conditioned matrices, by combining the pieces we have developed.

**Corollary 5.** *Given $\delta \in (0,1)$, constant $\epsilon > 0$, $\mathbf{K} \in \mathbb{S}_{\succ \mathbf{0}}^d$ such that $\kappa(\mathbf{K}) \leq 3\kappa^\star$, and such that $\kappa_o^\star(\mathbf{K}) = \kappa^\star$, we can compute diagonal $\mathbf{W} \in \mathbb{S}_{\succeq \mathbf{0}}^d$ such that $\kappa(\mathbf{W}^{\frac{1}{2}} \mathbf{K} \mathbf{W}^{\frac{1}{2}}) \leq (1+\epsilon)\kappa^\star$ with probability $\geq 1 - \delta$ in time*

$$O\left(\mathcal{T}_{\mathrm{mv}}(\mathbf{K}) \cdot (\kappa^\star)^{1.5} \cdot \log^6 \left(\frac{d\kappa^\star}{\delta}\right)\right).$$

*Proof.* The proof is essentially identical to the proof of Theorem 13 by way of Theorem 8. In particular, we parameterize Theorem 8 with $\mathbf{M}_i = a_i a_i^\top$ where $\mathbf{A} = \mathbf{K}^{\frac{1}{2}}$ is the positive definite square root of $\mathbf{K}$ with rows $\{a_i\}_{i \in [d]}$. Then, running Algorithm 1 with an incremental search for the optimal $\kappa^\star$ yields an overhead of $\widetilde{O}(\kappa^\star \log(d\kappa^\star))$. The cost of each iteration of Algorithm 1 follows by combining Lemmas 21, 22, 23, 24, 25, and Corollary 4. $\qquad\square$

### 5.2.3 Homotopy method

In this section, we use Corollary 5, in conjunction with a homotopy method similar to that of Section 3.2, to obtain our overall algorithm for outer scaling. We state here three simple helper lemmas which follow almost identically from corresponding helper lemmas in Section 3.2; we include proofs of these statements in Appendix A for completeness.

**Lemma 26.** *For any matrix $\mathbf{K} \in \mathbb{S}_{\succ \mathbf{0}}^d$ and $\lambda \geq 0$, $\kappa_o^\star(\mathbf{K} + \lambda \mathbf{I}) \leq \kappa_o^\star(\mathbf{K})$.*

**Lemma 27.** *Let $\mathbf{K} \in \mathbb{S}_{\succ \mathbf{0}}^d$. Then, for $\lambda \geq \frac{1}{\epsilon} \lambda_{\max}(\mathbf{K})$, $\kappa(\mathbf{K} + \lambda \mathbf{I}) \leq 1 + \epsilon$. Moreover, given a diagonal $\mathbf{W} \in \mathbb{S}_{\succeq \mathbf{0}}^d$ such that $\kappa(\mathbf{W}^{\frac{1}{2}}(\mathbf{K} + \lambda \mathbf{I})\mathbf{W}^{\frac{1}{2}}) \leq \kappa_{\mathrm{scale}}$ for $0 \leq \lambda \leq \frac{\epsilon \lambda_{\min}(\mathbf{K})}{1+\epsilon}$, $\kappa(\mathbf{W}^{\frac{1}{2}} \mathbf{K} \mathbf{W}^{\frac{1}{2}}) \leq (1+\epsilon)\kappa_{\mathrm{scale}}$.*

**Lemma 28.** *Let $\mathbf{K} \in \mathbb{S}_{\succ \mathbf{0}}^d$, and let $\mathbf{W} \in \mathbb{S}_{\succeq \mathbf{0}}^d$ be diagonal. Then for any $\lambda > 0$,*

$$\kappa\left(\mathbf{W}^{\frac{1}{2}} (\mathbf{K} + \lambda \mathbf{I}) \mathbf{W}^{\frac{1}{2}}\right) \leq 2\kappa\left(\mathbf{W}^{\frac{1}{2}} \left(\mathbf{K} + \frac{\lambda}{2}\mathbf{I}\right) \mathbf{W}^{\frac{1}{2}}\right).$$

**Theorem 14.** *Let $\epsilon > 0$ be a fixed constant.*[13] *There is an algorithm, which given full-rank $\mathbf{K} \in \mathbb{S}_{\succ 0}^d$ computes $w \in \mathbb{R}_{\geq 0}^d$ such that $\kappa(\mathbf{W}^{\frac{1}{2}} \mathbf{K} \mathbf{W}^{\frac{1}{2}}) \leq (1 + \epsilon)\kappa_o^\star(\mathbf{K})$ with probability $\geq 1 - \delta$ in time*

$$O\left(\mathcal{T}_{\mathrm{mv}}(\mathbf{K}) \cdot (\kappa_o^\star(\mathbf{K}))^{1.5} \cdot \log^8\left(\frac{d\kappa_o^\star(\mathbf{K})}{\delta}\right)\right).$$

*Proof.* We will assume we know the correct value of $\kappa_o^\star(\mathbf{K})$ up to a $1 + O(\epsilon)$ factor throughout this proof for simplicity, and call this estimate $\kappa^\star$. This will add an overall multiplicative overhead of $O(1)$ by using an incremental search as in Theorem 8. We will also assume that $\kappa(\mathbf{K}) = O((\kappa^\star)^2)$ by first applying the Jacobi preconditioner; see Appendix C for a proof.

Our algorithm follows the framework of Section 3.2 and runs in phases indexed by $k$ for $0 \leq k \leq K$ for some $K$, each computing a scaling of $\mathbf{K} + \lambda_k \mathbf{I}$ with condition number $(1 + \epsilon)\kappa^\star$; note that a scaling with condition number $\kappa^\star$ is always feasible for any $\lambda_k \geq 0$ by Lemma 26. We will define $\lambda_0 = \frac{1}{\epsilon}V$ where $V$ is a constant-factor overestimate of $\lambda_{\max}(\mathbf{K})$, which can be obtained by Fact 3 without dominating the runtime. We will then set

$$\lambda_k = \frac{\lambda_0}{2^k}, \ K = O\left(\log \kappa^\star\right).$$

Lemma 27 shows that we have a trivial scaling attaing condition number $(1 + \epsilon)\kappa^\star$ for $\mathbf{K} + \lambda_0 \mathbf{I}$, and that if we can compute rescalings for all $\lambda_k$ where $1 \leq k \leq K$, then the last rescaling is also a $(1 + \epsilon)\kappa^\star$-conditioned rescaling for $\mathbf{K}$ up to adjusting $\epsilon$ by a constant.

Finally, we show how to implement each phase of the algorithm, given access to the reweighting from the previous phase. Note that Lemma 28 shows that the reweighting $\mathbf{W}$ computed in phase $k$ yields a rescaling $\mathbf{W}^{\frac{1}{2}}(\mathbf{K} + \lambda_{k+1}\mathbf{I})\mathbf{W}^{\frac{1}{2}}$ which is $3\kappa^\star$-conditioned. By running the algorithm of Corollary 5 on $\mathbf{K} \leftarrow \mathbf{W}^{\frac{1}{2}}(\mathbf{K} + \lambda_{k+1}\mathbf{I})\mathbf{W}^{\frac{1}{2}}$, we compute the desired reweighting for phase $k+1$. The final runtime loses one logarithmic factor over Corollary 5 due to running for $K$ phases. $\square$

## 6 Statistical applications of diagonal scaling

In this section, we give a number of applications of our rescaling methods to problems in statistical settings (i.e., linear system solving or statistical regression) where reducing conditioning measures are effective. We begin by discussing connections between diagonal preconditioning and a semi-random noise model for linear systems in Section 6.1. We then apply rescaling methods to reduce risk bounds for statistical models of linear regression in Section 6.2.

### 6.1 Semi-random linear systems

Consider the following semi-random noise model for solving an overdetermined, consistent linear system $\mathbf{A}x_{\mathrm{true}} = b$ where $\mathbf{A} \in \mathbb{R}^{n \times d}$ for $n \geq d$.

**Definition 1** (Semi-random linear systems)**.** *In the* semi-random noise model *for linear systems, a matrix $\mathbf{A}_g \in \mathbb{R}^{m \times d}$ with $\kappa(\mathbf{A}_g^\top \mathbf{A}_g) = \kappa_g$, $m \geq d$ is "planted" as a subset of rows of a larger matrix $\mathbf{A} \in \mathbb{R}^{n \times d}$. We observe the vector $b = \mathbf{A}x_{\mathrm{true}}$ for some $x_{\mathrm{true}} \in \mathbb{R}^d$ we wish to recover.*

We remark that we call the model in Definition 1 "semi-random" because of the following motivating example: the rows $\mathbf{A}_g$ are feature vectors drawn from some "nice" (e.g., well-conditioned) distribution, and the dataset is contaminated by an adversary supplying additional data (a priori indistinguishable from the "nice" data), aiming to hinder conditioning of the resulting system.

---

[13] We do not focus on the $\epsilon$ dependence and instead take it to be constant since, in applications involving solving linear systems, there is little advantage to obtaining better than a two factor approximation (i.e., setting $\epsilon = 1$).

Interestingly, Definition 1 demonstrates in some sense a shortcoming of existing linear system solvers: their brittleness to *additional, consistent information.* In particular, $\kappa(\mathbf{A}^\top \mathbf{A})$ can be arbitrarily larger than $\kappa_g$. However, if we were given the indices of the subset of rows $\mathbf{A}_g$, we could instead solve the linear system $b_g = \mathbf{A}_g x_{\text{true}}$ with iteration count dependent on the condition number of $\mathbf{A}_g$. Counterintuitively, by giving additional rows, the adversary can arbitrarily increase the condition number of the linear system, hindering the runtime of conditioning-dependent solvers.

The inner rescaling algorithms we develop in Section 5 are well-suited for robustifying linear system solvers to the type of adversary in Definition 1. In particular, note that

$$\kappa_i^\star(\mathbf{A}) \leq \kappa\left(\mathbf{A}^\top \mathbf{W}_g \mathbf{A}\right) = \kappa\left(\mathbf{A}_g^\top \mathbf{A}_g\right) = \kappa_g,$$

where $\mathbf{W}_g$ is the diagonal matrix which is the 0-1 indicator of rows of $\mathbf{A}_g$. Our solvers for reweightings approximating $\kappa_i^\star$ can thus be seen as trading off the *sparsity* of $\mathbf{A}_g$ for the potential of "mixing rows" to attain a runtime dependence on $\kappa_i^\star(\mathbf{A}) \leq \kappa_g$. In particular, our resulting runtimes scale with $\text{nnz}(\mathbf{A})$ instead of $\text{nnz}(\mathbf{A}_g)$, but also depend on $\kappa_i^\star(\mathbf{A})$ rather than $\kappa_g$.

We remark that the other solvers we develop are also useful in robustifying against variations on the adversary in Definition 1. For instance, the adversary could instead aim to increase $\tau(\mathbf{A}^\top \mathbf{A})$, or give additional irrelevant features (i.e., columns of $\mathbf{A}$) such that only some subset of coordinates $x_g$ are important to recover. For brevity, we focus on the model in Definition 1 in this work.

## 6.2 Statistical linear regression

The second application we give is in solving noisy variants of the linear system setting of Definition 1. In particular, we consider statistical regression problems with various generative models.

**Definition 2** (Statistical linear regression). *Given full rank* $\mathbf{A} \in \mathbb{R}^{n \times d}$ *and* $b \in \mathbb{R}^d$ *produced via*

$$b = \mathbf{A}x_{\text{true}} + \xi, \ \xi \sim \mathcal{N}(0, \boldsymbol{\Sigma}), \tag{27}$$

*where we wish to recover unknown* $x_{\text{true}} \in \mathbb{R}^d$, *return* $x$ *so that (where expectations are taken over the randomness of* $\xi$) *the* risk *(mean-squared error)* $\mathbb{E}[\|x - x_{\text{true}}\|_2^2]$ *is small.*

In this section, we define a variety of generative models (i.e., specifying a covariance matrix $\boldsymbol{\Sigma}$ of the noise) for the problem in Definition 2. For each of the generative models, applying our rescaling procedures will yield computational gains, improved risk bounds, or both. We give statistical and computational results for statistical linear regression in both the *homoskedastic* and *heteroskedastic* settings. In particular, when $\boldsymbol{\Sigma} = \sigma^2 \mathbf{I}$ (i.e., the noise for every data point has the same variance), this is the well-studied homoskedastic setting pervasive in stastical modeling. When $\boldsymbol{\Sigma}$ varies with the data $\mathbf{A}$, the model is called heteroskedastic (cf. [Gre90]).

In most cases, we do not directly give guarantees on exact mean squared errors via our preprocessing, but rather certify (possibly loose) upper bound surrogates. We leave direct certification of conditioning and risk simultaneously without a surrogate bound as an interesting future direction.

### 6.2.1 Heteroskedastic statistical guarantees

We specify two types of heteroskedastic generative models (i.e., defining the covariance $\boldsymbol{\Sigma}$ in (27)), and analyze the effect of rescaling a regression data matrix on reducing risk.

**Noisy features.** Consider the setting where the covariance in (27) has the form $\boldsymbol{\Sigma} = \mathbf{A}\boldsymbol{\Sigma}'\mathbf{A}^\top$, for matrix $\boldsymbol{\Sigma}' \in \mathbb{S}_{\succeq \mathbf{0}}^d$. Under this assumption, we can rewrite (27) as $b = \mathbf{A}(x_{\text{true}} + \xi')$, where $\xi' \sim \mathcal{N}(0, \boldsymbol{\Sigma}')$. Intuitively, this corresponds to exact measurements through $\mathbf{A}$, under noisy features $x_{\text{true}} + \xi'$. As in this case $b \in \text{Im}(\mathbf{A})$ always, regression is equivalent to linear system solving, and thus directly solving any reweighted linear system $\mathbf{W}^{\frac{1}{2}}\mathbf{A}x^* = \mathbf{W}^{\frac{1}{2}}b$ will yield $x^* = x_{\text{true}} + \xi'$.

We thus directly obtain improved computational guarantees by computing a reweighting $\mathbf{W}^{\frac{1}{2}}$ with $\kappa(\mathbf{A}^\top \mathbf{W} \mathbf{A}) = O(\kappa_i^\star(\mathbf{A}))$. Moreover, we note that the risk (Definition 2) of the linear system solution $x^*$ is independent of the reweighting:

$$\mathbb{E}\left[\|x^* - x_{\text{true}}\|_2^2\right] = \mathbb{E}\left[\|\xi'\|_2^2\right] = \text{Tr}\left(\mathbf{\Sigma}'\right).$$

Hence, computational gains from reweighting the system are without statistical loss in the risk.

**Row norm noise.** Consider the setting where the covariance in (27) has the form

$$\mathbf{\Sigma} = \sigma^2 \mathbf{diag}\left(\left\{\|a_i\|_2^2\right\}_{i\in[n]}\right). \tag{28}$$

Intuitively, this corresponds to the setting where noise is independent across examples and the size of the noise scales linearly with the squared row norm. We first recall a standard characterization of the regression minimizer.

**Fact 5** (Regression minimizer). *Let the regression problem $\|\mathbf{A}x - b\|_2^2$ have minimizer $x^\star$, and suppose that $\mathbf{A}^\top \mathbf{A}$ is invertible. Then,*

$$x^\star = \left(\mathbf{A}^\top \mathbf{A}\right)^{-1} \mathbf{A}^\top b.$$

Using Fact 5, we directly prove the following upper bound surrogate holds on the risk under the model (27), (28) for the solution to any reweighted regression problem.

**Lemma 29.** *Under the generative model (27), (28), letting $\mathbf{W} \in \mathbb{S}_{\succeq \mathbf{0}}^n$ be a diagonal matrix and*

$$x_w^\star := \text{argmin}_x \left\{\left\|\mathbf{W}^{\frac{1}{2}}\left(\mathbf{A}x - b\right)\right\|_2^2\right\},$$

*we have*

$$\mathbb{E}\left[\|x_w^\star - x_{\text{true}}\|_2^2\right] \le \sigma^2 \frac{\text{Tr}\left(\mathbf{A}^\top \mathbf{W} \mathbf{A}\right)}{\lambda_{\min}\left(\mathbf{A}^\top \mathbf{W} \mathbf{A}\right)}.$$

*Proof.* By applying Fact 5, we have that

$$x_w^\star = \left(\mathbf{A}^\top \mathbf{W} \mathbf{A}\right)^{-1} \mathbf{A}^\top \mathbf{W}\left(\mathbf{A}x_{\text{true}} + \xi\right) = x_{\text{true}} + \left(\mathbf{A}^\top \mathbf{W} \mathbf{A}\right)^{-1} \mathbf{A}^\top \mathbf{W}\xi.$$

Thus, we have the sequence of derivations

$$
\begin{aligned}
\mathbb{E}\left[\|x_w^\star - x_{\text{true}}\|_{\mathbf{A}^\top \mathbf{W} \mathbf{A}}^2\right] &= \mathbb{E}\left[\left\|\left(\mathbf{A}^\top \mathbf{W} \mathbf{A}\right)^{-1} \mathbf{A}^\top \mathbf{W}\xi\right\|_{\mathbf{A}^\top \mathbf{W} \mathbf{A}}^2\right] \\
&= \mathbb{E}\left[\left\langle \mathbf{W}^{\frac{1}{2}}\xi\xi^\top \mathbf{W}^{\frac{1}{2}}, \mathbf{W}^{\frac{1}{2}}\mathbf{A}\left(\mathbf{A}^\top \mathbf{W} \mathbf{A}\right)^{-1}\mathbf{A}^\top \mathbf{W}^{\frac{1}{2}}\right\rangle\right] \\
&= \sigma^2 \left\langle \mathbf{diag}\left(\left\{w_i \|a_i\|_2^2\right\}\right), \mathbf{W}^{\frac{1}{2}}\mathbf{A}\left(\mathbf{A}^\top \mathbf{W} \mathbf{A}\right)^{-1}\mathbf{A}^\top \mathbf{W}^{\frac{1}{2}}\right\rangle \\
&\le \sigma^2 \text{Tr}\left(\mathbf{A}^\top \mathbf{W} \mathbf{A}\right).
\end{aligned}
\tag{29}
$$

The last inequality used the $\ell_1$-$\ell_\infty$ matrix Hölder inequality and that $\mathbf{W}^{\frac{1}{2}}\mathbf{A}\left(\mathbf{A}^\top \mathbf{W} \mathbf{A}\right)^{-1}\mathbf{A}^\top \mathbf{W}^{\frac{1}{2}}$ is

a projection matrix, so $\|\mathbf{W}^{\frac{1}{2}}\mathbf{A}\left(\mathbf{A}^\top\mathbf{W}\mathbf{A}\right)^{-1}\mathbf{A}^\top\mathbf{W}^{\frac{1}{2}}\|_\infty = 1$. Lower bounding the squared $\mathbf{A}^\top\mathbf{W}\mathbf{A}$ norm by a $\lambda_{\min}(\mathbf{A}^\top\mathbf{W}\mathbf{A})$ multiple of the squared Euclidean norm yields the conclusion. $\qquad\square$

We remark that the analysis in Lemma 29 of the surrogate upper bound we provide was loose in two places: the application of Hölder and the norm conversion. Lemma 29 shows that the risk under the generative model (28) can be upper bounded by a quantity proportional to $\tau(\mathbf{A}^\top\mathbf{W}\mathbf{A})$, the average conditioning of the reweighted matrix.

Directly applying Lemma 29, our risk bounds improve with the conditioning of the reweighted system. Hence, our scaling procedures improve both the computational and statistical guarantees of regression under this generative model, albeit only helping the latter through an upper bound.

### 6.2.2 Homoskedastic statistical guarantees

In this section, we work under the homoskedastic generative model assumption. In particular, throughout the covariance matrix in (27) will be a multiple of the identity:

$$\mathbf{\Sigma} = \sigma^2\mathbf{I}. \tag{30}$$

We begin by providing a risk upper bound under the model (27), (30).

**Lemma 30.** *Under the generative model* (27), (30), *let* $x^\star := \operatorname{argmin}_x\{\|\mathbf{A}x - b\|_2^2\}$. *Then,*

$$\mathbb{E}\left[\|x^\star - x_{\text{true}}\|_{\mathbf{A}^\top\mathbf{A}}^2\right] = \sigma^2 d \implies \mathbb{E}\left[\|x^* - x_{\text{true}}\|_2^2\right] \le \frac{\sigma^2 d}{\lambda_{\min}(\mathbf{A}^\top\mathbf{A})}. \tag{31}$$

*Proof.* Using Fact 5, we compute

$$x^\star - x_{\text{true}} = \left(\mathbf{A}^\top\mathbf{A}\right)^{-1}\mathbf{A}^\top b - x_{\text{true}}$$
$$= \left(\mathbf{A}^\top\mathbf{A}\right)^{-1}\mathbf{A}^\top\left(\mathbf{A}x_{\text{true}} + \xi\right) - x_{\text{true}} = \left(\mathbf{A}^\top\mathbf{A}\right)^{-1}\mathbf{A}^\top\xi.$$

Therefore via directly expanding, and using linearity of expectation,

$$\mathbb{E}\left[\|x^* - x_{\text{true}}\|_{\mathbf{A}^\top\mathbf{A}}^2\right] = \mathbb{E}\left[\left\|\mathbf{A}\left(\mathbf{A}^\top\mathbf{A}\right)^{-1}\mathbf{A}^\top\xi\right\|_2^2\right]$$
$$= \mathbb{E}\left[\left\langle\xi\xi^\top, \mathbf{A}\left(\mathbf{A}^\top\mathbf{A}\right)^{-1}\mathbf{A}^\top\right\rangle\right] = \sigma^2\left(\mathbf{A}\left(\mathbf{A}^\top\mathbf{A}\right)^{-1}\mathbf{A}^\top\right) = \sigma^2 d.$$

The final implication follows from $\lambda_{\min}(\mathbf{A}^\top\mathbf{A})\|x^* - x_{\text{true}}\|_2^2 \le \|x^* - x_{\text{true}}\|_{\mathbf{A}^\top\mathbf{A}}^2$. $\qquad\square$

Lemma 30 shows that in regards to our upper bound (which is loose in the norm conversion at the end), the notion of adversarial semi-random noise is at odds in the computational and statistical senses. Namely, given additional rows of the matrix $\mathbf{A}$, the bound (31) can only improve, since $\lambda_{\min}$ is monotonically increasing as rows are added. To address this, we give guarantees about recovering reweightings which match the best possible upper bound anywhere along the "computational-statistical tradeoff curve." We begin by providing a weighted analog of Lemma 30.

**Lemma 31.** *Under the generative model* (27), (30), *letting* $\mathbf{W} \in \mathbb{S}_{\succeq\mathbf{0}}^n$ *be a diagonal matrix and*

$$x_w^\star := \operatorname{argmin}_x\left\{\left\|\mathbf{W}^{\frac{1}{2}}\left(\mathbf{A}x - b\right)\right\|_2^2\right\},$$

*we have*

$$\mathbb{E}\left[\|x_w^\star - x_{\text{true}}\|_2^2\right] \leq \sigma^2 d \cdot \frac{\|w\|_\infty}{\lambda_{\min}\left(\mathbf{A}^\top\mathbf{W}\mathbf{A}\right)}. \tag{32}$$

*Proof.* By following the derivations (29) (and recalling the definition of $x_w^\star$),

$$\begin{aligned}
\mathbb{E}\left[\|x_w^\star - x_{\text{true}}\|_{\mathbf{A}^\top\mathbf{W}\mathbf{A}}^2\right] &= \mathbb{E}\left[\left\langle \xi\xi^\top, \mathbf{W}\mathbf{A}\left(\mathbf{A}^\top\mathbf{W}\mathbf{A}\right)^{-1}\mathbf{A}^\top\mathbf{W}\right\rangle\right] \\
&= \sigma^2\mathrm{Tr}\left(\mathbf{W}\mathbf{A}\left(\mathbf{A}^\top\mathbf{W}\mathbf{A}\right)^{-1}\mathbf{A}^\top\mathbf{W}\right).
\end{aligned} \tag{33}$$

Furthermore, by $\mathbf{W} \preceq \|w\|_\infty \mathbf{I}$ we have $\mathbf{A}^\top\mathbf{W}^2\mathbf{A} \preceq \|w\|_\infty \mathbf{A}^\top\mathbf{W}\mathbf{A}$. Thus,

$$\mathrm{Tr}\left(\mathbf{W}\mathbf{A}\left(\mathbf{A}^\top\mathbf{W}\mathbf{A}\right)^{-1}\mathbf{A}^\top\mathbf{W}\right) = \left\langle \mathbf{A}^\top\mathbf{W}^2\mathbf{A}, \left(\mathbf{A}^\top\mathbf{W}\mathbf{A}\right)^{-1}\right\rangle \leq \|w\|_\infty \mathrm{Tr}(\mathbf{I}) = d\|w\|_\infty.$$

Using this bound in (33) and converting to Euclidean norm risk yields the conclusion. $\qquad\square$

Lemma 31 gives a quantitative version of a computational-statistical tradeoff curve. Specifically, we give guarantees which target the best possible condition number of a 0-1 reweighting, subject to a given level of $\lambda_{\min}(\mathbf{A}^\top\mathbf{W}\mathbf{A})$. In the following discussion we assume there exists $\mathbf{A}_g \subseteq \mathbf{A}$, a subset of rows, satisfying (for known $\kappa_g$, $\nu_g$, and sufficiently small constant $\epsilon \in (0,1)$)

$$\kappa_g \leq \kappa\left(\mathbf{A}_g^\top\mathbf{A}_g\right) \leq (1+\epsilon)\kappa_g, \quad \frac{1}{\lambda_{\min}\left(\mathbf{A}_g^\top\mathbf{A}_g\right)} \leq \nu_g. \tag{34}$$

Our key observation is that we can use existence of a row subset satisfying (34), combined with a slight modification of Algorithm 1, to find a reweighting $w$ such that

$$\kappa\left(\mathbf{A}^\top\mathbf{W}\mathbf{A}\right) = O\left(\kappa_g\right), \quad \frac{\|w\|_\infty}{\lambda_{\min}\left(\mathbf{A}^\top\mathbf{W}\mathbf{A}\right)} = O(\nu_g). \tag{35}$$

**Lemma 32.** *Consider running Algorithm 1, with the modification that in Line 7, we set*

$$x_t \leftarrow an \frac{\epsilon}{10}\text{-multiplicative approximation of } \mathrm{argmax}_{\substack{\sum_{i\in[n]} w_i\widetilde{\mathbf{A}}_i \preceq \mathbf{I} \\ x\in\mathbb{R}_{\geq 0}^n}} \langle \kappa v_t, w\rangle,$$

$$\text{where for all } i \in [n], \ \widetilde{\mathbf{A}}_i := \begin{pmatrix} \mathbf{A}_i & \mathbf{0}_{d\times n} \\ \mathbf{0}_{n\times d} & \mathbf{diag}\left(\frac{\kappa_g}{\nu_g}e_i\right) \end{pmatrix}. \tag{36}$$

*Then, if (34) is satisfied for some $\mathbf{A} \in \mathbb{R}^{n\times d}$ and row subset $\mathbf{A}_g \subseteq \mathbf{A}$, Algorithm 1 run on $\kappa \leftarrow \kappa_g$ and $\{\mathbf{A}_i = a_i a_i^\top\}_{i\in[n]}$ where $\{a_i\}_{i\in[n]}$ are rows of $\mathbf{A}$ will produce $w$ satisfying (35).*

*Proof.* We note that each matrix $\widetilde{\mathbf{A}}_i$ is the same as the corresponding $\mathbf{A}_i$, with a single nonzero coordinate along the diagonal bottom-right block. The proof is almost identical to the proof of Lemma 1, so we highlight the main differences here. The main property that Lemma 1 used was that Line 9 did not pass, which lets us conclude (15). Hence, by the approximation guarantee on

each $x_t$, it suffices to show that for any $\mathbf{Y}_t \in \mathbb{S}_{\succeq \mathbf{0}}^d$ with $\mathrm{Tr}(\mathbf{Y}_t) = 1$, (analogously to (14)),

$$\max_{\substack{\sum_{i \in [n]} w_i \widetilde{\mathbf{A}}_i \preceq \mathbf{I} \\ x \in \mathbb{R}_{\geq 0}^n}} \kappa_g \left\langle \mathbf{Y}_t, \sum_{i \in [n]} w_i \mathbf{A}_i \right\rangle \geq 1 - O(\epsilon). \tag{37}$$

However, by taking $w$ to be the 0-1 indicator of the rows of $\mathbf{A}_g$ scaled down by $\lambda_{\max}(\mathbf{A}_g^\top \mathbf{A}_g)$, we have by the promise (34) that

$$\sum_{i \in [n]} w_i \widetilde{\mathbf{A}}_i = \frac{1}{\lambda_{\max}(\mathbf{A}_g^\top \mathbf{A}_g)} \preceq \mathbf{I} \impliedby \frac{1}{\lambda_{\max}(\mathbf{A}_g^\top \mathbf{A}_g)} \mathbf{A}_g^\top \mathbf{A}_g \preceq \mathbf{I}, \ \frac{\kappa_g}{\nu_g} \cdot \frac{1}{\lambda_{\max}(\mathbf{A}_g^\top \mathbf{A}_g)} \leq 1. \tag{38}$$

Now, it suffices to observe that (38) implies our indicator $w$ is feasible for (37), so

$$\max_{\substack{\sum_{i \in [n]} w_i \widetilde{\mathbf{A}}_i \preceq \mathbf{I} \\ x \in \mathbb{R}_{\geq 0}^n}} \kappa_g \left\langle \mathbf{Y}_t, \sum_{i \in [n]} w_i \mathbf{A}_i \right\rangle \geq \frac{\lambda_{\min}\left(\mathbf{A}_g^\top \mathbf{A}_g\right)}{\lambda_{\max}\left(\mathbf{A}_g^\top \mathbf{A}_g\right)} \cdot \kappa_g \geq 1 - O(\epsilon).$$

The remainder of the proof is identical to Lemma 1, where we note the output $w$ satisfies

$$\sum_{i \in [n]} w_i \widetilde{\mathbf{A}}_i \preceq \mathbf{I}, \ \sum_{i \in [n]} w_i \mathbf{A}_i \succeq \frac{1 - O(\epsilon)}{\kappa_g} \mathbf{I},$$

which upon rearrangement and adjusting $\epsilon$ by a constant yields (35). $\qquad\square$

By running the modification of Algorithm 1 described for a given level of $\nu_g$, it is straightforward to perform an incremental search on $\kappa_g$ to find a value satisfying the bound (35) as described in Theorem 13. It is simple to verify that the modification in (36) is not the dominant runtime in any of Theorems 13 or 14 since the added constraint is diagonal and $\widetilde{\mathbf{A}}_i$ is separable. Hence, for every "level" of $\nu_g$ in (34) yielding an appropriate risk bound (32), we can match this risk bound up to a constant factor while obtaining computational speedups scaling with $\kappa_g$.

### Acknowledgments

AS was supported in part by a Microsoft Research Faculty Fellowship, NSF CAREER Award CCF-1844855, NSF Grant CCF-1955039, a PayPal research award, and a Sloan Research Fellowship. KS was supported by a Stanford Data Science Scholarship and a Dantzig-Lieberman Operations Research Fellowship. KT was supported by a Google Ph.D. Fellowship, a Simons-Berkeley VMware Research Fellowship, a Microsoft Research Faculty Fellowship, NSF CAREER Award CCF-1844855, NSF Grant CCF-1955039, and a PayPal research award.

We would like to thank Huishuai Zhang for his contributions to an earlier version of this project, Moses Charikar and Yin Tat Lee for helpful conversations, and anonymous reviewers for feedback on earlier variations of this paper.

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

## A  Deferred proofs from Sections 3 and 5

### A.1  Proof of Proposition 1

We give a proof of Proposition 1 in this section. First, we recall an algorithm for the testing variant of a pure packing SDP problem given in [JLT20].

**Proposition 5** (Theorem 5, [JLT20])**.** *There is an algorithm,* $\mathcal{A}_{\text{test}}$*, which given matrices* $\{\mathbf{M}_i\}_{i\in[n]}$ *and a parameter* $C$*, is an* $\epsilon$*-approximate tester for the decision problem*

$$\text{does there exist } w \in \Delta^n \text{ such that } \sum_{i\in[n]} w_i \mathbf{M}_i \preceq C\mathbf{I}? \tag{39}$$

*The algorithm* $\mathcal{A}_{\text{test}}$ *succeeds with probability* $\geq 1 - \delta$ *and runs in time*

$$O\left(\mathcal{T}_{\text{mv}}\left(\{\mathbf{M}_i\}_{i\in[n]}\right) \cdot \frac{\log^2(nd(\delta\epsilon)^{-1})\log^2 d}{\epsilon^5}\right).$$

*Proof of Proposition 1.* As an immediate result of Proposition 5, we can solve (11) to multiplicative accuracy $\epsilon$ using a binary search. This reduction is derived as Lemma A.1 of [JLL$^+$20], but we give a brief summary here. We subdivide the range $[\text{OPT}_-, \text{OPT}_+]$ into $K$ buckets of multiplicative range $1 + \frac{\epsilon}{3}$, i.e., with endpoints $\text{OPT}_- \cdot (1 + \frac{\epsilon}{3})^k$ for $0 \leq k \leq K$ and

$$K = O\left(\frac{1}{\epsilon} \cdot \log\left(\frac{\text{OPT}_+}{\text{OPT}_-}\right)\right).$$

We then binary search over $0 \leq k \leq K$ to determine the value of $\text{OPT}(v)$ to $\epsilon$-multiplicative accuracy, returning the largest endpoint for which the decision variant in Proposition 5 returns feasible (with accuracy $\frac{\epsilon}{3}$). By the guarantees of Proposition 5, the feasible point returned by Proposition 5 for this endpoint will attain an $\epsilon$-multiplicative approximation to the optimization variant (11), and the runtime is that of Proposition 5 with an overhead of $O(\log K)$.  □

### A.2  Polynomial approximation to the square root

We give a proof of Fact 4.

**Fact 4** (Polynomial approximation of $\sqrt{\cdot}$)**.** *Let* $\mathbf{M} \in \mathbb{S}_{\succ\mathbf{0}}^d$ *have* $\mu\mathbf{I} \preceq \mathbf{M} \preceq \kappa\mu\mathbf{I}$ *where* $\mu$ *is known. Then for any* $\delta \in (0,1)$*, there is an explicit polynomial* $p$ *of degree* $O(\sqrt{\kappa}\log\frac{\kappa}{\delta})$ *with*

$$(1-\delta)\mathbf{M}^{\frac{1}{2}} \preceq p(\mathbf{M}) \preceq (1+\delta)\mathbf{M}^{\frac{1}{2}}.$$

*Proof.* We will instead prove the following fact: for any $\epsilon \in (0,1)$, there is an explicit degree-$O\left(\sqrt{\kappa}\log\frac{\kappa}{\epsilon}\right)$ polynomial $p$ satisfying

$$\max_{x\in[\frac{1}{\kappa},1]} |p(x) - \sqrt{x}| \leq \epsilon.$$

The conclusion for arbitrary scalars with multiplicative range $[\mu, \kappa\mu]$ will then follow from setting $\epsilon = \delta\kappa^{-\frac{1}{2}}$ (giving a multiplicative error guarantee), and the fact that rescaling the range $[\frac{1}{\kappa}, 1]$ will preserve this multiplicative guarantee (adjusting the coefficients of the polynomial as necessary, since $\mu$ is known). Finally, the conclusion for matrices follows since $p(\mathbf{M})$ and $\mathbf{M}^{\frac{1}{2}}$ commute.

Denote $\gamma = \frac{1}{\kappa}$ for convenience. We first shift and scale the function $\sqrt{x}$ to adjust the region of approximation from $[\gamma, 1]$ to $[-1, 1]$. In particular, let $h(x) = \sqrt{\frac{1-\gamma}{2}x + \frac{\gamma+1}{2}}$. If we can find some

degree-$\Delta$ polynomial $g(x)$ with $|g(x) - h(x)| \leq \epsilon$ for all $x \in [-1, 1]$, then

$$p(x) = g\left(\frac{2}{1-\gamma}x - \frac{1+\gamma}{1-\gamma}\right)$$

provides the required approximation to $\sqrt{x}$.

To construct $g$, we take the Chebyshev interpolant of $h(x)$ on the interval $[-1, 1]$. Since $h$ is analytic on $[-1, 1]$, we can apply standard results on the approximation of analytic functions by polynomials, and specifically Chebyshev interpolants. Specifically, by Theorem 8.2 in [Tre12], if $h(z)$ is analytic in an open Bernstein ellipse with parameter $\rho$ in the complex plane, then:

$$\max_{x \in [-1, 1]} |g(x) - h(x)| \leq \frac{4M}{\rho - 1}\rho^{-\Delta},$$

where $M$ is the maximum of $|h(z)|$ for $z$ in the ellipse. It can be checked that $h(x)$ is analytic on an open Bernstein ellipse with parameter $\rho = \frac{1+\sqrt{\gamma}}{1-\sqrt{\gamma}}$ — i.e., with major axis length $\rho + \rho^{-1} = 2\frac{1+\gamma}{1-\gamma}$. We can then check that $M = \sqrt{1+\gamma} \leq \sqrt{2}$ and $\rho - 1 \geq 2\sqrt{\gamma}$. Since for all $\gamma < 1$,

$$\left(\frac{1-\sqrt{\gamma}}{1+\sqrt{\gamma}}\right)^{1/2\gamma} \leq \frac{1}{e},$$

we conclude that $\frac{4M}{\rho-1}\rho^{-\Delta} \leq \epsilon$ as long as $\Delta \geq \frac{1}{2\gamma}\log\left(\frac{\epsilon}{\sqrt{2\gamma}}\right)$, which completes the proof. $\square$

## A.3  Deferred proofs from Section 5.2

**Lemma 18.** *Let $\mathbf{B} \in \mathbb{R}^{d \times d}$ and let $\mathbf{A} \in \mathbb{S}_{\succ 0}^d$. Then $\min\left(\|\mathbf{AB}\|_2, \|\mathbf{BA}\|_2\right) \geq \frac{1}{\kappa(\mathbf{A})}\|\mathbf{B}\|_2\|\mathbf{A}\|_2$.*

*Proof.* We begin with the first entry in the above minimum. Let $v$ be the unit vector with $\|\mathbf{B}v\|_2 = \|\mathbf{B}\|_\infty$, and note $\|\mathbf{AB}v\|_2 \geq \frac{1}{\kappa(\mathbf{A})}\|\mathbf{A}\|_\infty\|\mathbf{B}v\|_2$ by definition of $\kappa(\mathbf{A})$. Hence,

$$\|\mathbf{AB}\|_\infty \geq \|\mathbf{AB}v\|_2 \geq \frac{1}{\kappa(\mathbf{A})}\|\mathbf{A}\|_\infty\|\mathbf{B}v\|_2 = \frac{1}{\kappa(\mathbf{A})}\|\mathbf{A}\|_\infty\|\mathbf{B}\|_\infty.$$

We move onto the second entry. Let $v$ be a vector such that $\|\mathbf{A}v\|_2$ and $\|\mathbf{BA}v\|_2 = \|\mathbf{B}\|_\infty$; note that $\|v\|_2 \leq \frac{\kappa(\mathbf{A})}{\|\mathbf{A}\|_\infty}$. The conclusion follows from rearranging the following display:

$$\frac{\kappa(\mathbf{A})\|\mathbf{BA}\|_\infty}{\|\mathbf{A}\|_\infty} \geq \|\mathbf{BA}\|_\infty\|v\|_2 \geq \|\mathbf{BA}v\|_2 = \|\mathbf{B}\|_\infty.$$

$\square$

**Lemma 26.** *For any matrix $\mathbf{K} \in \mathbb{S}_{\succ 0}^d$ and $\lambda \geq 0$, $\kappa_o^\star(\mathbf{K} + \lambda\mathbf{I}) \leq \kappa_o^\star(\mathbf{K})$.*

*Proof.* By scaling $\mathbf{K}$ by $\lambda$ appropriately (since $\kappa_o^\star$ is invariant under scalar multiplication), it suffices to take $\lambda = 1$. The definition of $\kappa_o^\star$ implies there exists a diagonal matrix $\mathbf{W}$ such that

$$\mathbf{I} \preceq \mathbf{W}^{\frac{1}{2}}\mathbf{K}\mathbf{W}^{\frac{1}{2}} \preceq \kappa_o^\star(\mathbf{K})\mathbf{I} \iff \mathbf{W}^{-1} \preceq \mathbf{K} \preceq \kappa_o^\star(\mathbf{K})\mathbf{W}^{-1}. \tag{40}$$

Thus, to demonstrate $\kappa_o^\star(\mathbf{K} + \mathbf{I}) \leq \kappa_o^\star(\mathbf{K})$ it suffices to exhibit a diagonal $\widetilde{\mathbf{W}}$ such that

$$\widetilde{\mathbf{W}} \preceq \mathbf{K} + \mathbf{I} \preceq \kappa_o^\star(\mathbf{K})\widetilde{\mathbf{W}}.$$

We choose $\widetilde{\mathbf{W}} = \mathbf{W}^{-1} + \mathbf{I}$; then, the above display follows from (40) and $\mathbf{I} \preceq \mathbf{I} \preceq \kappa_o^\star(\mathbf{K})\mathbf{I}$. $\qquad \square$

**Lemma 27.** *Let* $\mathbf{K} \in \mathbb{S}_{\succ \mathbf{0}}^d$. *Then, for* $\lambda \geq \frac{1}{\epsilon}\lambda_{\max}(\mathbf{K})$, $\kappa(\mathbf{K}+\lambda\mathbf{I}) \leq 1+\epsilon$. *Moreover, given a diagonal* $\mathbf{W} \in \mathbb{S}_{\succeq \mathbf{0}}^d$ *such that* $\kappa(\mathbf{W}^{\frac{1}{2}}(\mathbf{K}+\lambda\mathbf{I})\mathbf{W}^{\frac{1}{2}}) \leq \kappa_{\mathrm{scale}}$ *for* $0 \leq \lambda \leq \frac{\epsilon\lambda_{\min}(\mathbf{K})}{1+\epsilon}$, $\kappa(\mathbf{W}^{\frac{1}{2}}\mathbf{K}\mathbf{W}^{\frac{1}{2}}) \leq (1+\epsilon)\kappa_{\mathrm{scale}}$.

*Proof.* To see the first claim, the largest eigenvalue of $\mathbf{K} + \lambda\mathbf{I}$ is at most $\lambda + \lambda_{\max}(\mathbf{K})$ and the smallest is at least $\lambda$, so the condition number is at most $1 + \epsilon$ as desired.

To see the second claim, it follows from the fact that outer rescalings preserve Loewner order, and then combining

$$\mathbf{K} \preceq \mathbf{K} + \lambda\mathbf{I} \implies \lambda_{\max}\left(\mathbf{W}^{\frac{1}{2}}\mathbf{K}\mathbf{W}^{\frac{1}{2}}\right) \leq \lambda_{\max}\left(\mathbf{W}^{\frac{1}{2}}\left(\mathbf{K} + \lambda\mathbf{I}\right)\mathbf{W}^{\frac{1}{2}}\right),$$

$$\mathbf{K} \succeq \frac{1}{1+\epsilon}\left(\mathbf{K} + \lambda\mathbf{I}\right) \implies \lambda_{\min}\left(\mathbf{W}^{\frac{1}{2}}\mathbf{K}\mathbf{W}^{\frac{1}{2}}\right) \geq \frac{1}{1+\epsilon}\lambda_{\min}\left(\mathbf{W}^{\frac{1}{2}}\left(\mathbf{K} + \lambda\mathbf{I}\right)\mathbf{W}^{\frac{1}{2}}\right).$$

$\qquad \square$

**Lemma 28.** *Let* $\mathbf{K} \in \mathbb{S}_{\succ \mathbf{0}}^d$, *and let* $\mathbf{W} \in \mathbb{S}_{\succeq \mathbf{0}}^d$ *be diagonal. Then for any* $\lambda > 0$,

$$\kappa\left(\mathbf{W}^{\frac{1}{2}}\left(\mathbf{K} + \lambda\mathbf{I}\right)\mathbf{W}^{\frac{1}{2}}\right) \leq 2\kappa\left(\mathbf{W}^{\frac{1}{2}}\left(\mathbf{K} + \frac{\lambda}{2}\mathbf{I}\right)\mathbf{W}^{\frac{1}{2}}\right).$$

*Proof.* First, because outer rescalings preserve Loewner order, it is immediate that

$$\mathbf{K} + \frac{\lambda}{2}\mathbf{I} \preceq \mathbf{K} + \lambda\mathbf{I} \implies \lambda_{\max}\left(\mathbf{W}^{\frac{1}{2}}\left(\mathbf{K} + \frac{\lambda}{2}\right)\mathbf{W}^{\frac{1}{2}}\mathbf{I}\right) \leq \lambda_{\max}\left(\mathbf{W}^{\frac{1}{2}}\left(\mathbf{K} + \lambda\mathbf{I}\right)\mathbf{W}^{\frac{1}{2}}\right).$$

Moreover, the same argument shows that

$$\frac{1}{2}\mathbf{K} + \frac{\lambda}{2}\mathbf{I} \preceq \mathbf{K} + \frac{\lambda}{2}\mathbf{I} \implies \lambda_{\min}\left(\mathbf{W}^{\frac{1}{2}}\left(\mathbf{K} + \frac{\lambda}{2}\mathbf{I}\right)\mathbf{W}^{\frac{1}{2}}\right) \geq \frac{1}{2}\lambda_{\min}\left(\mathbf{W}^{\frac{1}{2}}\left(\mathbf{K} + \mathbf{I}\right)\mathbf{W}^{\frac{1}{2}}\right).$$

Combining the above two displays yields the conclusion. $\qquad \square$

# B   M-matrix and SDD matrix facts

Before proving Lemmas 15 and 16, we prove the following fact about the density of the inverses of irreducible invertible symmetric M-matrices. Recall that a matrix $\mathbf{M} \in \mathbb{R}^{n \times n}$ is irreducible if there does not exist a subset $S \subseteq [n]$ with $S \notin \{\emptyset, [n]\}$ such that $\mathbf{M}_{ij} = 0$ for all $i \in S$ and $j \notin S$.

**Lemma 33** (Density of inverses of irreducible symmetric M-matrices). *If* $\mathbf{M} \in \mathbb{R}^{n \times n}$ *is an irreducible invertible symmetric M-matrix then* $[\mathbf{M}^{-1}]_{ij} > 0$ *for all* $i, j \in [n]$.

*Proof.* Recall that $\mathbf{M}$ is an invertible M-matrix if and only if $\mathbf{M} = s\mathbf{I} - \mathbf{A}$ where $s > 0$, $\mathbf{A} \in \mathbb{R}_{\geq 0}^{n \times n}$ and $\rho(\mathbf{A}) < s$. In this case

$$[\mathbf{M}^{-1}]_{ij} = \frac{1}{s}\left[\left(\mathbf{I} - \frac{1}{s}\mathbf{A}\right)^{-1}\right]_{ij} = \frac{1}{s}\sum_{k=0}^{\infty}\left[\left(\frac{1}{s}\mathbf{A}\right)^k\right]_{ij}.$$

Consider the undirected graph $G$ which has $\mathbf{A}$ as its adjacency matrix, i.e., $\mathbf{A}_{ij}$ is the weight of an edge from $i$ to $j$ whenever $\mathbf{A}_{ij} \neq 0$. Now $[\mathbf{A}^k]_{ij} > 0$ if and only if there is a path of length $k$ from $i$ to $j$ in $G$. However, by the assumption that $\mathbf{M}$ is irreducible we have that $G$ is connected and therefore there is a path between any two vertices in the graph and the result follows. $\qquad \square$

**Lemma 15.** *Let* $\mathbf{M}$ *be an invertible symmetric M-matrix. Let* $x = \mathbf{M}^{-1}\mathbb{1}$ *and define* $\mathbf{X} = \mathbf{diag}\,(x)$. *Then* $\mathbf{XMX}$ *is a SDD Z-matrix.*

*Proof.* $\mathbf{XMX}$ is trivially symmetric and therefore it suffices to show that (1) $e_i^\top \mathbf{XMX}e_j < 0$ for all $i \neq j$ and (2) $\mathbf{XMX}\mathbb{1} \geq 0$ entrywise.

For (1) note for all $i \in [n]$, $\mathbf{X}_{ii} = e_i^\top \mathbf{M}^{-1}\mathbb{1} \geq 0$ as $\mathbf{M}^{-1}$ is nonnegative by Lemma 33 and $\mathbf{X}e_i = \mathbf{X}_{ii}e_i$ as $\mathbf{X}$ is diagonal. Using these two equalities for all $i \neq j$ we obtain $e_i^\top \mathbf{XMX}e_j = (\mathbf{X}_{ii}\mathbf{X}_{jj}) \cdot e_i^\top \mathbf{M}e_j \leq 0$ as $\mathbf{M}_{ij} \leq 0$ by definition of M-matrices.

For (2) note $\mathbf{X}\mathbb{1} = \mathbf{M}^{-1}\mathbb{1}$ and $\mathbf{XMX}\mathbb{1} = \mathbf{XMM}^{-1}\mathbb{1} = \mathbf{X}\mathbb{1} = \mathbf{M}^{-1}\mathbb{1} \geq 0$ where again, in the last inequality we used $\mathbf{M}^{-1}$ is entrywise nonnegative by Lemma 33. $\qquad\square$

**Lemma 16.** *Let* $\mathbf{A}$ *be an invertible SDD Z-matrix. For any* $\alpha \geq 0$, *the matrix* $\mathbf{B} = (\mathbf{A}^{-1} + \alpha\mathbf{I})^{-1}$ *is also an invertible SDD Z-matrix.*

*Proof.* $\mathbf{B}$ is clearly symmetric and therefore to show that $\mathbf{B}$ is a SDD Z-matrix it suffices to show that (1) $e_i^\top \mathbf{B}e_j \leq 0$ for all $i \neq j$ and (2) $\mathbf{B}\mathbb{1} \geq 0$.

The claim is trivial when $\alpha = 0$ so we assume without loss of generality that $\alpha > 0$. To prove the inequality we use that by the Woodbury matrix identity it holds that

$$\mathbf{B} = \alpha^{-1}\mathbf{I} - \alpha^{-2}\left(\mathbf{A} + \alpha^{-1}\mathbf{I}\right)^{-1} = \mathbf{A} - \mathbf{A}(\alpha^{-1}\mathbf{I} + \mathbf{A})^{-1}\mathbf{A}.$$

Further, we use that $\mathbf{A} + \alpha^{-1}\mathbf{I}$ is a M-matrix by definition and therefore $\left(\mathbf{A} + \alpha^{-1}\mathbf{I}\right)^{-1}$ is an inverse M-matrix that has entrywise nonnegative entries by Lemma 33.

Now for (1) by these two claims we have that for all $i \neq j$ it is the case that

$$e_i^\top \mathbf{B}e_j = e_i^\top\left(\alpha^{-1}\mathbf{I} - \alpha^{-2}\left(\mathbf{A} + \alpha^{-1}\mathbf{I}\right)^{-1}\right)e_j = \frac{1}{\alpha^2}e_i^\top\left(\mathbf{A} + \alpha^{-1}\mathbf{I}\right)^{-1}e_j \leq 0\ .$$

For (2) we use the other Woodbury matrix equality and see that

$$\mathbf{B}\mathbb{1} = \left[(\alpha^{-1}\mathbf{I} + \mathbf{A}) - \mathbf{A}\right](\alpha^{-1}\mathbf{I} + \mathbf{A})^{-1}\mathbf{A}\mathbb{1} = \alpha^{-1}(\alpha^{-1}\mathbf{I} + \mathbf{A})^{-1}\mathbf{A}\mathbb{1} \geq 0 \text{ entrywise.}$$

This final inequality follows from the fact that $\mathbf{A}\mathbb{1} \geq 0$ entrywise because $\mathbf{A}$ is a SDD Z-matrix and hence $(\alpha^{-1}\mathbf{I} + \mathbf{A})^{-1}(\mathbf{A}\mathbb{1}) \geq 0$ because $(\alpha^{-1}\mathbf{I} + \mathbf{A})^{-1}$ is entrywise nonnegative. $\qquad\square$

**Lemma 17.** *If* $\mathbf{A}^\dagger$ *is a Laplacian of a connected graph, then* $(\mathbf{A} + \alpha\mathbf{L}_{K_n})^\dagger$ *for any* $\alpha > 0$ *is a Laplacian matrix.*

*Proof.* It is clear that $(\mathbf{A} + \alpha\mathbf{L}_{K_n})^\dagger$ is symmetric, positive semidefinite, and has $\mathbb{1}$ in its kernel. By standard algebraic manipulation and the Woodbury matrix identity we have,

$$(\mathbf{A} + \alpha\mathbf{L}_{K_n})^\dagger = \left(\mathbf{A} + \alpha\mathbf{L}_{K_n} + 2\alpha\mathbb{1}\mathbb{1}^\top\right)^{-1} - \frac{1}{2\alpha n^2}\mathbb{1}\mathbb{1}^\top = \left(\mathbf{A} + \alpha\mathbb{1}\mathbb{1}^\top + \alpha n\mathbf{I}\right)^{-1} - \frac{1}{2\alpha n^2}\mathbb{1}\mathbb{1}^\top$$

$$= \left(\left(\mathbf{A}^\dagger + \frac{1}{\alpha n^2}\mathbb{1}\mathbb{1}^\top\right)^{-1} + \alpha n\mathbf{I}\right)^{-1} - \frac{1}{2\alpha n^2}\mathbb{1}\mathbb{1}^\top = \frac{1}{\alpha n}\left(\frac{1}{\alpha n}\left(\mathbf{A}^\dagger + \frac{1}{\alpha n^2}\mathbb{1}\mathbb{1}^\top\right)^{-1} + \mathbf{I}\right)^{-1} - \frac{1}{2\alpha n^2}\mathbb{1}\mathbb{1}^\top$$

$$= \frac{1}{\alpha n}\left[\left(\left(\alpha n\mathbf{A}^\dagger + \frac{1}{n}\mathbb{1}\mathbb{1}^\top\right)^{-1} + \mathbf{I}\right)^{-1} - \frac{1}{2n}\mathbb{1}\mathbb{1}^\top\right] = \frac{1}{\alpha n}\left[\mathbf{I} - \left(\alpha n\mathbf{A}^\dagger + \frac{1}{n}\mathbb{1}\mathbb{1}^\top + \mathbf{I}\right)^{-1} - \frac{1}{2n}\mathbb{1}\mathbb{1}^\top\right]$$

$$= \frac{1}{\alpha n}\left[\mathbf{I} - \left(\alpha n\mathbf{A}^\dagger + \mathbf{I}\right)^{-1}\right].$$

The conclusion follows since $\alpha n \mathbf{A}^\dagger + \mathbf{I}$ is a positive definite SDD matrix: its inverse is entrywise nonpositive on off-diagonals by Lemma 16, and thus $(\mathbf{A} + \alpha \mathbf{L}_{K_n})^\dagger$ is a Z-matrix, and hence also a Laplacian. $\qquad\square$

## C Jacobi preconditioning

In this section, we analyze a popular heuristic for computing diagonal preconditioners. Given a positive definite matrix $\mathbf{K} \in \mathbb{S}^d_{\succ \mathbf{0}}$, consider applying the outer scaling

$$\mathbf{W}^{\frac{1}{2}} \mathbf{K} \mathbf{W}^{\frac{1}{2}}, \text{ where } \mathbf{W} = \mathbf{diag}\,(w) \text{ and } w_i := \mathbf{K}_{ii}^{-1} \text{ for all } i \in [d]. \tag{41}$$

In other words, the result of this scaling is to simply normalize the diagonal of $\mathbf{K}$ to be all ones; we remark $\mathbf{W}$ has strictly positive diagonal entries, else $\mathbf{K}$ is not positive definite. Also called the Jacobi preconditioner, a result of Van de Sluis [GR89, vdS69] proves that *for any matrix* this scaling leads to a condition number that is within an $m$ factor of optimal, where $m \leq d$ is the maximum number of non-zeros in any row of $\mathbf{K}$. For completeness, we state a generalization of Van de Sluis's result below. We also require a simple fact; both are proven at the end of this section.

**Fact 6.** *For any* $\mathbf{A}, \mathbf{B} \in \mathbb{S}^d_{\succ \mathbf{0}}$, $\kappa(\mathbf{A}^{\frac{1}{2}} \mathbf{B} \mathbf{A}^{\frac{1}{2}}) \leq \kappa(\mathbf{A}) \kappa(\mathbf{B})$.

**Proposition 6.** *Let* $\mathbf{W}$ *be defined as in* (41) *and let $m$ denote the maximum number of non-zero's in any row of* $\mathbf{K}$*. Then,*

$$\kappa \left( \mathbf{W}^{\frac{1}{2}} \mathbf{K} \mathbf{W}^{\frac{1}{2}} \right) \leq \min \left( m, \sqrt{\mathrm{nnz}(\mathbf{K})} \right) \cdot \kappa_o^\star (\mathbf{K}).$$

Note that $m$ and $\sqrt{\mathrm{nnz}(\mathbf{K})}$ are both $\leq d$, so it follows that $\kappa(\mathbf{W}^{\frac{1}{2}} \mathbf{K} \mathbf{W}^{\frac{1}{2}}) \leq d \cdot \kappa_o^\star (\mathbf{K})$. While the approximation factor in Proposition 6 depends on the dimension or sparsity of $\mathbf{K}$, we show that a similar analysis actually yields a *dimension-independent* approximation. Specifically, the Jacobi preconditioner always obtains condition number no worse than the optimal squared. To the best of our knowledge, this simple but powerful bound has not been observed in prior work.

**Proposition 7.** *Let* $\mathbf{W}$ *be defined as in* (41)*. Then,*

$$\kappa \left( \mathbf{W}^{\frac{1}{2}} \mathbf{K} \mathbf{W}^{\frac{1}{2}} \right) \leq (\kappa_o^\star (\mathbf{K}))^2.$$

*Proof.* Let $\mathbf{W}_\star$ attain the minimum in the definition of $\kappa_o^\star$, i.e., $\kappa(\mathbf{K}_\star) = \kappa_o^\star(\mathbf{K})$ for $\mathbf{K}_\star := \mathbf{W}_\star^{\frac{1}{2}} \mathbf{K} \mathbf{W}_\star^{\frac{1}{2}}$. Note that since $[\mathbf{W}^{\frac{1}{2}} \mathbf{K} \mathbf{W}^{\frac{1}{2}}]_{ii} = 1$ by definition of $\mathbf{W}$ it follows that for all $i$

$$[\mathbf{W}_\star \mathbf{W}^{-1}]_{ii} = [\mathbf{W}_\star \mathbf{W}^{-1}]_{ii} \cdot [\mathbf{W}^{\frac{1}{2}} \mathbf{K} \mathbf{W}^{\frac{1}{2}}]_{ii} = [\mathbf{K}_\star]_{ii} = e_i^\top \mathbf{K}_\star e_i \in [\lambda_{\min}(\mathbf{K}_\star), \lambda_{\max}(\mathbf{K}_\star)]$$

where the last step used that $\lambda_{\min}(\mathbf{K}_\star)\mathbf{I} \preceq \mathbf{K}_\star \preceq \lambda_{\max}(\mathbf{K}_\star)\mathbf{I}$. Consequently, for $\widetilde{\mathbf{W}} := \mathbf{W}_\star^{-1} \mathbf{W}$ it follows that $\kappa(\widetilde{\mathbf{W}}) = \kappa(\widetilde{\mathbf{W}}^{-1}) \leq \lambda_{\max}(\mathbf{K}_\star)/\lambda_{\min}(\mathbf{K}_\star) = \kappa(\mathbf{K}_\star)$. The result follows from Fact 6 as

$$\kappa \left( \mathbf{W}^{\frac{1}{2}} \mathbf{K} \mathbf{W}^{\frac{1}{2}} \right) = \kappa \left( \widetilde{\mathbf{W}}^{\frac{1}{2}} \mathbf{K}_\star \widetilde{\mathbf{W}}^{\frac{1}{2}} \right) \leq \kappa \left( \widetilde{\mathbf{W}} \right) \kappa (\mathbf{K}_\star) \leq (\kappa_o^\star)^2.$$

$\qquad\square$

Next, we demonstrate that Proposition 7 is essentially tight by exhibiting a family of matrices which attain the bound of Proposition 7 up to a constant factor. At a high level, our strategy is

to create two blocks where the "scales" of the diagonal normalizing rescaling are at odds, whereas a simple rescaling of one of the blocks would result in a quadratic savings in conditioning.

**Proposition 8.** *Consider a $2d \times 2d$ matrix $\mathbf{M}$ such that*

$$\mathbf{K} = \begin{pmatrix} \mathbf{A} & \mathbf{0} \\ \mathbf{0} & \mathbf{B} \end{pmatrix}, \ \mathbf{A} = \sqrt{d}\mathbf{I} + \mathbb{1}\mathbb{1}^\top, \ \mathbf{B} = \mathbf{I} - \frac{1}{\sqrt{d} + d}\mathbb{1}\mathbb{1}^\top,$$

*where $\mathbf{A}$ and $\mathbf{B}$ are $d \times d$. Then, defining $\mathbf{W}$ as in (41),*

$$\kappa\left(\mathbf{W}^{\frac{1}{2}}\mathbf{K}\mathbf{W}^{\frac{1}{2}}\right) = \Theta(d), \ \kappa_o^\star(\mathbf{K}) = \Theta\left(\sqrt{d}\right).$$

*Proof.* Because $\mathbf{W}^{\frac{1}{2}}\mathbf{K}\mathbf{W}^{\frac{1}{2}}$ is blockwise separable, to understand its eigenvalue distribution it suffices to understand the eigenvalues of the two blocks. First, the upper-left block (the rescaling of the matrix $\mathbf{A}$) is multiplied by $\frac{1}{\sqrt{d}+1}$. It is straightforward to see that the resulting eigenvalues are

$$\frac{\sqrt{d}}{\sqrt{d}+1} \text{ with multiplicity } d-1, \ \sqrt{d} \text{ with multiplicity } 1.$$

Similarly, the bottom-right block is multiplied by $\frac{d+\sqrt{d}}{d+\sqrt{d}-1}$, and hence its rescaled eigenvalues are

$$\frac{d+\sqrt{d}}{d+\sqrt{d}-1} \text{ with multiplicity } d-1, \ \frac{\sqrt{d}}{d+\sqrt{d}-1} \text{ with multiplicity } 1.$$

Hence, the condition number of $\mathbf{W}^{\frac{1}{2}}\mathbf{K}\mathbf{W}^{\frac{1}{2}}$ is $d + \sqrt{d} - 1 = \Theta(d)$. However, had we rescaled the top-left block to be a $\sqrt{d}$ factor smaller, it is straightforward to see the resulting condition number is $O(\sqrt{d})$. On the other hand, since the condition number of $\mathbf{K}$ is $O(d)$, Proposition 7 shows that the optimal condition number $\kappa_o^\star(\mathbf{K})$ is $\Omega(\sqrt{d})$, and combining yields the claim. We remark that as $d \to \infty$, the constants in the upper and lower bounds agree up to a low-order term. $\square$

Finally, we provide the requisite proofs of Fact 6 and Proposition 6.

**Fact 6.** *For any $\mathbf{A}, \mathbf{B} \in \mathbb{S}_{\succ \mathbf{0}}^d$, $\kappa(\mathbf{A}^{\frac{1}{2}}\mathbf{B}\mathbf{A}^{\frac{1}{2}}) \leq \kappa(\mathbf{A})\kappa(\mathbf{B})$.*

*Proof.* It is straightforward from $\lambda_{\min}(\mathbf{A})\mathbf{I} \preceq \mathbf{A} \preceq \lambda_{\max}(\mathbf{A})\mathbf{I}$ that

$$\sqrt{\lambda_{\min}(\mathbf{A})}\|u\|_2 \leq \left\|\mathbf{A}^{\frac{1}{2}}u\right\|_2 \leq \sqrt{\lambda_{\max}(\mathbf{A})}\|u\|_2,$$

and an analogous fact holds for $\mathbf{B}$. Hence, we can bound the eigenvalues of $\mathbf{A}^{\frac{1}{2}}\mathbf{B}\mathbf{A}^{\frac{1}{2}}$:

$$\lambda_{\max}\left(\mathbf{A}^{\frac{1}{2}}\mathbf{B}\mathbf{A}^{\frac{1}{2}}\right) = \max_{\|u\|_2=1} u^\top \mathbf{A}^{\frac{1}{2}}\mathbf{B}\mathbf{A}^{\frac{1}{2}}\mathbf{B}u \leq \lambda_{\max}(\mathbf{A}) \max_{\|v\|_2=1} v^\top \mathbf{B}v = \lambda_{\max}(\mathbf{A})\lambda_{\max}(\mathbf{B}),$$

$$\lambda_{\min}\left(\mathbf{A}^{\frac{1}{2}}\mathbf{B}\mathbf{A}^{\frac{1}{2}}\right) = \min_{\|u\|_2=1} u^\top \mathbf{A}^{\frac{1}{2}}\mathbf{B}\mathbf{A}^{\frac{1}{2}}\mathbf{B}u \geq \lambda_{\min}(\mathbf{A}) \min_{\|v\|_2=1} v^\top \mathbf{B}v = \lambda_{\min}(\mathbf{A})\lambda_{\min}(\mathbf{B}).$$

Dividing the above two equations yields the claim. $\square$

**Proposition 6.** *Let $\mathbf{W}$ be defined as in (41) and let $m$ denote the maximum number of non-zero's in any row of $\mathbf{K}$. Then,*

$$\kappa\left(\mathbf{W}^{\frac{1}{2}}\mathbf{K}\mathbf{W}^{\frac{1}{2}}\right) \leq \min\left(m, \sqrt{\mathrm{nnz}(\mathbf{K})}\right) \cdot \kappa_o^\star(\mathbf{K}).$$

*Proof.* Throughout let $\kappa_o^\star := \kappa_o^\star(\mathbf{K})$ for notational convenience. Let $\mathbf{W}_\star$ obtain the minimum in the definition of $\kappa_o^\star$ and let $\mathbf{B} = \mathbf{W}_\star^{\frac{1}{2}}\mathbf{K}\mathbf{W}_\star^{\frac{1}{2}}$. Also let $\mathbf{W_B}$ be the inverse of a diagonal matrix with the same entries as $\mathbf{B}$'s diagonal. Note that $\kappa(\mathbf{B}) = \kappa_o^\star$ and $\mathbf{W_B}^{\frac{1}{2}}\mathbf{B}\mathbf{W_B}^{\frac{1}{2}} = \mathbf{W}^{\frac{1}{2}}\mathbf{K}\mathbf{W}^{\frac{1}{2}}$. So, to prove Proposition 6, it suffices to prove that

$$\kappa\left(\mathbf{W_B}^{\frac{1}{2}}\mathbf{B}\mathbf{W_B}^{\frac{1}{2}}\right) \leq \min\left(m, \sqrt{\mathrm{nnz}(\mathbf{K})}\right) \cdot \kappa_o^\star.$$

Let $d_{\max}$ denote the largest entry in $\mathbf{W_B}^{-1}$. We have that $d_{\max} \leq \lambda_{\max}(\mathbf{B})$. Then let $\mathbf{M} = (d_{\max}\mathbf{W_B})^{\frac{1}{2}}\mathbf{B}(d_{\max}\mathbf{W_B})^{\frac{1}{2}}$ and note that all of $\mathbf{M}$'s diagonal entries are equal to $d_{\max}$ and $\kappa(\mathbf{M}) = \kappa(\mathbf{W_B}^{\frac{1}{2}}\mathbf{B}\mathbf{W_B}^{\frac{1}{2}})$. Moreover, since $d_{\max}\mathbf{W_B}$ has all entries $\geq 1$, $\lambda_{\min}(\mathbf{M}) \geq \lambda_{\min}(\mathbf{B})$. Additionally, since a PSD matrix must have its largest entry on the diagonal, we have that $\|\mathbf{M}\|_{\mathrm{F}}^2 \leq \mathrm{nnz}(\mathbf{M})d_{\max}^2 \leq \mathrm{nnz}(\mathbf{M})\lambda_{\max}(\mathbf{B})^2$. Accordingly, $\lambda_{\max}(\mathbf{M}) = \|\mathbf{M}\|_2 \leq \|\mathbf{M}\|_{\mathrm{F}} \leq \sqrt{\mathrm{nnz}(\mathbf{M})}\lambda_{\max}(\mathbf{B})$.

From this lower bound on $\lambda_{\min}(\mathbf{M})$ and upper bound on $\lambda_{\max}(\mathbf{M})$, we have that

$$\kappa\left(\mathbf{W}^{\frac{1}{2}}\mathbf{K}\mathbf{W}^{\frac{1}{2}}\right) = \kappa(\mathbf{M}) \leq \frac{\sqrt{\mathrm{nnz}(\mathbf{M})}\lambda_{\max}(\mathbf{B})}{\lambda_{\min}(\mathbf{B})} = \sqrt{\mathrm{nnz}(\mathbf{M})} \cdot \kappa(\mathbf{B}).$$

This proves one part of the minimum in Proposition 6. The second, which was already proven in [vdS69] follows similarly. In particular, by the Gershgorin circle theorem we have that $\lambda_{\max}(\mathbf{M}) \leq \max_{i\in[d]}\|\mathbf{M}_{i:}\|_1$, where $\mathbf{M}_{i:}$ denotes the $i^{\mathrm{th}}$ row for $\mathbf{M}$. Since all entries in $\mathbf{M}$ are bounded by $d_{\max} \leq \lambda_{\max}(\mathbf{B})$, we have that $\max_{i\in[d]}\|\mathbf{M}_{i:}\|_1 \leq m\lambda_{\max}(\mathbf{B})$, and thus

$$\kappa\left(\mathbf{W}^{\frac{1}{2}}\mathbf{K}\mathbf{W}^{\frac{1}{2}}\right) = \kappa(\mathbf{M}) \leq \frac{m\lambda_{\max}(\mathbf{B})}{\lambda_{\min}(\mathbf{B})} = m \cdot \kappa(\mathbf{B}).$$

$\square$

# D Faster scalings with a conjectured subroutine

In this section, we demonstrate algorithms which achieve runtimes which scale as $\widetilde{O}(\sqrt{\kappa^\star})$[14] matrix-vector multiplies for computing approximately optimal scalings, assuming the existence of a sufficiently general *width-independent* mixed packing and covering (MPC) SDP solver. Such runtimes (which improve each of Theorems 14 and 13 by roughly a $\kappa^\star$ factor) would nearly match the cost of the fastest solvers *after* rescaling, e.g. conjugate gradient methods. We also demonstrate that we can achieve near-optimal algorithms for computing constant-factor optimal scalings for average-case notions of conditioning under this assumption.

We first recall the definition of the general MPC SDP feasibility problem.

**Definition 3** (MPC feasibility problem). *Given sets of matrices $\{\mathbf{P}_i\}_{i\in[n]} \in \mathbb{S}_{\succeq\mathbf{0}}^{d_p}$ and $\{\mathbf{C}_i\}_{i\in[n]} \in \mathbb{S}_{\succeq\mathbf{0}}^{d_c}$, and error tolerance $\epsilon \in (0,1)$, the mixed packing-covering (MPC) feasibility problem asks to return weights $w \in \mathbb{R}_{\geq 0}^n$ such that*

$$\lambda_{\max}\left(\sum_{i\in[n]} w_i\mathbf{P}_i\right) \leq (1+\epsilon)\lambda_{\min}\left(\sum_{i\in[n]} w_i\mathbf{C}_i\right), \tag{42}$$

---

[14]Throughout this section for brevity, we use $\kappa^\star$ to interchangeably refer to the quantities $\kappa_i^\star$ or $\kappa_o^\star$ of a particular appropriate inner or outer rescaling problem.

*or conclude that the following is infeasible for $w \in \mathbb{R}^n_{\geq 0}$:*

$$\lambda_{\max}\left(\sum_{i\in[n]} w_i \mathbf{P}_i\right) \leq \lambda_{\min}\left(\sum_{i\in[n]} w_i \mathbf{C}_i\right). \tag{43}$$

*If both (42) is feasible and (43) is infeasible, either answer is acceptable.*

Throughout this section, we provide efficient algorithms under Assumption 1: namely, that there exists a solver for the MPC feasibility problem at constant $\epsilon$ with polylogarithmic iteration complexity and sufficient approximation tolerance. Such a solver would improve upon our algorithm in Section 3 both in generality (i.e. without the restriction that the constraint matrices are multiples of each other) and in the number of iterations.

**Assumption 1.** *There is an algorithm* MPC *which takes inputs* $\{\mathbf{P}_i\}_{i\in[n]} \in \mathbb{S}^{d_p}_{\succeq 0}$, $\{\mathbf{C}_i\}_{i\in[n]} \in \mathbb{S}^{d_c}_{\succeq 0}$, *and error tolerance* $\epsilon$, *and solves problem (42), (43), in* $\mathrm{poly}(\log(nd\rho), \epsilon^{-1})$ *iterations, where* $d := \max(d_p, d_c)$, $\rho := \max_{i\in[n]} \frac{\lambda_{\max}(\mathbf{C}_i)}{\lambda_{\max}(\mathbf{P}_i)}$. *Each iteration uses* $O(1)$ $n$*-dimensional vector operations, and for* $\epsilon' = \Theta(\epsilon)$ *with an appropriate constant, additionally requires computation of*

$$\epsilon'\text{-multiplicative approximations to } \left\langle \mathbf{P}_i, \frac{\exp\left(\sum_{i\in[n]} w_i \mathbf{P}_i\right)}{\mathrm{Tr}\exp\left(\sum_{i\in[n]} w_i \mathbf{P}_i\right)} \right\rangle \ \forall i \in [n],$$

$$\left(\epsilon', e^{\frac{-\log(nd\rho)}{\epsilon'}}\mathrm{Tr}(\mathbf{C}_i)\right)\text{-approximations to } \left\langle \mathbf{C}_i, \frac{\exp\left(-\sum_{i\in[n]} w_i \mathbf{C}_i\right)}{\mathrm{Tr}\exp\left(-\sum_{i\in[n]} w_i \mathbf{C}_i\right)} \right\rangle \ \forall i \in [n], \tag{44}$$

*for* $w \in \mathbb{R}^n_{\geq 0}$ *with* $\lambda_{\max}\left(\sum_{i\in[n]} w_i \mathbf{P}_i\right), \lambda_{\min}\left(\sum_{i\in[n]} w_i \mathbf{C}_i\right) \leq R$ *for* $R = O(\frac{\log(nd\rho)}{\epsilon})$.

In particular, we observe that the number of iterations of this conjectured subroutine depends polylogarithmically on $\rho$, i.e. the runtime is *width-independent*.[15] In our settings computing optimal rescaled condition numbers, $\rho = \Theta(\kappa^\star)$; our solver in Section 3 has an iteration count depending linearly on $\rho$. Such runtimes are known for MPC linear programs [MRWZ16a], however, such rates have been elusive in the SDP setting. While the form of requirements in (44) may seem somewhat unnatural at first glance, we observe that this is the natural generalization of the error tolerance of known width-independent MPC LP solvers [MRWZ16a]. Moreover, these approximations mirror the tolerances of our width-dependent solver in Section 3 (see Line 6 and Corollary 4).

We first record the following technical lemma, which we will repeatedly use.

**Lemma 34.** *Given a matrix* $\mathbf{0} \preceq \mathbf{M} \preceq R\mathbf{I}$ *for some* $R > 0$, *sufficiently small constant* $\epsilon$, *and* $\delta \in (0, 1)$, *we can compute* $\epsilon$*-multiplicative approximations to the quantities*

$$\left\langle a_i a_i^\top, \exp(\mathbf{M}) \right\rangle \text{ for all } i \in [n], \text{ and } \mathrm{Tr}\exp(\mathbf{M})$$

*in time* $O((\mathcal{T}_{\mathrm{mv}}(\mathbf{M})R + \mathrm{nnz}(\mathbf{A}))\log\frac{n}{\delta})$, *with probability at least* $1 - \delta$.

*Proof.* We discuss both parts separately. Regarding computing the inner products, equivalently, the goal is to compute approximations to all $\left\|\exp(\frac{1}{2}\mathbf{M})a_i\right\|_2^2$ for $i \in [n]$. First, by an application of

---

[15]The literature on approximate solvers for positive linear programs and semidefinite programs refer to logarithmic dependences on $\rho$ as width-independent, and we follow this convention in our exposition.

Fact 2 with $\delta = \frac{\epsilon}{8}\exp(-2R)$, and then multiplying all sides of the inequality by $\exp(R)$, there is a degree-$O(R)$ polynomial such that

$$\left(1 - \frac{\epsilon}{8}\right)\exp\left(\frac{1}{2}\mathbf{M}\right) \preceq \exp\left(\frac{1}{2}\mathbf{M}\right) - \frac{\epsilon}{8}\mathbf{I} \preceq p\left(\frac{1}{2}\mathbf{M}\right) \preceq \exp\left(\frac{1}{2}\mathbf{M}\right) + \frac{\epsilon}{8}\mathbf{I} \preceq \left(1 + \frac{\epsilon}{8}\right)\exp\left(\frac{1}{2}\mathbf{M}\right)$$

$$\implies \left(1 - \frac{\epsilon}{3}\right)\exp(\mathbf{M}) \preceq p\left(\frac{1}{2}\mathbf{M}\right)^2 \preceq \left(1 + \frac{\epsilon}{3}\right)\exp(\mathbf{M}).$$

This implies that $\left\|p(\frac{1}{2}\mathbf{M})a_i\right\|_2^2$ approximates $\left\|\exp(\frac{1}{2}\mathbf{M})a_i\right\|_2^2$ to a multiplicative $\frac{\epsilon}{3}$ by the definition of Loewner order. Moreover, applying Fact 1 with a sufficiently large $k = O(\log\frac{n}{\delta})$ implies by a union bound that for all $i \in [n]$, $\left\|\mathbf{Q}p(\frac{1}{2}\mathbf{M})a_i\right\|_2^2$ is a $\epsilon$-multiplicative approximation to $\left\|\exp(\frac{1}{2}\mathbf{M})a_i\right\|_2^2$. To compute all the vectors $\mathbf{Q}p(\frac{1}{2}\mathbf{M})a_i$, it suffices to first apply $p(\frac{1}{2}\mathbf{M})$ to all rows of $\mathbf{Q}$, which takes time $O(\mathcal{T}_{\mathrm{mv}}(\mathbf{M}) \cdot kR)$ since $p$ is a degree-$O(R)$ polynomial. Next, once we have the explicit $k \times d$ matrix $\mathbf{Q}p(\frac{1}{2}\mathbf{M})$, we can apply it to all $\{a_i\}_{i\in[n]}$ in time $O(\mathrm{nnz}(\mathbf{A}) \cdot k)$.

Next, consider computing $\mathrm{Tr}\exp(\mathbf{M})$, which by definition has

$$\mathrm{Tr}\exp(\mathbf{M}) = \sum_{j\in[d]}\left\|\left[\exp\left(\frac{1}{2}\mathbf{M}\right)\right]_{j:}\right\|_2^2.$$

Applying the same $\mathbf{Q}$ and $p$ as before, we have by the following sequence of equalities

$$\sum_{j\in[d]}\left\|\mathbf{Q}\left[\exp\left(\frac{1}{2}\mathbf{M}\right)\right]_{j:}\right\|_2^2 = \mathrm{Tr}\left(\exp\left(\frac{1}{2}\mathbf{M}\right)\mathbf{Q}^\top\mathbf{Q}\exp\left(\frac{1}{2}\mathbf{M}\right)\right)$$

$$= \mathrm{Tr}\left(\mathbf{Q}\exp(\mathbf{M})\mathbf{Q}^\top\right) = \sum_{\ell\in[k]}\left\|\exp\left(\frac{1}{2}\mathbf{M}\right)\mathbf{Q}_{\ell:}\right\|_2^2,$$

that for the desired approximation, it instead suffices to compute

$$\sum_{\ell\in[k]}\left\|p\left(\frac{1}{2}\mathbf{M}\right)\mathbf{Q}_{\ell:}\right\|_2^2.$$

This can be performed in time $O(\mathcal{T}_{\mathrm{mv}}(\mathbf{M}) \cdot kR)$ as previously argued. $\qquad\square$

A straightforward modification of this proof alongside Lemma 19 also implies that we can compute these same quantities to $p(\mathbf{AWA})$, when we are only given $\mathbf{K} = \mathbf{A}^2$, assuming that $\mathbf{K}$ is reasonably well-conditioned. We omit the proof, as it follows almost identically to the proofs of Lemmas 34, 22, and 23, the latter two demonstrating how to appropriately apply Lemma 19.

**Corollary 6.** *Let $\mathbf{K} \in \mathbb{S}_{\succ 0}^d$ such that $\mathbf{K} = \mathbf{A}^2$ and $\kappa(\mathbf{K}) \leq \kappa_{\mathrm{scale}}$. Let $\mathbf{W}$ be a diagonal matrix such that $\lambda_{\max}(\mathbf{AWA}) \leq R$. For $\delta, \epsilon \in (0, 1)$, we can compute $\epsilon$-multiplicative approximations to*

$$\left\langle a_i a_i^\top, \exp(\mathbf{AWA})\right\rangle \text{ for all } i \in [n], \text{ and } \mathrm{Tr}\exp(\mathbf{AWA})$$

*with probability $\geq 1 - \delta$ in time $O\left(\mathcal{T}_{\mathrm{mv}}(\mathbf{K}) \cdot R \cdot \left(R + \sqrt{\kappa_{\mathrm{scale}}}\log\frac{d\kappa_{\mathrm{scale}}}{\delta}\right)\log\frac{n}{\delta}\right)$.*

### D.1 Approximating $\kappa^\star$ under Assumption 1

In this section, we show that, given Assumption 1, we obtain improved runtimes for all three types of diagonal scaling problems, roughly improving Theorems 14 and 13 by a $\kappa^\star$ factor.

**Inner scalings.** We first demonstrate this improvement for inner scalings.

**Theorem 15.** *Under Assumption 1, there is an algorithm which, given full-rank $\mathbf{A} \in \mathbb{R}^{n \times d}$ for $n \geq d$ computes $w \in \mathbb{R}_{\geq 0}^n$ such that $\kappa(\mathbf{A}^\top \mathbf{W} \mathbf{A}) \leq (1 + \epsilon) \kappa_i^\star(\mathbf{A})$ for arbitrarily small $\epsilon = \Theta(1)$, with probability $\geq 1 - \delta$ in time*

$$O\left( \mathrm{nnz}(\mathbf{A}) \cdot \sqrt{\kappa_i^\star(\mathbf{A})} \cdot \mathrm{poly} \log \frac{n \kappa_i^\star(\mathbf{A})}{\delta} \right).$$

*Proof.* For now, assume we know $\kappa_i^\star(\mathbf{A})$ exactly, which we denote as $\kappa_i^\star$ for brevity. Let $\{a_i\}_{i \in [n]}$ denote the rows of $\mathbf{A}$, and assume that $\|a_i\|_2 = 1$ for all $i \in [n]$. By scale invariance, this assumption is without loss of generality. We instantiate Assumption 1 with $\mathbf{P}_i = a_i a_i^\top$ and $\mathbf{C}_i = \kappa_i^\star a_i a_i^\top$, for $i \in [n]$. It is immediate that a solution yields an inner scaling with the same quality up to a $1 + \epsilon$ factor, because by assumption (43) is feasible so MPC cannot return "infeasible."

We now instantiate the primitives in (44) needed by Assumption 1. Throughout, note that $\rho = \kappa_i^\star$ in this setting. Since we run MPC for $\mathrm{poly}(\log n \kappa_i^\star)$ iterations, we will set $\delta' \leftarrow \delta \cdot (\mathrm{poly}(n \kappa_i^\star))^{-1}$ for the failure probability of each of our computations in (44), such that by a union bound all of these computations are correct.

By Lemma 34, we can instantiate the packing gradients to the desired approximation quality in time $O(\mathrm{nnz}(\mathbf{A}) \cdot \mathrm{poly} \log \frac{n \kappa_i^\star}{\delta})$ with probability $1 - \delta'$. By Lemmas 2 and 3, we can instantiate the covering gradients in time $O(\mathrm{nnz}(\mathbf{A}) \sqrt{\kappa_i^\star} \cdot \mathrm{poly} \log \frac{n \kappa_i^\star}{\delta})$ with probability $1 - \delta'$. In applying these lemmas, we use the assumption that $\lambda_{\min}(\sum_{i \in [n]} w_i \mathbf{C}_i) = O(\log n \kappa_i^\star)$ as in Assumption 1, and that the covering matrices are a $\kappa_i^\star$ multiple of the packing matrices so $\lambda_{\max}(\sum_{i \in [n]} w_i \mathbf{C}_i) = O(\kappa_i^\star \log n \kappa_i^\star)$. Thus, the overall runtime of all iterations is

$$O\left( \mathrm{nnz}(\mathbf{A}) \cdot \sqrt{\kappa_i^\star} \cdot \mathrm{poly} \log \frac{n \kappa_i^\star}{\delta} \right)$$

for $\epsilon = \Theta(1)$. To remove the assumption that we know $\kappa_i^\star(\mathbf{A})$, we can use an incremental search on the scaling multiple between $\{\mathbf{C}_i\}_{i \in [n]}$ and $\{\mathbf{P}_i\}_{i \in [n]}$, starting from 1 and increasing by factors of $1 + \epsilon$, adding a constant overhead to the runtime. Our width will never be larger than $O(\kappa_i^\star(\mathbf{A}))$ in any run, since MPC must conclude feasible when the width is sufficiently large. $\qquad \square$

**Outer scalings.** We next discuss the case where we wish to symmetrically outer scale a matrix $\mathbf{K} \in \mathbb{S}_{\succ \mathbf{0}}^d$ near-optimally (i.e. demonstrating an improvement to Theorem 14 under Assumption 1).

**Theorem 16.** *Under Assumption 1, there is an algorithm which, given $\mathbf{K} \in \mathbb{S}_{\succ \mathbf{0}}^d$ computes $w \in \mathbb{R}_{\geq 0}^d$ such that $\kappa(\mathbf{W}^{\frac{1}{2}} \mathbf{K} \mathbf{W}^{\frac{1}{2}}) \leq (1 + \epsilon) \kappa_o^\star(\mathbf{K})$ for arbitrarily small $\epsilon \in \Theta(1)$, with probability $\geq 1 - \delta$ in time*

$$O\left( \mathcal{T}_{\mathrm{mv}}(\mathbf{K}) \cdot \sqrt{\kappa_o^\star(\mathbf{K})} \cdot \mathrm{poly} \log \frac{d \kappa_o^\star(\mathbf{K})}{\delta} \right).$$

*Proof.* Throughout we denote $\kappa_o^\star := \kappa_o^\star(\mathbf{K})$ for brevity. Our proof follows that of Theorem 14, which demonstrates that it suffices to reduce to the case where we have a $\mathbf{K} \in \mathbb{S}_{\succ \mathbf{0}}^d$ with $\kappa(\mathbf{K}) \leq \kappa_{\mathrm{scale}} := 3\kappa_o^\star$, and we wish to find an outer diagonal scaling $\mathbf{W} \in \mathbb{S}_{\succ \mathbf{0}}^d$ such that $\kappa(\mathbf{W}^{\frac{1}{2}} \mathbf{K} \mathbf{W}^{\frac{1}{2}}) \leq (1 + \epsilon) \kappa_o^\star$. We incur a polylogarithmic overhead on the runtime of this subproblem, by using it to solve all phases of the homotopy method in Theorem 14, and the cost of an incremental search on $\kappa_o^\star$.

To solve this problem, we again instantiate Assumption 1 with $\mathbf{P}_i = a_i a_i^\top$ and $\mathbf{C}_i = \kappa_o^\star a_i a_i^\top$, where $\{a_i\}_{i \in [d]}$ are rows of $\mathbf{A} := \mathbf{K}^{\frac{1}{2}}$. As in the proof of Theorem 14, the main difficulty is to implement the gradients in (44) with only implicit access to $\mathbf{A}$, which we again will perform to probability $1 - \delta'$ for some $\delta' = \delta \cdot (\mathrm{poly}(n\kappa_o^\star))^{-1}$ which suffices by a union bound. Applying Lemmas 22 and 23 with the same parameters as in the proof of Theorem 14 (up to constants) implies that we can approximate the covering gradients in (44) to the desired quality within time

$$O\left(\mathcal{T}_{\mathrm{mv}}(\mathbf{K}) \cdot \sqrt{\kappa_o^\star} \cdot \mathrm{poly}\log \frac{n\kappa_o^\star}{\delta}\right).$$

Similarly, Corollary 6 implies we can compute the necessary approximate packing gradients in the same time. Multiplying by the overhead of the homotopy method in Theorem 14 gives the result. $\square$

### D.2 Average-case conditioning under Assumption 1

A number of recent linear system solvers depend on average notions of conditioning, namely the ratio between the average eigenvalue and smallest [SV06, LS13, JZ13, DBL14, AQRY16, All17, AKK$^+$20]. Normalized by dimension, we define this average conditioning as follows: for $\mathbf{M} \in \mathbb{S}_{\succ \mathbf{0}}^d$,

$$\tau(\mathbf{M}) := \frac{\mathrm{Tr}(\mathbf{M})}{\lambda_{\min}(\mathbf{M})}.$$

Observe that since $\mathrm{Tr}(\mathbf{M})$ is the sum of eigenvalues, the following inequalities always hold:

$$d \leq \tau(\mathbf{M}) \leq d\kappa(\mathbf{M}). \tag{45}$$

In analogy with $\kappa_i^\star$ and $\kappa_o^\star$, we define for full-rank $\mathbf{A} \in \mathbb{R}^{n \times d}$ for $n \geq d$, and $\mathbf{K} \in \mathbb{S}_{\succ \mathbf{0}}^d$,

$$\tau_i^\star(\mathbf{A}) := \min_{\text{diagonal } \mathbf{W} \succeq \mathbf{0}} \tau\left(\mathbf{A}^\top \mathbf{W} \mathbf{A}\right), \; \tau_o^\star(\mathbf{K}) := \min_{\text{diagonal } \mathbf{W} \succeq \mathbf{0}} \tau\left(\mathbf{W}^{\frac{1}{2}} \mathbf{K} \mathbf{W}^{\frac{1}{2}}\right). \tag{46}$$

We give an informal discussion on how to use Assumption 1 to develop a solver for approximating $\tau_i^\star$ to a constant factor, which has a runtime nearly-matching the fastest linear system solvers depending on $\tau_i^\star$ after applying the appropriate rescalings.[16] Qualitatively, this may be thought of as the average-case variant of Theorem 15. We defer an analogous result on approximating $\tau_o^\star$ (with or without a factorization) to future work for brevity.

To develop our algorithm for approximating $\tau_i^\star$, we require several tools. The first is the rational approximation analog of the polynomial approximation in Fact 2.

**Fact 7** (Rational approximation of exp [SV14], Theorem 7.1). *Let* $\mathbf{M} \in \mathbb{S}_{\succeq \mathbf{0}}^d$ *and* $\delta > 0$. *There is an explicit polynomial* $p$ *of degree* $\Delta = \Theta(\log(\delta^{-1}))$ *with absolute coefficients at most* $\Delta^{O(\Delta)}$ *with*

$$\exp(-\mathbf{M}) - \delta\mathbf{I} \preceq p\left(\left(\mathbf{I} + \frac{\mathbf{M}}{\Delta}\right)^{-1}\right) \preceq \exp(-\mathbf{M}) + \delta\mathbf{I}.$$

We also use the runtime of the fastest-known solver for linear systems based on row subsampling, with a runtime dependent on the average conditioning $\tau$. Our goal is to compute reweightings $\mathbf{W}$ which approximately attain the minimums in (46), with runtimes comparable to that of Fact 8.

---

[16]We remark that these problems may be solved to high precision by casting them as an appropriate SDP and applying general SDP solvers, but in this section we focus on fast runtimes.

**Fact 8** ([AKK$^+$20]). *There is an algorithm which given* $\mathbf{M} \in \mathbb{S}_{\succ \mathbf{0}}^d$, $b \in \mathbb{R}^d$, *and* $\delta, \epsilon \in (0,1)$ *returns* $v \in \mathbb{R}^d$ *such that* $\left\| v - \mathbf{M}^{-1} b \right\|_2 \leq \epsilon \left\| \mathbf{M}^{-1} b \right\|_2$ *with probability* $\geq 1 - \delta$ *in time*

$$O\left( \left( n + \sqrt{d\tau(\mathbf{M})} \right) \cdot d \cdot \operatorname{poly} \log \frac{n\tau(\mathbf{K})}{\delta\epsilon} \right).$$

*Remark.* The runtime of Fact 8 applies more broadly to quadratic optimization problems in $\mathbf{M}$, e.g. regression problems of the form $\|\mathbf{A}x - b\|_2^2$ where $\mathbf{A}^\top \mathbf{A} = \mathbf{M}$. Moreover, Fact 8 enjoys runtime improvements when the rows of $\mathbf{M}$ (or the factorization component $\mathbf{A}$) are sparse; our methods in the following discussion do as well as they are directly based on Fact 8, and we omit this discussion for simplicity. Finally, [AKK$^+$20] demonstrates how to improve the dependence on $\sqrt{d\tau(\mathbf{M})}$ to a more fine-grained quantity in the case of non-uniform eigenvalue distributions. We defer obtaining similar improvements for approximating optimal rescalings to interesting future work.

We now give a sketch of how to use Facts 7 and 8 to obtain near-optimal runtimes for computing a rescaling approximating $\tau_i^\star$ under Assumption 1. Let $\mathbf{A} \in \mathbb{R}^{n \times d}$ for $n \geq d$ be full rank, and assume that we known $\tau_i^\star := \tau_i^\star(\mathbf{A})$ for simplicity, which we can approximate using an incremental search with a logarithmic overhead. Denote the rows of $\mathbf{A}$ by $\{a_i\}_{i \in [n]}$. We instantiate Assumption 1 with

$$\mathbf{P}_i = \|a_i\|_2^2, \ \mathbf{C}_i = \tau_i^\star a_i a_i^\top, \ \text{for all } i \in [n], \tag{47}$$

from which it follows that (43) is feasible using the reweighting $\mathbf{W} = \mathbf{diag}(w)$ attaining $\tau_i^\star$:

$$\begin{aligned} \lambda_{\max}\left( \sum_{i \in [n]} w_i \mathbf{P}_i \right) &= \lambda_{\max}\left( \sum_{i \in [n]} w_i \|a_i\|_2^2 \right) = \operatorname{Tr}\left( \mathbf{A}^\top \mathbf{W} \mathbf{A} \right), \\ \lambda_{\min}\left( \sum_{i \in [n]} w_i \mathbf{C}_i \right) &= \tau_i^\star \lambda_{\min}\left( \sum_{i \in [n]} w_i a_i a_i^\top \right) = \tau_i^\star \lambda_{\min}\left( \mathbf{A}^\top \mathbf{W} \mathbf{A} \right). \end{aligned} \tag{48}$$

Hence, if we can efficiently implement each step of MPC with these matrices, it will return a reweighting satisfying (42), which yields a trace-to-bottom eigenvalue ratio approximating $\tau_i^\star$ to a $1 + \epsilon$ factor. We remark that in the algorithm parameterization, we have $\rho = \tau_i^\star$. Moreover, all of the packing gradient computations in (44) are one-dimensional and hence amount to vector operations, so we will only discuss the computation of covering gradients.

Next, observe that Assumption 1 guarantees that for all intermediate reweightings $\mathbf{W}$ computed by the algorithm and $R = O(\log n\tau_i^\star)$, $\lambda_{\max}(\sum_{i \in [n]} w_i \mathbf{P}_i) = \operatorname{Tr}(\mathbf{A}^\top \mathbf{W} \mathbf{A}) \leq R$. This implies that the trace of the matrix involved in covering gradient computations is always bounded:

$$\operatorname{Tr}\left( \sum_{i \in [n]} w_i \mathbf{C}_i \right) = \tau_i^\star \operatorname{Tr}\left( \mathbf{A}^\top \mathbf{W} \mathbf{A} \right) \leq \tau_i^\star R. \tag{49}$$

To implement the covering gradient computations, we appropriately modify Lemmas 2 and 3 to use the rational approximation in Fact 7 instead of the polynomial approximation in Fact 2. It is straightforward to check that the degree of the rational approximation required is $\Delta = O(\log n\tau_i^\star)$.

Moreover, each of the $\Delta$ linear systems which Fact 7 requires us to solve is in the matrix

$$\mathbf{M} := \mathbf{I} + \frac{\sum_{i \in [n]} w_i \mathbf{C}_i}{\Delta},$$

which by (49) and the fact that $\mathbf{I}$ has all eigenvalues 1, has $\tau(\mathbf{M}) = O(\tau_i^\star)$. Thus, we can apply Fact 8 to solve these linear systems in time

$$O\left(\left(n + \sqrt{d\tau_i^\star}\right) \cdot d \cdot \text{poly} \log \frac{n\tau_i^\star}{\delta}\right).$$

Here, we noted that the main fact that e.g. Lemmas 2 and 3 use is that the rational approximation approximates the exponential up to a $\text{poly}(n^{-1}, (\tau_i^\star)^{-1})$ multiple of the identity. Since all coefficients of the polynomial in Fact 7 are bounded by $\Delta^{O(\Delta)}$, the precision to which we need to apply Fact 8 to satisfy the requisite approximations is $\epsilon = \Delta^{-O(\Delta)}$, which only affects the runtime by polylogarithmic factors. Combining the cost of computing (44) with the iteration bound of Assumption 1, the overall runtime of our method for approximating $\tau_i^\star$ is

$$O\left(\left(n + \sqrt{d\tau_i^\star(\mathbf{A})}\right) \cdot d \cdot \text{poly} \log \frac{n\tau_i^\star(\mathbf{A})}{\delta}\right),$$

which matches Fact 8's runtime after rescaling in all parameters up to logarithmic factors.