# OpenReview forum: "Structured Semidefinite Programming for Recovering Structured Preconditioners"
_NeurIPS.cc/2023/Conference — NeurIPS 2023 poster_

### Official Review · Reviewer_2VGS · 2023-06-26

**Soundness:** 3 good
**Presentation:** 3 good
**Contribution:** 3 good
**Rating:** 6
**Confidence:** 3

**Summary:**

This paper develops a general preconditioning framework called matrix-dictionary recovery. This framework follows the matrix multiplicative weights update method and is applied to solve two classes of problems:
(1) Two diagonal preconditioning problems: outer scaling and inner scaling
This paper gives the first nontrivial approximation algorithms in nearly linear time, which is beyond the generic SDP solvers.
(2) Graph-structured matrix families
This paper proposes algorithms for perturbed Laplacian solver and the recovery of two structured matrices: M-matrix and Laplacian matrix.


**Strengths:**

(1) The proposed algorithm for computing diagonal preconditioners (Theorem 1 and Theorem 2) improves the method using general SDP solvers by $\text{poly}(d)$ factors when $\kappa_o^\star (\mathbf{K})$ and $\kappa_i^\star (\mathbf{A})$ are small.
(2) The proposed framework for solving graph-structured matrices (Theorem 3, 4, and 5) obtains $\widetilde{O}(n^2)$ running time and improves the state-of-the-art methods which have running time $O(n^\omega)$ by virtue of general linear systems solvers.
(2) This paper is well-written and easy to understand.


**Weaknesses:**

(1) The algorithms for diagonal preconditioning improve the existing method by a factor of $\text{poly}(d)$ at the cost of $(\kappa^\star)^{1.5}$. The applicable scenario is when $\kappa^\star$ is small but $\kappa$ is large.
(2) The algorithms for solving graph-structured matrices have time complexity $\widetilde{O}(n^2)$ that improves $O(n^\omega)$. They work for the case when the input is dense but in practice, the input is often sparse.
(3) Section 5 is placed at the end of the manuscript. It would be better to show the comparison with existing work in the Introduction part so that we can clearly know your contributions and advantages.

**Questions:**

The proposed method for graph-structured matrices has running time $\widetilde{O}(n^2)$ which is nearly linear when the input is dense.
(1) Can it be further improved when the input is sparse?
(2) The existing work has complexity $O(n^\omega)$. Is it still $O(n^\omega)$ when the input is sparse?

**Limitations:**

See Weaknesses.

---

> ### Author Rebuttal · Authors · 2023-08-09
>
> Thank you for your reviewing efforts. We are glad that you found the paper easy to understand.
>
> Regarding weakness (1), we note that our method computes a (constant-factor) optimal diagonal preconditioner, and typically algorithmic or statistical applications of diagonal preconditioning are interesting when the rescaled condition number $\kappa^\star$ is small (and hence the $\kappa^\star$ overhead of our runtime is also small). While we do not achieve the optimal dependence on $\kappa^\star$, we do outline an approach to improve our dependence to $\sqrt{\kappa^\star}$ in Appendix D of the supplement.
>
> Regarding weakness (3), we agree with your suggestion and will move various portions of Section 5 to appear shortly after the relevant results (i.e., Theorems 1-7) for improved readability.
>
> Regarding your questions concerning sparsity, the $n^2$ dependence in our algorithm is due to the inclusion of all $n^2$ single-edge Laplacians in our matrix dictionary, due to the possibility of there being any of the edges present. If the graph to be recovered has a known sparsity structure (i.e. it is known that all edges belong to a set $S$ with $|S| \ll n^2$), then we can restrict our matrix dictionary and obtain a runtime near-linear in $|S|$, but it is an interesting open direction to obtain a near-linear runtime on sparse graphs without this assumption. We note that in general, the inverse of the Laplacian of a sparse graph is not necessarily sparse. Further, in the case of M-matrices, the inverse is necessarily dense (Lemma 33, supplement).
>
> Finally, we discuss your question regarding existing work for sparse inputs. Current algorithms with runtimes listed as $\ge n^\omega$ are at least bottlenecked by solving sparse linear systems. Such algorithms with improved running times do exist in the recent literature (see e.g. “Solving Sparse Linear Systems Faster than Matrix Multiplication” and “Matrix anti-concentration inequalities with applications”), but these assume very sparse matrices, and yield runtimes which remain superquadratic (e.g., roughly $n^{2.27}$ time for $O(n)$-sparse matrices).

---

> > ### Comment · Reviewer_2VGS · 2023-08-15
> >
> > Thank you for your responses! I will retain my score.

---

### Official Review · Reviewer_pK6D · 2023-06-29

**Soundness:** 3 good
**Presentation:** 4 excellent
**Contribution:** 4 excellent
**Rating:** 8
**Confidence:** 3

**Summary:**

This paper studies preconditioning, which is one of the most important techniques in numerical linear algebra with numerous applications in optimization and machine learning. It proposes a general framework based on the matrix-dictionary recovery problem, where are given a matrix $M$ and $M_1,\dots, M_n$, and the goal is to find $w$ such that $\sum_{i\in [n]}w_i M_i$ approximates the optimal preconditioner for $M$. This paper develops a general-purpose solver for this problem. Then, they apply this framework to propose nearly linear-time algorithms for diagonal preconditioning (including outer scaling and inner scaling), semi-random regression, and structured liner systems (including dense matrix approximated by graph Laplacians, dense inverse Laplacians, and dense inverse M-matrices). Technically, their meta-solver for the matrix-dictionary recovery problem is based on the matrix multiplicative weights update method and employs packing SDP solvers to form the gain matrices. They also use other techniques, such as JL sketching, the homotopy method, etc., to implement their solvers.

**Strengths:**

This research question considered in this paper is crucial for many applications in both theory and practice. Their results show that a bunch of numerical linear algebra problems can be solved in nearly linear time, which was not known before. More specifically, for the diagonal preconditioning problem, prior to this work, we did not know how to $O(1)$-approximate the optimal condition number of the rescaled matrix without using a generic SDP solver, which takes $\Omega(d^{3.5})$-time. They improve it to a linear dependence on the number of non-zero entries of the matrix ($\leq d^2$). In addition, they also provide a tighter analysis of the Jacobi preconditioning method. For numerically solving linear systems, when the matrix is a perturbed Laplacians, prior to this work, the only approach is to apply generic matrix multiplication, which runs in $d^{\omega}$-time. Their new algorithm runs in about $d^2$-time when the matrix is well-approximated by some unknown Laplacian. Moreover, their algorithm also generalizes to larger families of dense matrices, which are very useful in many different applications. The framework of matrix dictionary recovery is also very interesting and useful. There are some previous results concerning the isotropic case with some restrictions on the condition number. In this paper, their solver works not only for the non-isotropic case but also for general condition numbers. This framework and the general-purpose solver will be very helpful in future studies of numerical linear algebra algorithms.

**Weaknesses:**

The running times of their preconditioning algorithms depend on $\kappa^{1.5}$, which may not be optimal. Moreover, the accuracy dependence is $poly(1/\epsilon)$. Also, the paper does not discuss the practical implications of their algorithms, e.g., is it possible to implement them in practice?


**Questions:**

In Algorithm 1, it is required that $\lambda_{\max}(M_i)\in [1,2]$. Is it just for convenience?

What is the runtime bottleneck of Algorithm 1?

**Limitations:**

The limitations are addressed.

---

> ### Author Rebuttal · Authors · 2023-08-09
>
> Thank you for your encouraging review; we are similarly optimistic of the utility of the tools we provide for future numerical linear algebra problems. Regarding Algorithm 1, indeed the eigenvalue assumption was just for simplicity in stating our error bounds. Each $\mathbf{M}_i$ can always be rescaled to have maximum eigenvalue in $[1, 2]$. This can be accomplished via the power method (see Fact 3 in the supplement), whose runtime cost is dominated by other components of the algorithm. In particular, as you asked, the runtime bottleneck of Algorithm 1 (in terms of polylogarithmic factors and factors of $\epsilon^{-1}$) is the cost of calling approximate packing SDP oracles from prior work on Line 7, though this is an active area of research and any improvements therein would also reflect in our algorithm’s runtime. We also wanted to mention that while we do not achieve the optimal dependence on $\kappa^\star$, we do outline an approach to improve our dependence to $\sqrt{\kappa^\star}$ in Appendix D of the supplement.

---

### Official Review · Reviewer_1BXc · 2023-06-30

**Soundness:** 2 fair
**Presentation:** 1 poor
**Contribution:** 2 fair
**Rating:** 3
**Confidence:** 1

**Summary:**

In this paper, a framework is presented to compute approximately-optimal preconditioners in order to solve linear systems.

In the case of diagonal preconditioning, an algorithm is provided whose runtime is (up to log factors) polynomial in the desired accuracy, optimal condition number of the re-scaled matrix (thus in the output conditioning) and the number of nonzero entries of the input matrix. This is explicitly stated in Theorem 1 and Theorem 2 for outer and inner scaling, respectively.

In the case of structured linear systems, an algorithm is provided whose runtime is (up to log factors) quadratic in the size of the input matrix.

The underlying algorithms are based on the so-called "matrix-dictionary" approximation semidefinite programs. They rely on these algorithms to solve a related problem called "matrix-dictionary recovery".


**Strengths:**

If they would be correct, the resulting algorithms improve current runtime of the best solvers given by SDP (by a factor of d sqrt(d)).

The comparison with existing literature seems well-explained.

**Weaknesses:**

Not being an expert in this field and with a very limited amount of time (too short to read the 63 pages of supplementary material), it is completely impossible to judge whether the framework is correct or not. The authors just state the theorems and do not provide neither proof sketches nor hints that could possibly convince the reader. I believe that this work, possibly sound and surely interesting, is not a good fit at all, considering the currently requested NeurIPS format.


**Questions:**

l43: What is an M-matrix? The definition should be recalled.

**Limitations:**

There is no limitation section provided by the authors in the main submission.

---

> ### Author Rebuttal · Authors · 2023-08-09
>
> We are glad you found our comparison to the literature well-explained, and that our results were interesting. We agree with your comment on M-matrices, which will be addressed in a revision.
>
> Like many theoretical papers that appear at NeurIPS, due to space constraints, the technical details of our proofs were moved to the supplemental material. We provided proof sketches for Theorems 6 and 7, the workhorse results for all of our preconditioning applications, on Page 7 (a full analysis of Algorithm 1 is in Pages 11-14 in the supplement). The remaining components of our framework (approximate matrix-vector access and recursive preconditioning), though technically tedious, are fairly routine in the literature. We believe our overall framework provides value to the community and should be verifiable to experts, and hence NeurIPS is an appropriate venue for publication. We hope our discussion elevates your view of our paper.

---

> > ### Comment · Reviewer_1BXc · 2023-08-17
> >
> > Thank you for your responses, I will retain my score.

---

### Official Review · Reviewer_HjkM · 2023-07-07

**Soundness:** 4 excellent
**Presentation:** 4 excellent
**Contribution:** 3 good
**Rating:** 7
**Confidence:** 3

**Summary:**

This is a theoretical paper that studies the problem of diagonal matrix preconditioning, where given a PSD matrix $A$, the goal is to find a (positive) diagonal scaling $W$, such that $WAW$ has a small condition number, given the promise that such scaling exists. This problem can be solved using SDP, but that is too expensive. Following previous work that uses spectral sparsification techniques, the authors give improved guarantees for this problem. In particular, previously it was only known how to compute an approximately optimal scaling when the resulting condition number is $\leq 1.1$, while the authors' result works for any condition number (and has a polynomial condition number dependency in the runtime). The authors use the matrix multiplicative weights algorithm, together with sketching techniques to efficiently implement its iterations, leading to a near-linear time algorithm for bounded condition number. The results can be generalized to other problem classes like graph-structured preconditioning.

**Strengths:**

- Preconditioning is a fundamental problem and this is a significant and original contribution to it. It is significantly stronger than previously known results.
- The analysis looks correct and technically strong.
- The presentation is thorough and the intro is well written.

**Weaknesses:**

- There is no empirical evaluation of the proposed algorithm, e.g. compared to Jacobi iteration, and so the practical impact of this approach is not clear.

**Questions:**

- Do the results extend to the case when we have both outer and inner preconditioning?
- Since the algorithm is based on multiplicative weights, it seems natural to try and generalize this to the online setting. Are there any significant difficulties there?
- Are there potential implications for preconditioned optimizers such as Adagrad, Shampoo, etc?

---

> ### Author Rebuttal · Authors · 2023-08-09
>
> Thank you for your kind review of our paper and your insightful questions.
>
> We are currently not aware of a variant of our algorithm (and matrix dictionary recovery framework) which extends to simultaneous inner and outer scaling, though it is worth noting that prior work [QGH+22] does obtain such a result via semidefinite programming. Obtaining such a variant is indeed an interesting open problem for future work. In practice, one could potentially alternate inner and outer scaling algorithms until they cease to make progress (as the algorithm gives a constant-factor approximation to $\kappa^\star$ as a certificate), but it is unclear to us how to prove the convergence rate of such a procedure.
>
> Regarding online variants, this is another intriguing extension of our algorithm (we assume the reviewer means that e.g., the matrix dictionary arrives online and we maintain a set of weights). This extension does not appear compatible with the way multiplicative weights is currently used in our algorithm, where it is called as a regret minimization procedure to reweight an estimate in response to “gain matrices” produced through a packing oracle, roughly measuring currently violated constraints. Our reduction requires knowledge of the set of matrices used in the dictionary in advance. Nevertheless, finding some type of online variant of our results is indeed an interesting open problem.
>
> Regarding Adagrad, etc. diagonal approximations to the full gradient second moment matrix are discussed in the original Adagrad paper as a method to improve runtimes at a hopefully-low overhead on the regret guarantees. The quality of the regret bound for the diagonal variant of Adagrad degrades from that obtained for the full matrix variant. However, the degradation factor (a ratio between traces of matrix square roots) is not directly comparable to our notion of approximation (a relative condition number), so our method does not have immediate implications for Adagrad (or related preconditioning techniques, e.g., Shampoo). We find the questions of 1) obtaining a diagonal preconditioning algorithm directly targeting the degradation notion in Adagrad, and 2) whether our optimal diagonal preconditioners obtain improvements in practice for adaptive gradient methods, to be interesting directions for future exploration.
>
> Thank you again for raising these multiple interesting directions for future work. We may add some discussion of these directions in the final version. We hope that the appeal of these questions highlights the utility of this submission in facilitating future research.

---

> > ### Comment · Reviewer_HjkM · 2023-08-18
> >
> > Thank you very much for the detailed response.

---

### Author Rebuttal · Authors · 2023-08-09

Reviewers HjkM and pK6D asked about practical implementations of our algorithm. We agree that experiments are an important next step towards bringing the results of our paper to practice. Our primary motivation was theoretical: existing guarantees for the problems we study are off from linear time by (sometimes large) polynomial factors in the problem dimension. We close these gaps through our main results, which pave the way for more practical implementations. Our framework builds upon several tools in lines of active research with existing implementations, such as packing SDP solvers and sketching techniques, but which need to be combined in a careful way in order to obtain fast implementations of our overall algorithm; we leave this endeavor to interesting future work.

We also note that Appendix C of the supplement provides a new constant-factor optimal analysis of the Jacobi preconditioner, which is already widely-used in practice for outer scaling.

---

### Decision · Program_Chairs · 2023-09-21

**Decision:**

Accept (poster)

**Comment:**

Three reviewers agree unanimously that the paper is solid and worthy of acceptance. The one dissenting reviewer confesses that they are "not [an] expert in this field and with a very limited amount of time". From my own reading, I am also very impressed by the progress made in this work compared to prior work. Please incorporate reviewer feedback into the final camera-ready version.